# JACOBIAN DESCENT FOR MULTI-OBJECTIVE OPTIMIZATION

## ABSTRACT

Many optimization problems require balancing multiple conflicting objectives. As gradient descent is limited to single-objective optimization, we introduce its direct generalization: Jacobian descent (JD). This algorithm iteratively updates parameters using the Jacobian matrix of a vector-valued objective function, in which each row is the gradient of an individual objective. While several methods to combine gradients already exist in the literature, they are generally hindered when the objectives conflict. In contrast, we propose projecting gradients to fully avoid conflict while ensuring that they preserve an influence proportional to their norm. We prove significantly stronger convergence guarantees with this approach, supported by our empirical results. Our method also enables instance-wise risk minimization (IWRM), a novel learning paradigm in which the loss of each training example is considered a separate objective. Applied to simple image classification tasks, IWRM exhibits promising results compared to the direct minimization of the average loss. Additionally, we outline an efficient implementation of JD using the Gramian of the Jacobian matrix to reduce time and memory requirements.

## 1 INTRODUCTION

The field of multi-objective optimization studies minimization of vector-valued objective functions (Sawaragi et al., 1985; Ehrgott, 2005; Branke, 2008; Deb et al., 2016). In deep learning, a widespread approach to train a model with multiple objectives is to combine those into a scalar loss function minimized by stochastic gradient descent. While this method is simple, it comes at the expense of potentially degrading some individual objectives. Without prior knowledge of their relative importance, this is undesirable. In opposition, multi-objective optimization methods typically attempt to optimize all objectives simultaneously, without making arbitrary compromises: the goal is to find points for which no improvement can be made on some objectives without degrading others.

Early works have attempted to extend gradient descent (GD) to consider several objectives simultaneously, and thus several gradients (Fliege & Svaiter, 2000; Désidéri, 2012). Essentially, they propose a heuristic to prevent the degradation of any individual objective. Several other works have built upon this method, analyzing its convergence properties or extending it to a stochastic setting (Fliege et al., 2019; Poirion et al., 2017; Mercier et al., 2018). Later, this has been applied to multi-task learning to tackle conflict between tasks, illustrated by contradicting gradient directions (Sener & Koltun, 2018). Many studies have followed, proposing various other algorithms for the training of multi-task models (Yu et al., 2020; Liu et al., 2021a;b; Lin et al., 2021; Navon et al., 2022; Senushkin et al., 2023; Chen et al., 2020). They commonly rely on an aggregator that maps a collection of task-specific gradients (a Jacobian matrix) to a shared parameter update.

We propose to unify all such methods under the *Jacobian descent* (JD) algorithm, specified by an aggregator.[1] This algorithm aims to minimize a differentiable vector-valued function $\boldsymbol{f} : \mathbb{R}^n \to \mathbb{R}^m$ iteratively without relying on a scalarization of the objective. Under this formulation, the existing methods are simply distinguished by their aggregator. Consequently, studying its properties is essential for understanding the behavior and convergence of JD. Under significant conflict, existing aggregators often fail to provide strong convergence guarantees. To address this, we propose $\mathcal{A}_{\text{UPGrad}}$, specifically designed to resolve conflicts while naturally preserving the relative influence of individual gradients.

---

[1] Our library enabling JD with `PyTorch` is available at https://github.com/***/***

Furthermore, we introduce a novel stochastic variant of JD that enables the training of neural networks with a large number of objectives. This unlocks a particularly interesting perspective: considering the minimization of instance-wise loss vectors rather than the usual minimization of the average training loss. As this paradigm is a direct generalization of the well-known empirical risk minimization (ERM) (Vapnik, 1995), we name it *instance-wise risk minimization* (IWRM).

Our contributions are organized as follows: In Section 2, we formalize the JD algorithm and its stochastic variants. We then introduce three important aggregator properties and define $\mathcal{A}_{\text{UPGrad}}$ to satisfy them. In the smooth convex case, we show convergence of JD with $\mathcal{A}_{\text{UPGrad}}$ to the Pareto front. We present applications for JD and aggregators in Section 3, emphasizing the IWRM paradigm. We then discuss existing aggregators and analyze their properties in Section 4. In Section 5, we report experiments with IWRM optimized with stochastic JD with various aggregators. Lastly, we address computational efficiency in Section 6, giving a path towards an efficient implementation.

## 2 THEORETICAL FOUNDATION

A suitable partial order between vectors must be considered to enable multi-objective optimization. Throughout this paper, $\preceq$ denotes the relation defined for any pair of vectors $\boldsymbol{u}, \boldsymbol{v} \in \mathbb{R}^m$ as $\boldsymbol{u} \preceq \boldsymbol{v}$ whenever $u_i \leq v_i$ for all coordinates $i$. Similarly, $\prec$ is the relation defined by $\boldsymbol{u} \prec \boldsymbol{v}$ whenever $u_i < v_i$ for all coordinates $i$. Furthermore, $\boldsymbol{u} \precneqq \boldsymbol{v}$ indicates that both $\boldsymbol{u} \preceq \boldsymbol{v}$ and $\boldsymbol{u} \neq \boldsymbol{v}$ hold. The Euclidean vector norm and the Frobenius matrix norm are denoted by $\|\cdot\|$ and $\|\cdot\|_{\text{F}}$, respectively. Finally, for any $m \in \mathbb{N}$, the symbol $[m]$ represents the range $\{i \in \mathbb{N} : 1 \leq i \leq m\}$.

### 2.1 JACOBIAN DESCENT

In the following, we introduce Jacobian descent, a natural extension of gradient descent supporting the optimization of vector-valued functions.

Suppose that $\boldsymbol{f} : \mathbb{R}^n \to \mathbb{R}^m$ is continuously differentiable. Let $\mathcal{J}\boldsymbol{f}(\boldsymbol{x}) \in \mathbb{R}^{m \times n}$ be the Jacobian matrix of $\boldsymbol{f}$ at $\boldsymbol{x}$, i.e.

$$\mathcal{J}\boldsymbol{f}(\boldsymbol{x}) = \begin{bmatrix} \nabla f_1(\boldsymbol{x})^\top \\ \nabla f_2(\boldsymbol{x})^\top \\ \vdots \\ \nabla f_m(\boldsymbol{x})^\top \end{bmatrix} = \begin{bmatrix} \frac{\partial}{\partial x_1} f_1(\boldsymbol{x}) & \frac{\partial}{\partial x_2} f_1(\boldsymbol{x}) & \cdots & \frac{\partial}{\partial x_n} f_1(\boldsymbol{x}) \\ \frac{\partial}{\partial x_1} f_2(\boldsymbol{x}) & \frac{\partial}{\partial x_2} f_2(\boldsymbol{x}) & \cdots & \frac{\partial}{\partial x_n} f_2(\boldsymbol{x}) \\ \vdots & \vdots & \ddots & \vdots \\ \frac{\partial}{\partial x_1} f_m(\boldsymbol{x}) & \frac{\partial}{\partial x_2} f_m(\boldsymbol{x}) & \cdots & \frac{\partial}{\partial x_n} f_m(\boldsymbol{x}) \end{bmatrix} \tag{1}$$

Given $\boldsymbol{x}, \boldsymbol{y} \in \mathbb{R}^n$, Taylor's theorem yields

$$\boldsymbol{f}(\boldsymbol{x} + \boldsymbol{y}) = \boldsymbol{f}(\boldsymbol{x}) + \mathcal{J}\boldsymbol{f}(\boldsymbol{x}) \cdot \boldsymbol{y} + o(\|\boldsymbol{y}\|), \tag{2}$$

where $o(\|\boldsymbol{y}\|)$ indicates that $\lim_{\|\boldsymbol{y}\| \to 0} \frac{\boldsymbol{f}(\boldsymbol{x}+\boldsymbol{y}) - \boldsymbol{f}(\boldsymbol{x}) - \mathcal{J}\boldsymbol{f}(\boldsymbol{x}) \cdot \boldsymbol{y}}{\|\boldsymbol{y}\|} = \boldsymbol{0}$. The term $\boldsymbol{f}(\boldsymbol{x}) + \mathcal{J}\boldsymbol{f}(\boldsymbol{x}) \cdot \boldsymbol{y}$ is the first-order Taylor approximation of $\boldsymbol{f}(\boldsymbol{x} + \boldsymbol{y})$. Via this approximation, we aim to select a small update $\boldsymbol{y}$ that reduces $\boldsymbol{f}(\boldsymbol{x} + \boldsymbol{y})$, ideally achieving $\boldsymbol{f}(\boldsymbol{x} + \boldsymbol{y}) \preceq \boldsymbol{f}(\boldsymbol{x})$. As the approximation depends on $\boldsymbol{y}$ only through $\mathcal{J}\boldsymbol{f}(\boldsymbol{x}) \cdot \boldsymbol{y}$, selecting the update based on the Jacobian is natural. A mapping $\mathcal{A} : \mathbb{R}^{m \times n} \to \mathbb{R}^n$ reducing such a matrix into a vector is called an *aggregator*. For any $J \in \mathbb{R}^{m \times n}$, $\mathcal{A}(J)$ is called the *aggregation* of $J$ by $\mathcal{A}$.

To minimize $\boldsymbol{f}$, consider the update $\boldsymbol{y} = -\eta \mathcal{A}\big(\mathcal{J}\boldsymbol{f}(\boldsymbol{x})\big)$, where $\eta$ is an appropriate step size, and $\mathcal{A}$ is an appropriate aggregator. Jacobian descent simply consists in applying this update iteratively, as shown in Algorithm 1. To put it into perspective, we also provide a minimal version of GD in Algorithm 2. Remarkably, when $m = 1$, the Jacobian has a single row, so GD is a special case of JD where the aggregator is the identity.

| **Algorithm 1:** Jacobian descent with aggregator $\mathcal{A}$ | **Algorithm 2:** Gradient descent |
|---|---|
| **Input:** $\boldsymbol{x} \in \mathbb{R}^n, 0 < \eta, T \in \mathbb{N}, \mathcal{A} : \mathbb{R}^{m \times n} \to \mathbb{R}^n$ | **Input:** $\boldsymbol{x} \in \mathbb{R}^n, 0 < \eta, T \in \mathbb{N}$ |
| **for** $t \leftarrow 1$ **to** $T$ **do** | **for** $t \leftarrow 1$ **to** $T$ **do** |
| $\quad \boldsymbol{x} \leftarrow \boldsymbol{x} - \eta \mathcal{A}\big(\mathcal{J}\boldsymbol{f}(\boldsymbol{x})\big)$ | $\quad \boldsymbol{x} \leftarrow \boldsymbol{x} - \eta \nabla f(\boldsymbol{x})$ |
| **Output:** $\boldsymbol{x}$ | **Output:** $\boldsymbol{x}$ |

Note that other gradient-based optimization algorithms, e.g. Adam (Kingma & Ba, 2014), can similarly be extended to the multi-objective case.

In some settings, the exact computation of the update can be prohibitively slow or even intractable. When dealing with a single objective, stochastic gradient descent (SGD) replaces the gradient $\nabla f(\boldsymbol{x})$ with some estimation. More generally, *stochastic Jacobian descent* (SJD) relies on estimates of the aggregation of the Jacobian. One approach, that we call *stochastically estimated Jacobian descent* (SEJD), is to compute and aggregate an estimation of the Jacobian. Alternatively, when the number of objectives is very large, we propose to aggregate a matrix whose rows are a random subset of the rows of the true Jacobian. We call this approach *stochastic sub-Jacobian descent* (SSJD).

## 2.2 Desirable properties for aggregators

An inherent challenge of multi-objective optimization is to manage conflicting objectives (Sener & Koltun, 2018; Yu et al., 2020; Liu et al., 2021a). Substituting the update $\boldsymbol{y} = -\eta \mathcal{A}\big(\mathcal{J}\boldsymbol{f}(\boldsymbol{x})\big)$ into the first-order Taylor approximation $\boldsymbol{f}(\boldsymbol{x}) + \mathcal{J}\boldsymbol{f}(\boldsymbol{x}) \cdot \boldsymbol{y}$ yields $\boldsymbol{f}(\boldsymbol{x}) - \eta \mathcal{J}\boldsymbol{f}(\boldsymbol{x}) \cdot \mathcal{A}\big(\mathcal{J}\boldsymbol{f}(\boldsymbol{x})\big)$. In particular, if $\boldsymbol{0} \leq \mathcal{J}\boldsymbol{f}(\boldsymbol{x}) \cdot \mathcal{A}\big(\mathcal{J}\boldsymbol{f}(\boldsymbol{x})\big)$, then no coordinate of the approximation of $\boldsymbol{f}$ will increase. A pair of vectors $\boldsymbol{x}, \boldsymbol{y} \in \mathbb{R}^n$ is said to *conflict* if $\boldsymbol{x}^\top \boldsymbol{y} < 0$. Hence, for a sufficiently small $\eta$, if any row of $\mathcal{J}\boldsymbol{f}(\boldsymbol{x})$ conflicts with $\mathcal{A}\big(\mathcal{J}\boldsymbol{f}(\boldsymbol{x})\big)$, the corresponding coordinate of $\boldsymbol{f}$ will increase. When minimizing $\boldsymbol{f}$, avoiding conflict between the aggregation and any gradient is thus desirable, motivating the first property.

**Definition 1** (Non-conflicting). Let $\mathcal{A} : \mathbb{R}^{m \times n} \to \mathbb{R}^n$ be an aggregator. If for all $J \in \mathbb{R}^{m \times n}$, $\boldsymbol{0} \leq J \cdot \mathcal{A}(J)$, then $\mathcal{A}$ is said to be *non-conflicting*.

For any collection of vectors $C \subseteq \mathbb{R}^n$, the *dual cone* of $C$ is $\{\boldsymbol{x} \in \mathbb{R}^n : \forall \boldsymbol{y} \in C, 0 \leq \boldsymbol{x}^\top \boldsymbol{y}\}$ (Boyd & Vandenberghe, 2004). Notice that an aggregator $\mathcal{A}$ is non-conflicting if and only if for any $J$, $\mathcal{A}(J)$ is in the dual cone of the rows of $J$.

In a step of GD, the update scales proportionally to the gradient norm. Small gradients thus lead to small updates, and conversely, large gradients lead to large updates. To maintain coherence with GD, it would be natural that the rows of the Jacobian also contribute to the aggregation proportionally to their norm. Scaling each row of $\mathcal{J}\boldsymbol{f}(\boldsymbol{x})$ by the corresponding element of some vector $\boldsymbol{c} \in \mathbb{R}^m$ yields $\mathrm{diag}(\boldsymbol{c}) \cdot \mathcal{J}\boldsymbol{f}(\boldsymbol{x})$. This insight can then be formalized as the following property.

**Definition 2** (Linear under scaling). Let $\mathcal{A} : \mathbb{R}^{m \times n} \to \mathbb{R}^n$ be an aggregator. If for all $J \in \mathbb{R}^{m \times n}$, the mapping from any $\boldsymbol{0} < \boldsymbol{c} \in \mathbb{R}^m$ to $\mathcal{A}\big(\mathrm{diag}(\boldsymbol{c}) \cdot J\big)$ is linear in $\boldsymbol{c}$, then $\mathcal{A}$ is said to be *linear under scaling*.

Finally, as $\|\boldsymbol{y}\|$ decreases asymptotically to 0, the precision of the first-order Taylor approximation $\boldsymbol{f}(\boldsymbol{x}) + \mathcal{J}\boldsymbol{f}(\boldsymbol{x}) \cdot \boldsymbol{y}$ improves, as highlighted in (2). The projection $\boldsymbol{y}'$ of any candidate update $\boldsymbol{y}$ onto the span of the rows of $\mathcal{J}\boldsymbol{f}(\boldsymbol{x})$ satisfies $\mathcal{J}\boldsymbol{f}(\boldsymbol{x}) \cdot \boldsymbol{y}' = \mathcal{J}\boldsymbol{f}(\boldsymbol{x}) \cdot \boldsymbol{y}$ and $\|\boldsymbol{y}'\| \leq \|\boldsymbol{y}\|$, so this projection decreases the norm of the update while preserving the value of the approximation. Without additional information about $\boldsymbol{f}$, it is thus reasonable to select $\boldsymbol{y}$ directly in the row span of $\mathcal{J}\boldsymbol{f}(\boldsymbol{x})$, i.e. to have a vector of weights $\boldsymbol{w} \in \mathbb{R}^m$ satisfying $\boldsymbol{y} = \mathcal{J}\boldsymbol{f}(\boldsymbol{x})^\top \cdot \boldsymbol{w}$. This yields the last desirable property.

**Definition 3** (Weighted). Let $\mathcal{A} : \mathbb{R}^{m \times n} \to \mathbb{R}^n$ be an aggregator. If for all $J \in \mathbb{R}^{m \times n}$, there exists $\boldsymbol{w} \in \mathbb{R}^m$ satisfying $\mathcal{A}(J) = J^\top \cdot \boldsymbol{w}$, then $\mathcal{A}$ is said to be *weighted*.

## 2.3 Unconflicting projection of gradients

We now define the *unconflicting projection of gradients* aggregator $\mathcal{A}_{\mathrm{UPGrad}}$, specifically designed to be non-conflicting, linear under scaling, and weighted. In essence, it projects each gradient onto the dual cone of the rows of the Jacobian and averages the results, as illustrated in Figure 1a.

For any $J \in \mathbb{R}^{m \times n}$ and $\boldsymbol{x} \in \mathbb{R}^n$, the *projection* of $\boldsymbol{x}$ onto the dual cone of the rows of $J$ is

$$\pi_J(\boldsymbol{x}) = \underset{\boldsymbol{y} \in \mathbb{R}^n : \, \boldsymbol{0} \leq J\boldsymbol{y}}{\arg\min} \|\boldsymbol{y} - \boldsymbol{x}\|^2. \tag{3}$$

Denoting by $\boldsymbol{e}_i \in \mathbb{R}^m$ the $i$th standard basis vector, $J^\top \boldsymbol{e}_i$ is the $i$th row of $J$. $\mathcal{A}_{\mathrm{UPGrad}}$ is defined as

$$\mathcal{A}_{\mathrm{UPGrad}}(J) = \frac{1}{m} \sum_{i \in [m]} \pi_J(J^\top \boldsymbol{e}_i). \tag{4}$$

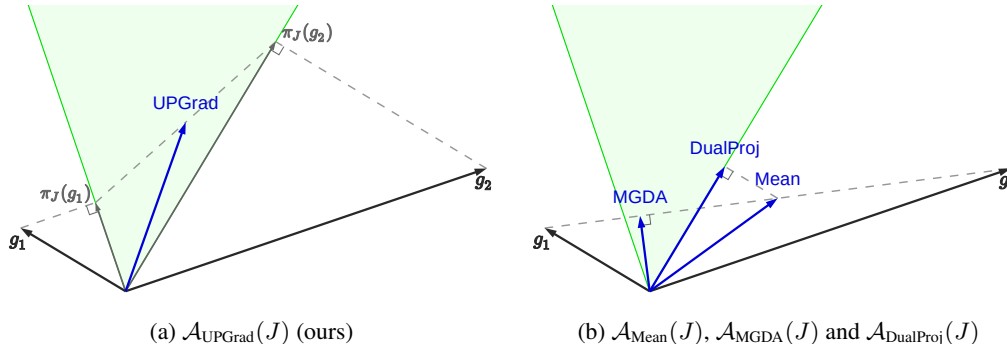

(a) $\mathcal{A}_{\mathrm{UPGrad}}(J)$ (ours)        (b) $\mathcal{A}_{\mathrm{Mean}}(J)$, $\mathcal{A}_{\mathrm{MGDA}}(J)$ and $\mathcal{A}_{\mathrm{DualProj}}(J)$

Figure 1: Aggregation of $J = [\boldsymbol{g}_1\ \boldsymbol{g}_2]^\top \in \mathbb{R}^{2\times 2}$ by four different aggregators. The dual cone of $\{\boldsymbol{g}_1, \boldsymbol{g}_2\}$ is represented in green.
(a) $\mathcal{A}_{\mathrm{UPGrad}}$ projects $\boldsymbol{g}_1$ and $\boldsymbol{g}_2$ onto the dual cone and averages the results.
(b) The mean $\mathcal{A}_{\mathrm{Mean}}(J) = \frac{1}{2}(\boldsymbol{g}_1 + \boldsymbol{g}_2)$ conflicts with $\boldsymbol{g}_1$. $\mathcal{A}_{\mathrm{DualProj}}$ projects this mean onto the dual cone, so it lies on its boundary. $\mathcal{A}_{\mathrm{MGDA}}(J)$ is almost orthogonal to $\boldsymbol{g}_2$ because of its larger norm.

Since the dual cone is convex, it is closed under positive combinations of its elements. For any $J$, $\mathcal{A}_{\mathrm{UPGrad}}(J)$ is thus always in the dual cone of the rows of $J$, so $\mathcal{A}_{\mathrm{UPGrad}}$ is non-conflicting. Note that if no pair of gradients conflicts, $\mathcal{A}_{\mathrm{UPGrad}}$ simply averages the rows of the Jacobian.

Since $\pi_J$ is a projection onto a closed convex cone, if $\boldsymbol{x} \in \mathbb{R}^n$ and $0 < a \in \mathbb{R}$, then $\pi_J(a \cdot \boldsymbol{x}) = a \cdot \pi_J(\boldsymbol{x})$. By (4), $\mathcal{A}_{\mathrm{UPGrad}}$ is thus linear under scaling.

When $n$ is large, the projection in (3) is prohibitively expensive to compute. An alternative but equivalent approach is to use its dual formulation, which is independent of $n$.

**Proposition 1.** Let $J \in \mathbb{R}^{m\times n}$. For any $\boldsymbol{u} \in \mathbb{R}^m$, $\pi_J(J^\top \boldsymbol{u}) = J^\top \boldsymbol{w}$ with

$$\boldsymbol{w} \in \underset{\boldsymbol{v}\in\mathbb{R}^m:\ \boldsymbol{u}\preceq\boldsymbol{v}}{\arg\min}\ \boldsymbol{v}^\top J J^\top \boldsymbol{v}. \tag{5}$$

*Proof.* See Appendix A.2.

The problem defined in (5) can be solved efficiently using a quadratic programming solver, such as those bundled in qpsolvers (Caron et al., 2024). For any $i \in [m]$, let $\boldsymbol{w}_i$ be given by (5) when substituting $\boldsymbol{u}$ with $\boldsymbol{e}_i$. Then, by Proposition 1,

$$\mathcal{A}_{\mathrm{UPGrad}}(J) = J^\top \left( \frac{1}{m} \sum_{i\in[m]} \boldsymbol{w}_i \right). \tag{6}$$

This provides an efficient implementation of $\mathcal{A}_{\mathrm{UPGrad}}$ and proves that it is weighted. $\mathcal{A}_{\mathrm{UPGrad}}$ can also be easily extended to incorporate a vector of preferences by replacing the average in (4) and (6) by a weighted sum with positive weights. This extension remains non-conflicting, linear under scaling, and weighted.

## 2.4 CONVERGENCE TO THE PARETO FRONT

We now provide theoretical convergence guarantees of JD with $\mathcal{A}_{\mathrm{UPGrad}}$ when minimizing some $\boldsymbol{f} : \mathbb{R}^n \to \mathbb{R}^m$ satisfying standard assumptions. If for a given $\boldsymbol{x} \in \mathbb{R}^n$, there exists no $\boldsymbol{y} \in \mathbb{R}^n$ satisfying $\boldsymbol{f}(\boldsymbol{y}) \precneqq \boldsymbol{f}(\boldsymbol{x})$, then $\boldsymbol{x}$ is said to be *Pareto optimal*. The set $X^* \subseteq \mathbb{R}^n$ of Pareto optimal points is called the *Pareto set*, and its image $\boldsymbol{f}(X^*)$ is called the *Pareto front*.

Whenever $\boldsymbol{f}(\lambda\boldsymbol{x} + (1-\lambda)\boldsymbol{y}) \preceq \lambda\boldsymbol{f}(\boldsymbol{x}) + (1-\lambda)\boldsymbol{f}(\boldsymbol{y})$ holds for any pair of vectors $\boldsymbol{x}, \boldsymbol{y} \in \mathbb{R}^n$ and any $\lambda \in [0,1]$, $\boldsymbol{f}$ is said to be $\preceq$-*convex*. Moreover, $\boldsymbol{f}$ is said to be $\beta$-*smooth* whenever $\left\| \mathcal{J}\boldsymbol{f}(\boldsymbol{x}) - \mathcal{J}\boldsymbol{f}(\boldsymbol{y}) \right\|_{\mathrm{F}} \leq \beta\|\boldsymbol{x} - \boldsymbol{y}\|$ holds for any pair of vectors $\boldsymbol{x}, \boldsymbol{y} \in \mathbb{R}^n$.

**Theorem 1.** Let $\boldsymbol{f} : \mathbb{R}^n \to \mathbb{R}^m$ be a $\beta$-smooth and $\preceq$-convex function. Suppose that the Pareto front $\boldsymbol{f}(X^*)$ is bounded and that for any $\boldsymbol{x} \in \mathbb{R}^n$, there is $\boldsymbol{x}^* \in X^*$ satisfying $\boldsymbol{f}(\boldsymbol{x}^*) \preceq \boldsymbol{f}(\boldsymbol{x})$.[2] Let $\boldsymbol{x}_1 \in \mathbb{R}^n$, and for all $t \geq 1$, $\boldsymbol{x}_{t+1} = \boldsymbol{x}_t - \eta \mathcal{A}_{\mathrm{UPGrad}}\big(\mathcal{J}\boldsymbol{f}(\boldsymbol{x}_t)\big)$, with $\eta = \frac{1}{\beta\sqrt{m}}$. Let $\boldsymbol{w}_t$ be the weights defining $\mathcal{A}_{\mathrm{UPGrad}}\big(\mathcal{J}\boldsymbol{f}(\boldsymbol{x}_t)\big)$ as per (6), i.e. $\mathcal{A}_{\mathrm{UPGrad}}\big(\mathcal{J}\boldsymbol{f}(\boldsymbol{x}_t)\big) = \mathcal{J}\boldsymbol{f}(\boldsymbol{x}_t)^\top \cdot \boldsymbol{w}_t$. If $\boldsymbol{w}_t$ is bounded, then $\boldsymbol{f}(\boldsymbol{x}_t)$ converges to $\boldsymbol{f}(\boldsymbol{x}^*)$ for some $\boldsymbol{x}^* \in X^*$. In other words, $\boldsymbol{f}(\boldsymbol{x}_t)$ converges to the Pareto front.

*Proof.* See Appendix A.3. ∎

Empirically, $\boldsymbol{w}_t$ appears to converge to some $\boldsymbol{w}^* \in \mathbb{R}^m$ satisfying both $\boldsymbol{0} \prec \boldsymbol{w}^*$ and $\mathcal{J}\boldsymbol{f}(\boldsymbol{x}^*)^\top \boldsymbol{w}^* = \boldsymbol{0}$. This suggests that the boundedness of $\boldsymbol{w}_t$ could be relaxed or even removed from the set of assumptions of Theorem 1.

Another commonly studied type of convergence for multi-objective optimization is convergence to a stationary point. If for a given $\boldsymbol{x} \in \mathbb{R}^n$, there exists $\boldsymbol{0} \preceq \boldsymbol{w}$ satisfying $\mathcal{J}\boldsymbol{f}(\boldsymbol{x})^\top \boldsymbol{w} = 0$ then $\boldsymbol{x}$ is said to be *Pareto stationary*. Even though every Pareto optimal point is Pareto stationary, the converse does not hold, even in the convex case. The function $\begin{bmatrix} x & y \end{bmatrix}^\top \mapsto \begin{bmatrix} x^2 & y^2 \end{bmatrix}^\top$ illustrates this discrepancy. Its Pareto set only contains the origin, but its set of Pareto stationary points is the union of the two axes. Despite being necessary, convergence to a Pareto stationary point is thus not a sufficient condition for optimality and, hence, constitutes a rather weak guarantee. To the best of our knowledge, $\mathcal{A}_{\mathrm{UPGrad}}$ is the first non-conflicting aggregator that provably converges to the Pareto front in the smooth convex case.

In addition to the asymptotic convergence guarantees of Theorem 1, Appendix A.3 provides the following rate of convergence for any number of iterations $T \in \mathbb{N}$:

$$\frac{1}{T} \sum_{t \in [T]} \boldsymbol{w}_t^\top \big(\boldsymbol{f}(\boldsymbol{x}_t) - \boldsymbol{f}(\boldsymbol{x}^*)\big) \leq \frac{\sqrt{m}}{T} \left( \|\boldsymbol{f}(\boldsymbol{x}_1) - \boldsymbol{f}(\boldsymbol{x}^*)\| + \frac{\beta}{2}\|\boldsymbol{x}_1 - \boldsymbol{x}^*\|^2 \right).$$

Although this result does not directly provide a convergence rate for $\boldsymbol{f}$, the bound $\boldsymbol{1} \preceq \boldsymbol{w}_t$, suggests a convergence rate of $\mathcal{O}\left(\frac{1}{T}\right)$, offering valuable insight into the algorithm's asymptotic behavior.

## 3 APPLICATIONS

**Instance-wise risk minimization.** In machine learning, we generally have access to a training set consisting of $m$ examples. The goal of empirical risk minimization (ERM) (Vapnik, 1995) is simply to minimize the average loss over the whole training set. More generally, instance-wise risk minimization (IWRM) considers the loss associated with each training example as a distinct objective. Formally, if $\boldsymbol{x} \in \mathbb{R}^n$ are the parameters of the model and $f_i(\boldsymbol{x})$ is the loss associated to the $i$th example, the respective objective functions of ERM and IWRM are:

$$\text{(Empirical risk)} \qquad \bar{f}(\boldsymbol{x}) = \frac{1}{m} \sum_{i \in [m]} f_i(\boldsymbol{x}) \tag{7}$$

$$\text{(Instance-wise risk)} \qquad \boldsymbol{f}(\boldsymbol{x}) = \begin{bmatrix} f_1(\boldsymbol{x}) & f_2(\boldsymbol{x}) & \cdots & f_m(\boldsymbol{x}) \end{bmatrix}^\top \tag{8}$$

Naively using GD for ERM is inefficient in most practical cases, so a prevalent alternative is to use SGD or one of its variants. Similarly, using JD for IWRM is typically intractable. Indeed, it would require computing a Jacobian matrix with one row per training example at each iteration. In contrast, we can use the Jacobian of a random batch of training example losses. Since it consists of a subset of the rows of the full Jacobian, this approach is a form of stochastic sub-Jacobian descent, as introduced in Section 2.1. IWRM can also be extended to cases where each $f_i$ is a vector-valued function. The objective would then be the concatenation of the losses of all examples.

**Multi-task learning.** In multi-task learning, a single model is trained to perform several related tasks simultaneously, leveraging shared representations to improve overall performance (Ruder, 2017). At its core, multi-task learning is a multi-objective optimization problem (Sener & Koltun,

---

[2]This condition is a generalization to the case $m \geq 1$ of the existence of a minimizer $\boldsymbol{x}^* \in \mathbb{R}^n$ when $m = 1$.

2018), making it a straightforward application for Jacobian descent. Yet, the conflict between tasks is often too limited to justify the overhead of computing all task-specific gradients, i.e. the whole Jacobian (Kurin et al., 2022; Xin et al., 2022). In such cases, a practical approach is to minimize some linear scalarization of the objectives using an SGD-based method. Nevertheless, we believe that a setting with inherent conflict between tasks naturally prescribes Jacobian descent with a non-conflicting aggregator. We analyze several related works applied to multi-task learning in Section 4.

**Adversarial training.** In adversarial domain adaptation, the feature extractor of a model is trained with two conflicting objectives: The features should be helpful for the main task and should be unable to discriminate the domain of the input (Ganin et al., 2016). Likewise, in adversarial fairness, the feature extractor is trained to both minimize the predictability of sensitive attributes, such as race or gender, and maximize the performance on the main task (Adel et al., 2019). Combining the corresponding gradients with a non-conflicting aggregator could enhance the optimization of such methods. We believe that the training of generative adversarial networks (Goodfellow et al., 2014) could be similarly formulated as a multi-objective optimization problem. The generator and discriminator could then be jointly optimized with JD.

**Momentum-based optimization.** In gradient-based single-objective optimization, several methods use some form of gradient momentum to improve their convergence speed (Polyak, 1964). Essentially, their updates consider an exponential moving average of past gradients rather than just the last one. An appealing idea is to modify those algorithms to make them combine the gradient and the momentum with some aggregator, such as $\mathcal{A}_{\text{UPGrad}}$, instead of summing them. This would apply to many popular optimizers, like SGD with Nesterov momentum (Nesterov, 1983), Adam (Kingma & Ba, 2014), AdamW (Loshchilov & Hutter, 2019) and NAdam (Dozat, 2016).

**Distributed optimization.** In a distributed data-parallel setting with multiple machines or multiple GPUs, model updates are computed in parallel. This can be viewed as multi-objective optimization with one objective per data share. Rather than the typical averaging, a specialized aggregator, such as $\mathcal{A}_{\text{UPGrad}}$, could thus combine the model updates. This consideration can even be extended to federated learning, in which multiple entities participate in the training of a common model from their own private data by sharing model updates (Kairouz et al., 2021). In this setting, as security is one of the main challenges, the non-conflicting property of the aggregator could be key.

## 4 EXISTING AGGREGATORS

In the context of multi-task learning, several works have proposed iterative optimization algorithms based on the combination of task-specific gradients (Sener & Koltun, 2018; Yu et al., 2020; Liu et al., 2021b;a; Lin et al., 2021; Navon et al., 2022; Senushkin et al., 2023). These methods can be formulated as variants of JD parameterized by different aggregators. More specifically, since the gradients are stochastically estimated from batches of data, these are cases of what we call SEJD. In the following, we briefly present the most prominent aggregators and summarize their properties in Table 1. As a baseline, we also consider $\mathcal{A}_{\text{Mean}}$, which simply averages the rows of the Jacobian. Their formal definitions are provided in Appendix B. Some of them are also illustrated in Figure 1b.

$\mathcal{A}_{\text{RGW}}$ aggregates the matrix using a random vector of weights (Lin et al., 2021). $\mathcal{A}_{\text{MGDA}}$ gives the aggregation that maximizes the smallest improvement (Désidéri, 2012; Sener & Koltun, 2018; Fliege & Svaiter, 2000). $\mathcal{A}_{\text{CAGrad}}$ maximizes the smallest improvement in a ball around the average gradient whose radius is parameterized by $c \in [0, 1[$ (Liu et al., 2021a). $\mathcal{A}_{\text{PCGrad}}$ projects each gradient onto the orthogonal hyperplane of other gradients in case of conflict, iteratively and in a random order (Yu et al., 2020). It is, however, only non-conflicting when $m \le 2$, in which case $\mathcal{A}_{\text{PCGrad}} = m \cdot \mathcal{A}_{\text{UPGrad}}$. IMTL-G is a method to balance some gradients with impartiality (Liu et al., 2021b). It is only defined for linearly independent gradients, but we generalize it as a formal aggregator, denoted $\mathcal{A}_{\text{IMTL-G}}$, in Appendix B.6. Aligned-MTL orthonormalizes the Jacobian and weights its rows according to some preferences (Senushkin et al., 2023). We denote by $\mathcal{A}_{\text{Aligned-MTL}}$ this method with uniform preferences. $\mathcal{A}_{\text{Nash-MTL}}$ aggregates Jacobians by finding the Nash equilibrium between task-specific gradients (Navon et al., 2022). Lastly, the GradDrop layer (Chen et al., 2020) defines a custom backward pass that combines gradients with respect to some internal activation. The corresponding

aggregator, denoted $\mathcal{A}_{\text{GradDrop}}$, randomly drops out some gradient coordinates based on their sign and sums the remaining ones.

In the context of continual learning, to limit forgetting, an idea is to project the gradient onto the dual cone of gradients computed with past examples (Lopez-Paz & Ranzato, 2017). This idea can be translated into an aggregator that projects the mean gradient onto the dual cone of the rows of the Jacobian. We name this $\mathcal{A}_{\text{DualProj}}$.

Several other works consider the gradients to be noisy when making their theoretical analysis (Liu & Vicente, 2021; Zhou et al., 2022; Fernando et al., 2022; Chen et al., 2024; Xiao et al., 2024). Their solutions for combining gradients are typically stateful. Although this could enhance practical convergence rates, we have restricted our focus to the analysis of stateless aggregators. Exploring and analyzing a generalized Jacobian descent algorithm, that would preserve some state over the iterations, is a promising future direction.

In the federated learning setting, several aggregators have been proposed to combine the model updates while being robust to adversaries (Blanchard et al., 2017; Guerraoui et al., 2018; Chen et al., 2017; Yin et al., 2018). We do not study them here as they mainly focus on security aspects.

Table 1: Properties satisfied for any number of objectives. Proofs are provided in Appendix B.

| Ref. | Aggregator | Non-conflicting | Linear under scaling | Weighted |
|------|------------|-----------------|----------------------|----------|
| — | $\mathcal{A}_{\text{Mean}}$ | ✗ | ✓ | ✓ |
| Désidéri (2012) | $\mathcal{A}_{\text{MGDA}}$ | ✓ | ✗ | ✓ |
| Lopez-Paz & Ranzato (2017) | $\mathcal{A}_{\text{DualProj}}$ | ✓ | ✗ | ✓ |
| Yu et al. (2020) | $\mathcal{A}_{\text{PCGrad}}$ | ✗ | ✓ | ✓ |
| Chen et al. (2020) | $\mathcal{A}_{\text{GradDrop}}$ | ✗ | ✗ | ✗ |
| Liu et al. (2021b) | $\mathcal{A}_{\text{IMTL-G}}$ | ✗ | ✗ | ✓ |
| Liu et al. (2021a) | $\mathcal{A}_{\text{CAGrad}}$ | ✗ | ✗ | ✓ |
| Lin et al. (2021) | $\mathcal{A}_{\text{RGW}}$ | ✗ | ✓ | ✓ |
| Navon et al. (2022) | $\mathcal{A}_{\text{Nash-MTL}}$ | ✓ | ✗ | ✓ |
| Senushkin et al. (2023) | $\mathcal{A}_{\text{Aligned-MTL}}$ | ✗ | ✗ | ✓ |
| (ours) | $\mathcal{A}_{\text{UPGrad}}$ | ✓ | ✓ | ✓ |

## 5 EXPERIMENTS

In the following, we present empirical results for instance-wise risk minimization on some simple image classification datasets. IWRM is performed by stochastic sub-Jacobian descent, as described in Section 3. A key consideration is that when the aggregator is $\mathcal{A}_{\text{Mean}}$, this approach becomes equivalent to empirical risk minimization with SGD. It is thus used as a baseline for comparison.

We train convolutional neural networks on subsets of SVHN (Netzer et al., 2011), CIFAR-10 (Krizhevsky et al., 2009), EuroSAT (Helber et al., 2019), MNIST (LeCun et al., 1998), Fashion-MNIST (Xiao et al., 2017) and Kuzushiji-MNIST (Clanuwat et al., 2018). To make the comparisons as fair as possible, we have tuned the learning rate very precisely for each aggregator, as explained in detail in Appendix C.1. We have also run the same experiments several times independently to gain confidence in our results. Since this leads to a total of 43776 training runs across all of our experiments, we have limited the size of each training dataset to 1024 images, greatly reducing computational costs. Note that this is strictly an optimization problem: we are not studying the generalization of the model, which would be captured by some performance metric on a test set. Other experimental settings, such as the network architectures and the total computational budget used to run our experiments, are given in Appendix C. Figure 2 reports the main results on SVHN and CIFAR-10, two of the datasets exhibiting the most substantial performance gap. Results on the other datasets and aggregators are reported in Appendix D.1. They also demonstrate a significant performance gap.

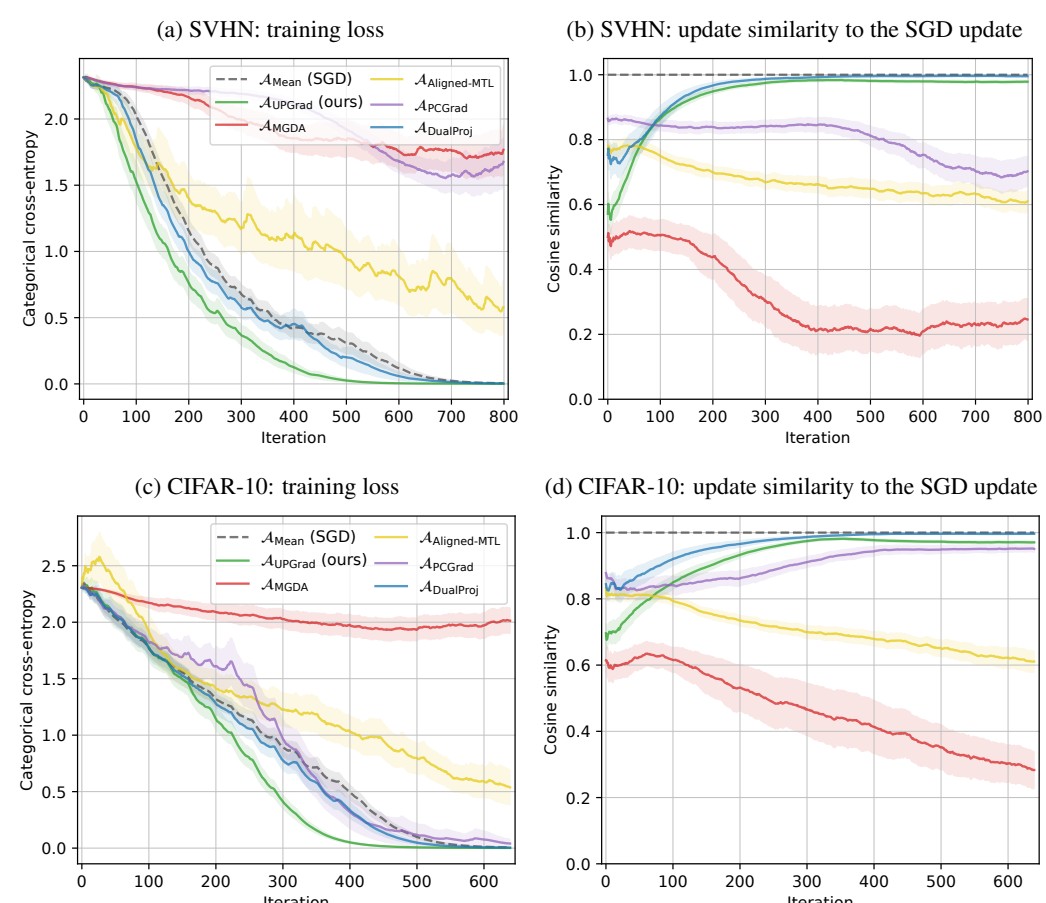

Figure 2: Optimization metrics obtained with IWRM with 1024 training examples and a batch size of 32, averaged over 8 independent runs. The shaded area around each curve shows the estimated standard error of the mean over the 8 runs. Curves are smoothed for readability. Best viewed in color.

Here, we compare the aggregators in terms of their average loss over the training set: the goal of ERM. For this reason, it is rather surprising that $\mathcal{A}_{\text{Mean}}$, which directly optimizes this objective, exhibits a slower convergence rate than some other aggregators. In particular, $\mathcal{A}_{\text{UPGrad}}$, and to a lesser extent $\mathcal{A}_{\text{DualProj}}$, provide improvements on all datasets.

Figures 2b and 2d show the similarity between the update of each aggregator and the update given by $\mathcal{A}_{\text{Mean}}$. For $\mathcal{A}_{\text{UPGrad}}$, a low similarity indicates that there are some conflicting gradients with imbalanced norms (a setting illustrated in Figure 1). Our interpretation is thus that $\mathcal{A}_{\text{UPGrad}}$ prevents gradients of hard examples from being dominated by those of easier examples early into the training. Since fitting those is more complex and time-consuming, it is beneficial to consider them earlier. We believe the similarity increases later on because the gradients become more balanced. This further suggests a greater stability of $\mathcal{A}_{\text{UPGrad}}$ compared to $\mathcal{A}_{\text{Mean}}$, which may allow it to perform effectively at a higher learning rate and, consequently, accelerate its convergence.

The sub-optimal performance of $\mathcal{A}_{\text{MGDA}}$ in this setting can be attributed to its sensitivity to small gradients. If any row of the Jacobian approaches zero, the aggregation by $\mathcal{A}_{\text{MGDA}}$ will also approach zero. This observation illustrates the discrepancy between stationarity and optimality, as discussed in Section 2.4. A notable advantage of linearity under scaling is to explicitly prevent this from happening.

Overall, these experiments demonstrate a high potential for the IWRM paradigm and confirm the relevance of JD, and more specifically of SSJD, as multi-objective optimization algorithms. Besides, the superiority of $\mathcal{A}_{\text{UPGrad}}$ in such a simple setting supports our theoretical results.

While increasing the batch size in SGD reduces variance, the effect of doing so in SSJD combined with $\mathcal{A}_{\text{UPGrad}}$ is non-trivial, as it also tightens the dual cone. Additional results obtained when varying the batch size or updating the parameters with the `Adam` optimizer are available in Appendices D.2 and D.3, respectively.

While an iteration of SSJD is more expensive than an iteration of SGD, its runtime is influenced by several factors, including the choice of aggregator, the parallelization capabilities of the hardware used for Jacobian computation, and the implementation. Appendix E provides memory usage and computation time considerations for our methods. Additionally, we propose a path towards a more efficient implementation in the next section.

## 6 GRAMIAN-BASED JACOBIAN DESCENT

When the number of objectives is dominated by the number of parameters of the model, the main overhead of JD comes from the usage of a Jacobian matrix rather than a single gradient. In the following, we motivate an alternative implementation of JD that only uses the inner products between each pair of gradients.

For any $J \in \mathbb{R}^{m \times n}$, the matrix $G = JJ^\top$ is called the *Gramian* of $J$ and is positive semi-definite. Let $\mathcal{M}_m \subseteq \mathbb{R}^{m \times m}$ be the set of positive semi-definite matrices. The Gramian of the Jacobian, denoted $\mathcal{G}\boldsymbol{f}(\boldsymbol{x}) = \mathcal{J}\boldsymbol{f}(\boldsymbol{x}) \cdot \mathcal{J}\boldsymbol{f}(\boldsymbol{x})^\top \in \mathcal{M}_m$, captures the relations – including conflicts – between all pairs of gradients. Whenever $\mathcal{A}$ is a weighted aggregator, the update of JD is $\boldsymbol{y} = -\eta \mathcal{J}\boldsymbol{f}(\boldsymbol{x})^\top \boldsymbol{w}$ for some vector of weights $\boldsymbol{w} \in \mathbb{R}^m$. Substituting this into the Taylor approximation of (2) gives

$$\boldsymbol{f}(\boldsymbol{x} + \boldsymbol{y}) = \boldsymbol{f}(\boldsymbol{x}) - \eta \mathcal{G}\boldsymbol{f}(\boldsymbol{x}) \cdot \boldsymbol{w} + o\left(\eta\sqrt{\boldsymbol{w}^\top \cdot \mathcal{G}\boldsymbol{f}(\boldsymbol{x}) \cdot \boldsymbol{w}}\right). \tag{9}$$

This expression only depends on the Jacobian through its Gramian. It is thus sensible to focus on aggregators whose weights are only a function of the Gramian. Denoting this function as $\mathcal{W} : \mathcal{M}_m \to \mathbb{R}^m$, those aggregators satisfy $\mathcal{A}(J) = J^\top \cdot \mathcal{W}(G)$. Remarkably, all weighted aggregators of Table 1 can be expressed in this form. In the case of $\mathcal{A}_{\text{UPGrad}}$, this is clearly demonstrated in Proposition 1, which shows that the weights depend on $G$. For such aggregators, substitution and linearity of differentiation[3] then yield

$$\mathcal{A}(\mathcal{J}\boldsymbol{f}(\boldsymbol{x})) = \nabla\left(\mathcal{W}(\mathcal{G}\boldsymbol{f}(\boldsymbol{x}))^\top \cdot \boldsymbol{f}\right)(\boldsymbol{x}). \tag{10}$$

After computing $\mathcal{W}(\mathcal{G}\boldsymbol{f}(\boldsymbol{x}))$, a step of JD would thus only require the backpropagation of a scalar function. The computational cost of applying $\mathcal{W}$ depends on the aggregator and is often dominated by the cost of computing the Gramian.

We now outline a method to compute the Gramian of the Jacobian without ever having to store the full Jacobian in memory. Similarly to the backpropagation algorithm, we can leverage the chain rule. Let $\boldsymbol{g} : \mathbb{R}^n \to \mathbb{R}^k$ and $\boldsymbol{f} : \mathbb{R}^k \to \mathbb{R}^m$, then for any $\boldsymbol{x} \in \mathbb{R}^n$, the chain rule for Gramians is

$$\mathcal{G}(\boldsymbol{f} \circ \boldsymbol{g})(\boldsymbol{x}) = \mathcal{J}\boldsymbol{f}(\boldsymbol{g}(\boldsymbol{x})) \cdot \mathcal{G}\boldsymbol{g}(\boldsymbol{x}) \cdot \mathcal{J}\boldsymbol{f}(\boldsymbol{g}(\boldsymbol{x}))^\top. \tag{11}$$

Moreover, when the function has multiple inputs, the Gramian can be computed as a sum of individual Gramians. Let $\boldsymbol{f} : \mathbb{R}^{n_1 + \cdots + n_k} \to \mathbb{R}^m$ and $\boldsymbol{x} = \begin{bmatrix} \boldsymbol{x}_1^\top & \cdots & \boldsymbol{x}_k^\top \end{bmatrix}^\top$. We can write $\mathcal{J}\boldsymbol{f}(\boldsymbol{x})$ as the concatenation of Jacobians $[\mathcal{J}_{\boldsymbol{x}_1}\boldsymbol{f}(\boldsymbol{x}) \quad \cdots \quad \mathcal{J}_{\boldsymbol{x}_k}\boldsymbol{f}(\boldsymbol{x})]$, where $\mathcal{J}_{\boldsymbol{x}_i}\boldsymbol{f}(\boldsymbol{x})$ is the Jacobian of $\boldsymbol{f}$ with respect to $\boldsymbol{x}_i$ evaluated at $\boldsymbol{x}$. For any $i \in [k]$, let $\mathcal{G}_{\boldsymbol{x}_i}\boldsymbol{f}(\boldsymbol{x}) = \mathcal{J}_{\boldsymbol{x}_i}\boldsymbol{f}(\boldsymbol{x}) \cdot \mathcal{J}_{\boldsymbol{x}_i}\boldsymbol{f}(\boldsymbol{x})^\top$. Then

$$\mathcal{G}\boldsymbol{f}(\boldsymbol{x}_1, \ldots, \boldsymbol{x}_k) = \sum_{i \in [k]} \mathcal{G}_{\boldsymbol{x}_i}\boldsymbol{f}(\boldsymbol{x}_1, \ldots, \boldsymbol{x}_k). \tag{12}$$

When a function is made of compositions and concatenations of elementary functions, the Gramian of the Jacobian can thus be expressed with sums and products of partial Jacobians.

We now provide an example algorithm to compute the Gramian of a sequence of layers. For $0 \le i < k$, let $\boldsymbol{f}_i : \mathbb{R}^{n_i} \times \mathbb{R}^{\ell_i} \to \mathbb{R}^{n_{i+1}}$ be a layer parameterized by $\boldsymbol{p}_i \in \mathbb{R}^{\ell_i}$. Given $\boldsymbol{x}_0 \in \mathbb{R}^{n_0}$, for $0 \le i < k$,

---

[3]For any $\boldsymbol{x} \in \mathbb{R}^n$ and any $\boldsymbol{w} \in \mathbb{R}^m$, $\mathcal{J}\boldsymbol{f}(\boldsymbol{x})^\top \boldsymbol{w} = \nabla\left(\boldsymbol{w}^\top \boldsymbol{f}\right)(\boldsymbol{x})$

the activations are recursively defined as $\boldsymbol{x}_{i+1} = \boldsymbol{f}_i(\boldsymbol{x}_i, \boldsymbol{p}_i)$. Algorithm 3 illustrates how (11) and (12) can be combined to compute the Gramian of the network with respect to its parameters.

---

**Algorithm 3:** *Gramian reverse accumulation* for a sequence of layers

---

$J_x \leftarrow I$   # Identity matrix of size $n_k \times n_k$
$G \leftarrow 0$   # Zero matrix of size $n_k \times n_k$
**for** $i \leftarrow k - 1$ **to** 0 **do**
   $J_p \leftarrow \mathcal{J}_{\boldsymbol{p}_i} \boldsymbol{f}_i(\boldsymbol{x}_i, \boldsymbol{p}_i) \cdot J_x$   # Jacobian of $\boldsymbol{x}_k$ w.r.t. $\boldsymbol{p}_i$
   $J_x \leftarrow \mathcal{J}_{\boldsymbol{x}_i} \boldsymbol{f}_i(\boldsymbol{x}_i, \boldsymbol{p}_i) \cdot J_x$   # Jacobian of $\boldsymbol{x}_k$ w.r.t. $\boldsymbol{x}_i$
   $G \leftarrow G + J_p J_p^\top$
**Output:** $G$

---

Generalizing Algorithm 3 to any computational graph and implementing it efficiently remains an open challenge extending beyond the scope of this work.

## 7 CONCLUSION

In this paper, we introduced Jacobian descent (JD), a multi-objective optimization algorithm defined by some aggregator that maps the Jacobian to an update direction. We identified desirable properties for aggregators and proposed $\mathcal{A}_{\text{UPGrad}}$, addressing the limitations of existing methods while providing stronger convergence guarantees. We also highlighted potential applications of JD and proposed IWRM, a novel learning paradigm considering the loss of each training example as a distinct objective. Given its promising empirical results, we believe this paradigm deserves further attention. Additionally, we see potential for $\mathcal{A}_{\text{UPGrad}}$ beyond JD, as a linear algebra tool for combining conflicting vectors in broader contexts. As speed is the primary limitation of JD, we have outlined an algorithm for efficiently computing the Gramian of the Jacobian, which could unlock JD's full potential. We hope this work serves as a foundation for future research in multi-objective optimization and encourages a broader adoption of these methods.

**Limitations and future directions.** Our experimentation has some limitations. First, we only evaluate JD on IWRM, a setting with moderately conflicting objectives. It would be essential to develop proper benchmarks to compare aggregators on a wide variety of problems. Ideally, such problems should involve substantially conflicting objectives, e.g. multi-task learning with inherently competing or even adversarial tasks. Then, we have limited our scope to the comparison of optimization speeds, disregarding generalization. While this simplifies the experiments and makes the comparison rigorous, optimization and generalization are sometimes intertwined. We thus believe that future works should focus on both aspects.

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

# A PROOFS

## A.1 SUPPLEMENTARY THEORETICAL RESULTS

Recall that a function $\boldsymbol{f} : \mathbb{R}^n \to \mathbb{R}^m$ is $\preceq$-convex if for all $\boldsymbol{x}, \boldsymbol{y} \in \mathbb{R}^n$ and any $\lambda \in [0, 1]$,

$$\boldsymbol{f}\big(\lambda \boldsymbol{x} + (1 - \lambda)\boldsymbol{y}\big) \preceq \lambda \boldsymbol{f}(\boldsymbol{x}) + (1 - \lambda)\boldsymbol{f}(\boldsymbol{y}).$$

**Lemma 1.** If $\boldsymbol{f} : \mathbb{R}^n \to \mathbb{R}^m$ is a continuously differentiable $\preceq$-convex function, then for any pair of vectors $\boldsymbol{x}, \boldsymbol{y} \in \mathbb{R}^n$, $\mathcal{J}\boldsymbol{f}(\boldsymbol{x})(\boldsymbol{y} - \boldsymbol{x}) \preceq \boldsymbol{f}(\boldsymbol{y}) - \boldsymbol{f}(\boldsymbol{x})$.

*Proof.*

$$\begin{aligned}
\mathcal{J}\boldsymbol{f}(\boldsymbol{x})(\boldsymbol{y} - \boldsymbol{x}) &= \lim_{\lambda \to 0^+} \frac{\boldsymbol{f}\big(\boldsymbol{x} + \lambda(\boldsymbol{y} - \boldsymbol{x})\big) - \boldsymbol{f}(\boldsymbol{x})}{\lambda} && \big(\text{differentiation}\big) \\
&\preceq \lim_{\lambda \to 0^+} \frac{\boldsymbol{f}(\boldsymbol{x}) + \lambda\big(\boldsymbol{f}(\boldsymbol{y}) - \boldsymbol{f}(\boldsymbol{x})\big) - \boldsymbol{f}(\boldsymbol{x})}{\lambda} && \big(\preceq\text{-convexity}\big) \\
&= \boldsymbol{f}(\boldsymbol{y}) - \boldsymbol{f}(\boldsymbol{x}),
\end{aligned}$$

which concludes the proof. $\square$

**Lemma 2.** Let $J \in \mathbb{R}^{m \times n}$, let $\boldsymbol{u} \in \mathbb{R}^m$ and let $\boldsymbol{x} \in \mathbb{R}^n$, then

$$\boldsymbol{u}^\top J \boldsymbol{x} \leq \|\boldsymbol{u}\| \cdot \|J\|_{\mathrm{F}} \cdot \|\boldsymbol{x}\|$$

*Proof.* Let $J_i$ be the $i$th row of $J$, then

$$\begin{aligned}
\big(\boldsymbol{u}^\top J \boldsymbol{x}\big)^2 &\leq \|\boldsymbol{u}\|^2 \cdot \|J\boldsymbol{x}\|^2 && \left(\begin{smallmatrix}\text{Cauchy-Schwartz} \\ \text{inequality}\end{smallmatrix}\right) \\
&= \|\boldsymbol{u}\|^2 \cdot \sum_{i \in [m]} \big(J_i^\top \boldsymbol{x}\big)^2 \\
&\leq \|\boldsymbol{u}\|^2 \cdot \sum_{i \in [m]} \|J_i\|^2 \cdot \|\boldsymbol{x}\|^2 && \left(\begin{smallmatrix}\text{Cauchy-Schwartz} \\ \text{inequality}\end{smallmatrix}\right) \\
&= \|\boldsymbol{u}\|^2 \cdot \|J\|_{\mathrm{F}}^2 \cdot \|\boldsymbol{x}\|^2,
\end{aligned}$$

which concludes the proof. $\square$

Recall that a function $\boldsymbol{f} : \mathbb{R}^n \to \mathbb{R}^m$ is $\beta$-smooth if for all $\boldsymbol{x}, \boldsymbol{y} \in \mathbb{R}^n$,

$$\big\|\mathcal{J}\boldsymbol{f}(\boldsymbol{x}) - \mathcal{J}\boldsymbol{f}(\boldsymbol{y})\big\|_{\mathrm{F}} \leq \beta \|\boldsymbol{x} - \boldsymbol{y}\| \tag{13}$$

**Lemma 3.** Let $\boldsymbol{f} : \mathbb{R}^n \to \mathbb{R}^m$ be $\beta$-smooth, then for any $\boldsymbol{w} \in \mathbb{R}^m$ and any $\boldsymbol{x}, \boldsymbol{y} \in \mathbb{R}^n$,

$$\boldsymbol{w}^\top \big(\boldsymbol{f}(\boldsymbol{x}) - \boldsymbol{f}(\boldsymbol{y}) - \mathcal{J}\boldsymbol{f}(\boldsymbol{y})(\boldsymbol{x} - \boldsymbol{y})\big) \leq \frac{\beta}{2} \|\boldsymbol{w}\| \cdot \|\boldsymbol{x} - \boldsymbol{y}\|^2 \tag{14}$$

*Proof.*

$$\begin{aligned}
&\boldsymbol{w}^\top \big(\boldsymbol{f}(\boldsymbol{x}) - \boldsymbol{f}(\boldsymbol{y}) - \mathcal{J}\boldsymbol{f}(\boldsymbol{y})(\boldsymbol{x} - \boldsymbol{y})\big) \\
&= \boldsymbol{w}^\top \left(\int_0^1 \mathcal{J}\boldsymbol{f}\big(\boldsymbol{y} + t(\boldsymbol{x} - \boldsymbol{y})\big)(\boldsymbol{x} - \boldsymbol{y})\, dt - \mathcal{J}\boldsymbol{f}(\boldsymbol{y})(\boldsymbol{x} - \boldsymbol{y})\right) && \left(\begin{smallmatrix}\text{fundamental} \\ \text{theorem} \\ \text{of calculus}\end{smallmatrix}\right) \\
&= \int_0^1 \boldsymbol{w}^\top \big(\mathcal{J}\boldsymbol{f}\big(\boldsymbol{y} + t(\boldsymbol{x} - \boldsymbol{y})\big) - \mathcal{J}\boldsymbol{f}(\boldsymbol{y})\big)(\boldsymbol{x} - \boldsymbol{y})\, dt \\
&\leq \int_0^1 \|\boldsymbol{w}\| \cdot \big\|\mathcal{J}\boldsymbol{f}\big(\boldsymbol{y} + t(\boldsymbol{x} - \boldsymbol{y})\big) - \mathcal{J}\boldsymbol{f}(\boldsymbol{y})\big\|_{\mathrm{F}} \cdot \|\boldsymbol{x} - \boldsymbol{y}\|\, dt && (\text{Lemma 2}) \\
&\leq \int_0^1 \|\boldsymbol{w}\| \cdot \beta t \cdot \|\boldsymbol{x} - \boldsymbol{y}\|^2\, dt && \big(\beta\text{-smoothness 13}\big) \\
&= \frac{\beta}{2} \|\boldsymbol{w}\| \cdot \|\boldsymbol{x} - \boldsymbol{y}\|^2,
\end{aligned}$$

which concludes the proof. $\square$

## A.2 PROPOSITION 1

**Proposition 1.** Let $J \in \mathbb{R}^{m \times n}$. For any $\boldsymbol{u} \in \mathbb{R}^m$, $\pi_J(J^\top \boldsymbol{u}) = J^\top \boldsymbol{w}$ with

$$\boldsymbol{w} \in \underset{\boldsymbol{v} \in \mathbb{R}^m : \, \boldsymbol{u} \preceq \boldsymbol{v}}{\arg\min} \, \boldsymbol{v}^\top J J^\top \boldsymbol{v}. \tag{5}$$

*Proof.* This is a direct consequence of Lemma 4. $\qquad \square$

**Lemma 4.** Let $J \in \mathbb{R}^{m \times n}$, $G = J J^\top$, $\boldsymbol{u} \in \mathbb{R}^m$. For any $\boldsymbol{w} \in \mathbb{R}^m$ satisfying

$$\begin{cases} \boldsymbol{u} \preceq \boldsymbol{w} & \text{(15a)} \\ \boldsymbol{0} \preceq G\boldsymbol{w} & \text{(15b)} \\ \boldsymbol{u}^\top G\boldsymbol{w} = \boldsymbol{w}^\top G\boldsymbol{w} & \text{(15c)} \end{cases}$$

we have $\pi_J(J^\top \boldsymbol{u}) = J^\top \boldsymbol{w}$. Such a $\boldsymbol{w}$ is the solution to

$$\boldsymbol{w} \in \underset{\boldsymbol{u} \preceq \boldsymbol{v}}{\arg\min} \, \boldsymbol{v}^\top G\boldsymbol{v}.$$

*Proof.* The projection

$$\pi_J(J^\top \boldsymbol{u}) = \underset{\substack{\boldsymbol{x} \in \mathbb{R}^n : \\ \boldsymbol{0} \preceq J\boldsymbol{x}}}{\arg\min} \, \frac{1}{2} \|\boldsymbol{x} - J^\top \boldsymbol{u}\|^2$$

is a convex program. Consequently, the KKT conditions are both necessary and sufficient. The Lagragian is given by $\mathcal{L}(\boldsymbol{x}, \boldsymbol{v}) = \frac{1}{2} \|\boldsymbol{x} - J^\top \boldsymbol{u}\|^2 - \boldsymbol{v}^\top J\boldsymbol{x}$. The KKT conditions are then given by

$$\begin{cases} \nabla_{\boldsymbol{x}} \mathcal{L}(\boldsymbol{x}, \boldsymbol{v}) = \boldsymbol{0} \\ \boldsymbol{0} \preceq \boldsymbol{v} \\ \boldsymbol{0} \preceq J\boldsymbol{x} \\ 0 = \boldsymbol{v}^\top J\boldsymbol{x} \end{cases}$$

$$\Leftrightarrow \begin{cases} \boldsymbol{x} = J^\top (\boldsymbol{u} + \boldsymbol{v}) \\ \boldsymbol{0} \preceq \boldsymbol{v} \\ \boldsymbol{0} \preceq G(\boldsymbol{u} + \boldsymbol{v}) \\ 0 = \boldsymbol{v}^\top G(\boldsymbol{u} + \boldsymbol{v}) \end{cases}$$

$$\Leftrightarrow \begin{cases} \boldsymbol{x} = J^\top (\boldsymbol{u} + \boldsymbol{v}) \\ \boldsymbol{u} \preceq \boldsymbol{u} + \boldsymbol{v} \\ \boldsymbol{0} \preceq G(\boldsymbol{u} + \boldsymbol{v}) \\ \boldsymbol{u}^\top G(\boldsymbol{u} + \boldsymbol{v}) = (\boldsymbol{u} + \boldsymbol{v})^\top G(\boldsymbol{u} + \boldsymbol{v}) \end{cases}$$

The simple change of variable $\boldsymbol{w} = \boldsymbol{u} + \boldsymbol{v}$ finishes the proof of the first part.

Since $\boldsymbol{x} = J^\top (\boldsymbol{u} + \boldsymbol{v})$, the Wolfe dual program of $\pi_J(J^\top \boldsymbol{u})$ gives

$$\begin{aligned} \boldsymbol{w} &\in \boldsymbol{u} + \underset{\boldsymbol{v} \in \mathbb{R}^m : \, \boldsymbol{0} \preceq \boldsymbol{v}}{\arg\max} \, \mathcal{L}\big(J^\top (\boldsymbol{u} + \boldsymbol{v}), \boldsymbol{v}\big) \\ &= \boldsymbol{u} + \underset{\boldsymbol{v} \in \mathbb{R}^m : \, \boldsymbol{0} \preceq \boldsymbol{v}}{\arg\max} \, \frac{1}{2} \left\| J^\top \boldsymbol{v} \right\|^2 - \boldsymbol{v}^\top J J^\top (\boldsymbol{u} + \boldsymbol{v}) \\ &= \boldsymbol{u} + \underset{\boldsymbol{v} \in \mathbb{R}^m : \, \boldsymbol{0} \preceq \boldsymbol{v}}{\arg\max} \, -\frac{1}{2} \boldsymbol{v}^\top G\boldsymbol{v} - \boldsymbol{v}^\top G\boldsymbol{u} \\ &= \boldsymbol{u} + \underset{\boldsymbol{v} \in \mathbb{R}^m : \, \boldsymbol{u} \preceq \boldsymbol{u} + \boldsymbol{v}}{\arg\min} \, \frac{1}{2} (\boldsymbol{u} + \boldsymbol{v})^\top G(\boldsymbol{u} + \boldsymbol{v}) \\ &= \underset{\boldsymbol{v}' \in \mathbb{R}^m : \, \boldsymbol{u} \preceq \boldsymbol{v}'}{\arg\min} \, \frac{1}{2} \boldsymbol{v}'^\top G\boldsymbol{v}', \end{aligned}$$

which concludes the proof. $\qquad \square$

## A.3 THEOREM 1

**Theorem 1.** Let $\boldsymbol{f} : \mathbb{R}^n \to \mathbb{R}^m$ be a $\beta$-smooth and $\preceq$-convex function. Suppose that the Pareto front $\boldsymbol{f}(X^*)$ is bounded and that for any $\boldsymbol{x} \in \mathbb{R}^n$, there is $\boldsymbol{x}^* \in X^*$ satisfying $\boldsymbol{f}(\boldsymbol{x}^*) \preceq \boldsymbol{f}(\boldsymbol{x})$. Let $\boldsymbol{x}_1 \in \mathbb{R}^n$, and for all $t \in \mathbb{N}$, $\boldsymbol{x}_{t+1} = \boldsymbol{x}_t - \eta \mathcal{A}_{\text{UPGrad}}\big(\mathcal{J}\boldsymbol{f}(\boldsymbol{x}_t)\big)$, with $\eta = \frac{1}{\beta\sqrt{m}}$. Let $\boldsymbol{w}_t$ be the weights defining $\mathcal{A}_{\text{UPGrad}}\big(\mathcal{J}\boldsymbol{f}(\boldsymbol{x}_t)\big)$ as per (6), i.e. $\mathcal{A}_{\text{UPGrad}}\big(\mathcal{J}\boldsymbol{f}(\boldsymbol{x}_t)\big) = \mathcal{J}\boldsymbol{f}(\boldsymbol{x}_t)^\top \cdot \boldsymbol{w}_t$. If $\boldsymbol{w}_t$ is bounded, then $\boldsymbol{f}(\boldsymbol{x}_t)$ converges to $\boldsymbol{f}(\boldsymbol{x}^*)$ for some $\boldsymbol{x}^* \in X^*$. In other words, $\boldsymbol{f}(\boldsymbol{x}_t)$ converges to the Pareto front.

To prove the theorem we will need Lemmas 5, 6 and 7 below.

**Lemma 5.** Let $J \in \mathbb{R}^{m \times n}$ and $\boldsymbol{w} = \frac{1}{m} \sum_{i=1}^m \boldsymbol{w}_i$ be the weights defining $\mathcal{A}_{\text{UPGrad}}(J)$ as per (6). Let, as usual, $G = JJ^\top$, then,

$$\boldsymbol{w}^\top G \boldsymbol{w} \leq \mathbf{1}^\top G \boldsymbol{w}.$$

*Proof.* Observe that if, for any $\boldsymbol{u}, \boldsymbol{v} \in \mathbb{R}^m$, $\langle \boldsymbol{u}, \boldsymbol{v} \rangle = \boldsymbol{u}^\top G \boldsymbol{v}$, then $\langle \cdot, \cdot \rangle$ is an inner product. In this Hilbert space, the Cauchy-Schwartz inequality reads as

$$\begin{aligned}
(\boldsymbol{u}^\top G \boldsymbol{v})^2 &= \langle \boldsymbol{u}, \boldsymbol{v} \rangle^2 \\
&\leq \langle \boldsymbol{u}, \boldsymbol{u} \rangle \cdot \langle \boldsymbol{v}, \boldsymbol{v} \rangle \\
&= \boldsymbol{u}^\top G \boldsymbol{u} \cdot \boldsymbol{v}^\top G \boldsymbol{v}.
\end{aligned}$$

Therefore

$$\begin{aligned}
& \boldsymbol{w}^\top G \boldsymbol{w} \\
&= \frac{1}{m^2} \sum_{i,j} \boldsymbol{w}_i^\top G \boldsymbol{w}_j \\
&\leq \frac{1}{m^2} \sum_{i,j} \sqrt{\boldsymbol{w}_i^\top G \boldsymbol{w}_i} \cdot \sqrt{\boldsymbol{w}_j^\top G \boldsymbol{w}_j} && \left(\begin{smallmatrix}\text{Cauchy-Schwartz}\\\text{inequality}\end{smallmatrix}\right) \\
&= \left( \sum_i \frac{1}{m} \sqrt{\boldsymbol{w}_i^\top G \boldsymbol{w}_i} \right)^2 \\
&\leq \sum_i \frac{1}{m} \left( \sqrt{\boldsymbol{w}_i^\top G \boldsymbol{w}_i} \right)^2 && (\text{Jensen's inequality}) \\
&= \frac{1}{m} \sum_i \boldsymbol{w}_i^\top G \boldsymbol{w}_i && (G \text{ positive semi-definite}) \\
&= \frac{1}{m} \sum_i \boldsymbol{e}_i^\top G \boldsymbol{w}_i && (\text{Lemma 4, (15c)}) \\
&\leq \frac{1}{m} \sum_i \mathbf{1}^\top G \boldsymbol{w}_i && \left(\begin{smallmatrix}\text{Lemma 4, (15b)}\\\boldsymbol{e}_i \preceq \mathbf{1}\end{smallmatrix}\right) \\
&= \mathbf{1}^\top G \boldsymbol{w},
\end{aligned}$$

which concludes the proof. $\square$

**Lemma 6.** Under the assumptions of Theorem 1, for any $\boldsymbol{w} \in \mathbb{R}^m$ and any $t \in \mathbb{N}$,

$$\boldsymbol{w}^\top \big( \boldsymbol{f}(\boldsymbol{x}_{t+1}) - \boldsymbol{f}(\boldsymbol{x}_t) \big) \leq \frac{\|\boldsymbol{w}\|}{\beta\sqrt{m}} \left( \frac{\mathbf{1}}{2\sqrt{m}} - \frac{\boldsymbol{w}}{\|\boldsymbol{w}\|} \right)^\top G_t \boldsymbol{w}_t.$$

*Proof.* For all $t \in \mathbb{N}$, let $J_t = \mathcal{J}\boldsymbol{f}(\boldsymbol{x}_t), G_t = J_t J_t^\top$. Then $\boldsymbol{x}_{t+1} = \boldsymbol{x}_t - \eta \mathcal{A}_{\text{UPGrad}}(J_t) = \boldsymbol{x}_t - \eta J_t^\top \boldsymbol{w}_t$. Therefore

$$\boldsymbol{w}^\top \big(\boldsymbol{f}(\boldsymbol{x}_{t+1}) - \boldsymbol{f}(\boldsymbol{x}_t)\big)$$

$$\leq -\eta \boldsymbol{w}^\top J_t J_t^\top \boldsymbol{w}_t + \frac{\beta \eta^2}{2} \|\boldsymbol{w}\| \cdot \|J_t^\top \boldsymbol{w}_t\|^2 \qquad (\text{Lemma } 3)$$

$$= -\frac{1}{\beta\sqrt{m}} \boldsymbol{w}^\top G_t \boldsymbol{w}_t + \frac{1}{2\beta m} \|\boldsymbol{w}\| \cdot \boldsymbol{w}_t^\top G_t \boldsymbol{w}_t \qquad \left(\eta = \tfrac{1}{\beta\sqrt{m}}\right)$$

$$\leq -\frac{1}{\beta\sqrt{m}} \boldsymbol{w}^\top G_t \boldsymbol{w}_t + \frac{1}{2\beta m} \|\boldsymbol{w}\| \cdot \boldsymbol{1}^\top G_t \boldsymbol{w}_t \qquad (\text{Lemma } 5)$$

$$= \frac{\|\boldsymbol{w}\|}{\beta\sqrt{m}} \left(\frac{\boldsymbol{1}}{2\sqrt{m}} - \frac{\boldsymbol{w}}{\|\boldsymbol{w}\|}\right)^\top G_t \boldsymbol{w}_t,$$

which concludes the proof. $\qquad\square$

**Lemma 7.** Under the assumptions of Theorem 1, if $\boldsymbol{x}^* \in X^*$ satisfies $\boldsymbol{1}^\top \boldsymbol{f}(\boldsymbol{x}^*) \leq \boldsymbol{1}^\top \boldsymbol{f}(\boldsymbol{x}_t)$ for all $t \in \mathbb{N}$, then

$$\frac{1}{T} \sum_{t \in [T]} \boldsymbol{w}_t^\top \big(\boldsymbol{f}(\boldsymbol{x}_t) - \boldsymbol{f}(\boldsymbol{x}^*)\big) \leq \frac{1}{T}\left(\boldsymbol{1}^\top \big(\boldsymbol{f}(\boldsymbol{x}_1) - \boldsymbol{f}(\boldsymbol{x}^*)\big) + \frac{\beta\sqrt{m}}{2} \|\boldsymbol{x}_1 - \boldsymbol{x}^*\|^2\right). \qquad (16)$$

*Proof.* We first bound, for any $t \in \mathbb{N}$, $\boldsymbol{1}^\top \big(\boldsymbol{f}(\boldsymbol{x}_{t+1}) - \boldsymbol{f}(\boldsymbol{x}_t)\big)$ as follows

$$\boldsymbol{1}^\top \big(\boldsymbol{f}(\boldsymbol{x}_{t+1}) - \boldsymbol{f}(\boldsymbol{x}_t)\big)$$

$$\leq -\frac{1}{2\beta\sqrt{m}} \cdot \boldsymbol{1}^\top G_t \boldsymbol{w}_t \qquad \left(\begin{smallmatrix}\text{Lemma } 6 \\ \text{with } \boldsymbol{w}=\boldsymbol{1}\end{smallmatrix}\right)$$

$$\leq -\frac{1}{2\beta\sqrt{m}} \cdot \boldsymbol{w}_t^\top G_t \boldsymbol{w}_t. \qquad (\text{Lemma } 5)$$

Summing this over $t \in [T]$ yields

$$\frac{1}{2\beta\sqrt{m}} \sum_{t \in [T]} \boldsymbol{w}_t^\top G_t \boldsymbol{w}_t$$

$$\leq \sum_{t \in [T]} \boldsymbol{1}^\top \big(\boldsymbol{f}(\boldsymbol{x}_t) - \boldsymbol{f}(\boldsymbol{x}_{t+1})\big)$$

$$= \boldsymbol{1}^\top \big(\boldsymbol{f}(\boldsymbol{x}_1) - \boldsymbol{f}(\boldsymbol{x}_{T+1})\big) \qquad (\text{Telescoping sum})$$

$$\leq \boldsymbol{1}^\top \big(\boldsymbol{f}(\boldsymbol{x}_1) - \boldsymbol{f}(\boldsymbol{x}^*)\big). \qquad \left(\begin{smallmatrix}\text{Assumption} \\ \boldsymbol{1}^\top \boldsymbol{f}(\boldsymbol{x}^*) \leq \boldsymbol{1}^\top \boldsymbol{f}(\boldsymbol{x}_{T+1})\end{smallmatrix}\right) \qquad (17)$$

Since $\boldsymbol{0} \leq \boldsymbol{w}_t$,

$$\boldsymbol{w}_t^\top \big(\boldsymbol{f}(\boldsymbol{x}_t) - \boldsymbol{f}(\boldsymbol{x}^*)\big)$$

$$\leq \boldsymbol{w}_t^\top J_t(\boldsymbol{x}_t - \boldsymbol{x}^*) \qquad (\text{Lemma } 1)$$

$$= \frac{1}{\eta}(\boldsymbol{x}_t - \boldsymbol{x}_{t+1})^\top (\boldsymbol{x}_t - \boldsymbol{x}^*) \qquad \left(\boldsymbol{x}_{t+1} = \boldsymbol{x}_t - \eta J_t^\top \boldsymbol{w}_t\right)$$

$$= \frac{1}{2\eta}\left(\|\boldsymbol{x}_t - \boldsymbol{x}_{t+1}\|^2 + \|\boldsymbol{x}_t - \boldsymbol{x}^*\|^2 - \|\boldsymbol{x}_{t+1} - \boldsymbol{x}^*\|^2\right) \qquad \left(\begin{smallmatrix}\text{Parallelogram} \\ \text{law}\end{smallmatrix}\right)$$

$$= \frac{1}{2\beta\sqrt{m}} \boldsymbol{w}_t^\top G_t \boldsymbol{w}_t + \frac{\beta\sqrt{m}}{2}\left(\|\boldsymbol{x}_t - \boldsymbol{x}^*\|^2 - \|\boldsymbol{x}_{t+1} - \boldsymbol{x}^*\|^2\right). \qquad \left(\eta = \tfrac{1}{\beta\sqrt{m}}\right)$$

Summing this over $t \in [T]$ yields

$$\sum_{t \in [T]} \boldsymbol{w}_t^\top \big( \boldsymbol{f}(\boldsymbol{x}_t) - \boldsymbol{f}(\boldsymbol{x}^*) \big)$$

$$\leq \frac{1}{2\beta\sqrt{m}} \sum_{t \in [T]} \boldsymbol{w}_t^\top G_t \boldsymbol{w}_t + \frac{\beta\sqrt{m}}{2} \Big( \|\boldsymbol{x}_1 - \boldsymbol{x}^*\|^2 - \|\boldsymbol{x}_{T+1} - \boldsymbol{x}^*\|^2 \Big) \qquad \text{(Telescoping sum)}$$

$$\leq \frac{1}{2\beta\sqrt{m}} \sum_{t \in [T]} \boldsymbol{w}_t^\top G_t \boldsymbol{w}_t + \frac{\beta\sqrt{m}}{2} \|\boldsymbol{x}_1 - \boldsymbol{x}^*\|^2$$

$$\leq \boldsymbol{1}^\top \big( \boldsymbol{f}(\boldsymbol{x}_1) - \boldsymbol{f}(\boldsymbol{x}^*) \big) + \frac{\beta\sqrt{m}}{2} \|\boldsymbol{x}_1 - \boldsymbol{x}^*\|^2. \qquad \text{(By (17))}$$

Scaling down this inequality by $T$ yields

$$\frac{1}{T} \sum_{t \in [T]} \boldsymbol{w}_t^\top \big( \boldsymbol{f}(\boldsymbol{x}_t) - \boldsymbol{f}(\boldsymbol{x}^*) \big) \leq \frac{1}{T} \bigg( \boldsymbol{1}^\top (\boldsymbol{f}(\boldsymbol{x}_1) - \boldsymbol{f}(\boldsymbol{x}^*)) + \frac{\beta\sqrt{m}}{2} \|\boldsymbol{x}_1 - \boldsymbol{x}^*\|^2 \bigg),$$

which concludes the proof. $\qquad \square$

We are now ready to prove Theorem 1.

*Proof.* For all $t \in \mathbb{N}$, let $J_t = \mathcal{J}\boldsymbol{f}(\boldsymbol{x}_t)$, $G_t = J_t J_t^\top$. Then

$$\boldsymbol{x}_{t+1} = \boldsymbol{x}_t - \eta \mathcal{A}_{\text{UPGrad}}(J_t)$$
$$= \boldsymbol{x}_t - \eta J_t^\top \boldsymbol{w}_t.$$

Substituting $\boldsymbol{w} = \boldsymbol{1}$ in the term $\frac{\boldsymbol{1}}{2\sqrt{m}} - \frac{\boldsymbol{w}}{\|\boldsymbol{w}\|}$ of Lemma 6 yields

$$\frac{\boldsymbol{1}}{2\sqrt{m}} - \frac{\boldsymbol{w}}{\|\boldsymbol{w}\|} = -\frac{\boldsymbol{1}}{2\sqrt{m}}$$
$$< \boldsymbol{0}.$$

Therefore there exists some $\varepsilon > 0$ such that any $\boldsymbol{w} \in \mathbb{R}^m$ with $\|\boldsymbol{1} - \boldsymbol{w}\| < \varepsilon$ satisfies $\frac{\boldsymbol{1}}{2\sqrt{m}} < \frac{\boldsymbol{w}}{\|\boldsymbol{w}\|}$. Denote by $B_\varepsilon(\boldsymbol{1}) = \{ \boldsymbol{w} \in \mathbb{R}^m : \|\boldsymbol{1} - \boldsymbol{w}\| < \varepsilon \}$, i.e. for all $\boldsymbol{w} \in B_\varepsilon(\boldsymbol{1})$, $\frac{\boldsymbol{1}}{2\sqrt{m}} < \frac{\boldsymbol{w}}{\|\boldsymbol{w}\|}$. By the non-conflicting property of $\mathcal{A}_{\text{UPGrad}}$, $\boldsymbol{0} \leq G_t \boldsymbol{w}_t$ and therefore for all $\boldsymbol{w} \in B_\varepsilon(\boldsymbol{1})$,

$$\boldsymbol{w}^\top \big( \boldsymbol{f}(\boldsymbol{x}_{t+1}) - \boldsymbol{f}(\boldsymbol{x}_t) \big)$$

$$\leq \frac{\|\boldsymbol{w}\|}{\beta\sqrt{m}} \bigg( \frac{\boldsymbol{1}}{2\sqrt{m}} - \frac{\boldsymbol{w}}{\|\boldsymbol{w}\|} \bigg) G_t \boldsymbol{w}_t \qquad \text{(Lemma 6)}$$

$$\leq 0.$$

Since $\boldsymbol{w}^\top \boldsymbol{f}(\boldsymbol{x}_t)$ is bounded and non-increasing, it converges. Since $B_\varepsilon(\boldsymbol{1})$ contains a basis of $\mathbb{R}^m$, $\boldsymbol{f}(\boldsymbol{x}_t)$ converges to some $\boldsymbol{f}^* \in \mathbb{R}^m$. By assumption on $\boldsymbol{f}$, there exists $\boldsymbol{x}^*$ in the Pareto set satisfying $\boldsymbol{f}(\boldsymbol{x}^*) \preceq \boldsymbol{f}^*$.

We now prove that $\boldsymbol{f}(\boldsymbol{x}^*) = \boldsymbol{f}^*$. Since $\boldsymbol{f}(\boldsymbol{x}^*) \preceq \boldsymbol{f}^*$, it is sufficient to show that $\boldsymbol{1}^\top \big( \boldsymbol{f}^* - \boldsymbol{f}(\boldsymbol{x}^*) \big) \leq 0$.

First, the additional assumption of Lemma 7 applies since $\mathbf{1}^\top \boldsymbol{f}(\boldsymbol{x}_t)$ decreases to $\mathbf{1}^\top \boldsymbol{f}^*$ which is larger than $\mathbf{1}^\top \boldsymbol{f}(\boldsymbol{x}^*)$. Therefore

$$
\begin{aligned}
&\mathbf{1}^\top \left(\boldsymbol{f}^* - \boldsymbol{f}(\boldsymbol{x}^*)\right) \\
&\leq \left(\frac{m}{T} \sum_{t\in[T]} \boldsymbol{w}_t\right)^\top \left(\boldsymbol{f}^* - \boldsymbol{f}(\boldsymbol{x}^*)\right) && \left(\begin{array}{c} \boldsymbol{f}(\boldsymbol{x}^*) \preceq \boldsymbol{f}^* \\ \mathbf{1} \preceq m\boldsymbol{w}_t \\ \text{by (15a)} \end{array}\right) \\
&= \frac{m}{T} \sum_{t\in[T]} \boldsymbol{w}_t^\top \left(\boldsymbol{f}^* - \boldsymbol{f}(\boldsymbol{x}_t) + \boldsymbol{f}(\boldsymbol{x}_t) - \boldsymbol{f}(\boldsymbol{x}^*)\right) \\
&= \frac{m}{T} \left(\sum_{t\in[T]} \boldsymbol{w}_t^\top \left(\boldsymbol{f}^* - \boldsymbol{f}(\boldsymbol{x}_t)\right) + \sum_{t\in[T]} \boldsymbol{w}_t^\top \left(\boldsymbol{f}(\boldsymbol{x}_t) - \boldsymbol{f}(\boldsymbol{x}^*)\right)\right) \\
&\leq \frac{m}{T} \left(\sum_{t\in[T]} \boldsymbol{w}_t^\top \left(\boldsymbol{f}^* - \boldsymbol{f}(\boldsymbol{x}_t)\right) + \mathbf{1}^\top \left(\boldsymbol{f}(\boldsymbol{x}_1) - \boldsymbol{f}(\boldsymbol{x}^*)\right) + \frac{\beta\sqrt{m}}{2}\|\boldsymbol{x}_1 - \boldsymbol{x}^*\|^2\right) && (\text{Lemma 7})
\end{aligned}
$$

$$(18)$$

Taking the limit as $T \to \infty$, we get

$$
\begin{aligned}
&\mathbf{1}^\top \left(\boldsymbol{f}^* - \boldsymbol{f}(\boldsymbol{x}^*)\right) \\
&\leq \lim_{T\to\infty} \frac{m}{T} \sum_{t\in[T]} \boldsymbol{w}_t^\top \left(\boldsymbol{f}^* - \boldsymbol{f}(\boldsymbol{x}_t)\right) \\
&\leq \lim_{T\to\infty} \frac{m}{T} \sum_{t\in[T]} \|\boldsymbol{w}_t\| \cdot \|\boldsymbol{f}^* - \boldsymbol{f}(\boldsymbol{x}_t)\| && \left(\begin{array}{c} \text{Cauchy-Schwartz} \\ \text{inequality} \end{array}\right) \\
&= 0, && \left(\begin{array}{c} \boldsymbol{w}_t \text{ bounded} \\ \boldsymbol{f}(\boldsymbol{x}_t) \to \boldsymbol{f}^* \end{array}\right)
\end{aligned}
$$

which concludes the proof. $\qquad\square$

The proof of Theorem 1 provides some additional insights about the convergence rate as well as some notion of convergence in the non-convex case.

**Convergence rate.** Combining the equality $\boldsymbol{f}^* = \boldsymbol{f}(\boldsymbol{x}^*)$ and (18), we have

$$
\frac{1}{T} \sum_{t\in[T]} \boldsymbol{w}_t^\top \left(\boldsymbol{f}(\boldsymbol{x}_t) - \boldsymbol{f}(\boldsymbol{x}^*)\right) \leq \frac{1}{T}\mathbf{1}^\top \left(\boldsymbol{f}(\boldsymbol{x}_1) - \boldsymbol{f}(\boldsymbol{x}^*)\right) + \frac{\beta\sqrt{m}}{2T}\|\boldsymbol{x}_1 - \boldsymbol{x}^*\|^2
$$

Furthermore, the Cauchy-Schwartz inequality yields $\mathbf{1}^\top \left(\boldsymbol{f}(\boldsymbol{x}_1) - \boldsymbol{f}(\boldsymbol{x}^*)\right) \leq \sqrt{m} \cdot \|\boldsymbol{f}(\boldsymbol{x}_1) - \boldsymbol{f}(\boldsymbol{x}^*)\|$, so

$$
\frac{1}{T} \sum_{t\in[T]} \boldsymbol{w}_t^\top \left(\boldsymbol{f}(\boldsymbol{x}_t) - \boldsymbol{f}(\boldsymbol{x}^*)\right) \leq \frac{\sqrt{m}}{T} \left(\|\boldsymbol{f}(\boldsymbol{x}_1) - \boldsymbol{f}(\boldsymbol{x}^*)\| + \frac{\beta}{2}\|\boldsymbol{x}_1 - \boldsymbol{x}^*\|^2\right) \qquad (19)
$$

This hints a convergence rate of order $\mathcal{O}\left(\frac{1}{T}\right)$, whose constant depends on the initial point $\boldsymbol{x}_1$.

**Non-convex setting.** The proof of (17) does not require the convexity of the objective function. This bound can be equivalently formulated as

$$
\frac{1}{T} \sum_{t=1}^{T} \left\|J_t^\top w_t\right\|^2 \leq \frac{2\beta\sqrt{m}}{T}\mathbf{1}^\top \left(f(x_1) - f(x^*)\right) \qquad (20)
$$

This shows the convergence of the updates in the non-convex setting, under the smoothness condition of Theorem 1. Note that this does not prove the convergence of $x_t$, which is in line with the current limitations of gradient descent in the single-objective setting.

# B PROPERTIES OF EXISTING AGGREGATORS

In the following, we prove the properties of the aggregators from Table 1. Some aggregators, e.g. $\mathcal{A}_{\text{RGW}}$, $\mathcal{A}_{\text{GradDrop}}$ and $\mathcal{A}_{\text{PCGrad}}$, are non-deterministic and are thus not technically functions but rather random variables whose distribution depends on the matrix $J \in \mathbb{R}^{m \times n}$ to aggregate. Still, the properties of Section 2.2 can be easily adapted to a random setting. If $\mathcal{A}$ is a random aggregator, then for any $J$, $\mathcal{A}(J)$ is a random vector in $\mathbb{R}^n$. The aggregator is non-conflicting if $\mathcal{A}(J)$ is in the dual cone of the rows of $J$ with probability 1. It is linear under scaling if for all $J \in \mathbb{R}^{m \times n}$, there is a – possibly random – matrix $\mathfrak{J} \in \mathbb{R}^{m \times n}$ such that for all $\mathbf{0} \prec \boldsymbol{c} \in \mathbb{R}^m$, $\mathcal{A}(\text{diag}(\boldsymbol{c}) \cdot J) = \mathfrak{J}^\top \cdot \boldsymbol{c}$. Finally, $\mathcal{A}$ is weighted if for any $J \in \mathbb{R}^{m \times n}$ there is a – possibly random – weighting $\boldsymbol{w} \in \mathbb{R}^m$ satisfying $\mathcal{A}(J) = J^\top \cdot \boldsymbol{w}$.

## B.1 MEAN

$\mathcal{A}_{\text{Mean}}$ simply averages the rows of the input matrix, i.e. for all $J \in \mathbb{R}^{m \times n}$,

$$\mathcal{A}_{\text{Mean}}(J) = \frac{1}{m} J^\top \cdot \mathbf{1} \tag{21}$$

✗ **Non-conflicting.** $\mathcal{A}_{\text{Mean}}\left(\begin{bmatrix} -2 \\ 4 \end{bmatrix}\right) = [1]$, which conflicts with $[-2]$, so $\mathcal{A}_{\text{Mean}}$ is not non-conflicting.

✓ **Linear under scaling.** For any $\boldsymbol{c} \in \mathbb{R}^m$, $\mathcal{A}_{\text{Mean}}(\text{diag}(\boldsymbol{c}) \cdot J) = \frac{1}{m} J^\top \cdot \boldsymbol{c}$, which is linear in $\boldsymbol{c}$. $\mathcal{A}_{\text{Mean}}$ is therefore linear under scaling.

✓ **Weighted.** By (21), $\mathcal{A}_{\text{Mean}}$ is weighted with constant weighting equal to $\frac{1}{m}\mathbf{1}$.

## B.2 MGDA

The optimization algorithm presented in Désidéri (2012), called MGDA, is tied to a particular method for aggregating the gradients. We thus refer to this aggregator as $\mathcal{A}_{\text{MGDA}}$. The dual problem of this method was also introduced independently in Fliege & Svaiter (2000). We show the equivalence between the two solutions to make the analysis of $\mathcal{A}_{\text{MGDA}}$ easier.

For all $J \in \mathbb{R}^{m \times n}$, the aggregation described in Désidéri (2012) is defined as

$$\mathcal{A}_{\text{MGDA}}(J) = J^\top \cdot \boldsymbol{w} \tag{22}$$

$$\text{with} \quad \boldsymbol{w} \in \underset{\substack{\mathbf{0} \leq \boldsymbol{v}: \\ \mathbf{1}^\top \boldsymbol{v} = 1}}{\arg\min} \left\| J^\top \boldsymbol{v} \right\|^2 \tag{23}$$

In Equation (3) of Fliege & Svaiter (2000), the following problem is studied:

$$\min_{\substack{\alpha \in \mathbb{R}, \boldsymbol{x} \in \mathbb{R}^n: \\ J\boldsymbol{x} \leq \alpha \mathbf{1}}} \alpha + \frac{1}{2}\|\boldsymbol{x}\|^2 \tag{24}$$

We show that the problems in (23) and (24) are dual to each other. Furthermore, the duality gap is null since this is a convex problem. The Lagrangian of the problem in (24) is given by $\mathcal{L}(\alpha, \boldsymbol{x}, \boldsymbol{\mu}) = \alpha + \frac{1}{2}\|\boldsymbol{x}\|^2 - \boldsymbol{\mu}^\top(\alpha \mathbf{1} - J\boldsymbol{x})$. Differentiating w.r.t. $\alpha$ and $\boldsymbol{x}$ gives respectively $1 - \mathbf{1}^\top \boldsymbol{\mu}$ and $\boldsymbol{x} + J^\top \boldsymbol{\mu}$. The dual problem is obtained by setting those two to $\mathbf{0}$ and then maximizing the

Lagrangian on $\mathbf{0} \leq \boldsymbol{\mu}$ and $\alpha$, i.e.

$$\underset{\substack{\alpha, \mathbf{0} \leq \boldsymbol{\mu}: \\ \mathbf{1}^\top \boldsymbol{\mu} = 1}}{\arg\max} \alpha + \frac{1}{2} \left\| J^\top \boldsymbol{\mu} \right\|^2 - \boldsymbol{\mu}^\top \left( \alpha \mathbf{1} + J J^\top \boldsymbol{\mu} \right)$$

$$= \underset{\substack{\alpha, \mathbf{0} \leq \boldsymbol{\mu}: \\ \mathbf{1}^\top \boldsymbol{\mu} = 1}}{\arg\max} \alpha + \frac{1}{2} \left\| J^\top \boldsymbol{\mu} \right\|^2 - \alpha \boldsymbol{\mu}^\top \mathbf{1} - \boldsymbol{\mu}^\top J J^\top \boldsymbol{\mu}$$

$$= \underset{\substack{\mathbf{0} \leq \boldsymbol{\mu}: \\ \mathbf{1}^\top \boldsymbol{\mu} = 1}}{\arg\min} \frac{1}{2} \left\| J^\top \boldsymbol{\mu} \right\|^2$$

Therefore, (23) and (24) are equivalent, with $\boldsymbol{x} = -J^\top \boldsymbol{w}$.

✓ **Non-conflicting.** Observe that since in (24), $\alpha = 0$ and $\boldsymbol{x} = \mathbf{0}$ is feasible, the objective is non-positive and therefore $\alpha \leq 0$. Substituting $\boldsymbol{x} = -J^\top \boldsymbol{w}$ in $J \cdot \boldsymbol{x} \leq \alpha \mathbf{1} \leq \mathbf{0}$ yields $\mathbf{0} \leq J J^\top \boldsymbol{w}$, i.e. $\mathbf{0} \leq J \cdot \mathcal{A}_{\mathrm{MGDA}}(J)$, so $\mathcal{A}_{\mathrm{MGDA}}$ is non-conflicting.

✗ **Linear under scaling.** With $J = \begin{bmatrix} 2 & 0 \\ 0 & 2 \\ a & a \end{bmatrix}$, if $0 \leq a \leq 1$, $\mathcal{A}_{\mathrm{MGDA}}(J) = \begin{bmatrix} a \\ a \end{bmatrix}$. However, if $a \geq 1$, $\mathcal{A}_{\mathrm{MGDA}}(J) = \begin{bmatrix} 1 \\ 1 \end{bmatrix}$. This is not affine in $a$, so $\mathcal{A}_{\mathrm{MGDA}}$ is not linear under scaling. In particular, if any row of $J$ is $\mathbf{0}$, $\mathcal{A}_{\mathrm{MGDA}}(J) = \mathbf{0}$. This implies that the optimization will stop whenever one objective has converged.

✓ **Weighted.** By (22), $\mathcal{A}_{\mathrm{MGDA}}$ is weighted.

### B.3 DUALPROJ

The projection of a gradient of interest onto a dual cone was first described in Lopez-Paz & Ranzato (2017). When this gradient is the average of the rows of the Jacobian, we call this aggregator $\mathcal{A}_{\mathrm{DualProj}}$. Formally,

$$\mathcal{A}_{\mathrm{DualProj}}(J) = \frac{1}{m} \cdot \pi_J \left( J^\top \cdot \mathbf{1} \right) \tag{25}$$

where $\pi_J$ is the projection operator defined in (3).

✓ **Non-conflicting.** By the constraint in (3), $\mathcal{A}_{\mathrm{DualProj}}$ is non-conflicting.

✗ **Linear under scaling.** With $J = \begin{bmatrix} 2 & 0 \\ -2a & 2a \end{bmatrix}$, if $a \geq 1$, $\mathcal{A}_{\mathrm{DualProj}}(J) = \begin{bmatrix} 0 \\ a \end{bmatrix}$. However, if $0.5 \leq a \leq 1$, $\mathcal{A}_{\mathrm{DualProj}}(J) = \begin{bmatrix} 1 - a \\ a \end{bmatrix}$. This is not affine in $a$, so $\mathcal{A}_{\mathrm{DualProj}}$ is not linear under scaling.

✓ **Weighted.** By Proposition 1, $\mathcal{A}_{\mathrm{DualProj}}(J) = \frac{1}{m} J^\top \cdot \boldsymbol{w}$, with $\boldsymbol{w} \in \arg\min_{\mathbf{1} \leq \boldsymbol{v}} \boldsymbol{v}^\top J J^\top \boldsymbol{v}$. $\mathcal{A}_{\mathrm{DualProj}}$ is thus weighted.

### B.4 PCGRAD

$\mathcal{A}_{\mathrm{PCGrad}}$ is described in Yu et al. (2020). It projects each gradient onto the orthogonal hyperplane of other gradients in case of conflict with them, iteratively and in random order. When $m \leq 2$, $\mathcal{A}_{\mathrm{PCGrad}}$ is deterministic and satisfies $\mathcal{A}_{\mathrm{PCGrad}} = m \cdot \mathcal{A}_{\mathrm{UPGrad}}$. Therefore, in this case, it satisfies all three properties. When $m > 2$, $\mathcal{A}_{\mathrm{PCGrad}}$ is non-deterministic, so $\mathcal{A}_{\mathrm{PCGrad}}(J)$ is a random vector.

For any index $i \in [m]$, let $\boldsymbol{g}_i = J^\top \cdot \boldsymbol{e}_i$ and let $\mathbf{p}(i)$ be a random vector distributed uniformly on the set of permutations of the elements in $[m] \setminus \{i\}$. For instance, if $m = 3$, $\mathbf{p}(2) = [1 \quad 3]^\top$ with probability 0.5 and $\mathbf{p}(2) = [3 \quad 1]^\top$ with probability 0.5. For notation convenience, whenever $i$

is clear from context, we denote $j_k = \mathbf{p}(i)_k$. The iterative projection of $\mathcal{A}_{\text{PCGrad}}$ is then defined recursively as:

$$\boldsymbol{g}_{i,1}^{\text{PC}} = \boldsymbol{g}_i \tag{26}$$

$$\boldsymbol{g}_{i,k+1}^{\text{PC}} = \boldsymbol{g}_{i,k}^{\text{PC}} - \mathbb{1}\{\boldsymbol{g}_{i,k}^{\text{PC}} \cdot \boldsymbol{g}_{j_k} < 0\} \frac{\boldsymbol{g}_{i,k}^{\text{PC}} \cdot \boldsymbol{g}_{j_k}}{\|\boldsymbol{g}_{j_k}\|^2} \boldsymbol{g}_{j_k} \tag{27}$$

We noticed that an equivalent formulation to the conditional projection of (27) is the projection onto the dual cone of $\{\boldsymbol{g}_{j_k}\}$:

$$\boldsymbol{g}_{i,k+1}^{\text{PC}} = \pi_{\boldsymbol{g}_{j_k}^{\top}}(\boldsymbol{g}_{i,k}^{\text{PC}}) \tag{28}$$

Finally, the aggregation is given by

$$\mathcal{A}_{\text{PCGrad}}(J) = \sum_{i=1}^{m} \boldsymbol{g}_{i,m}^{\text{PC}}. \tag{29}$$

✗ **Non-conflicting.** If $J = \begin{bmatrix} 1 & 0 \\ 0 & 1 \\ -0.5 & -1 \end{bmatrix}$, the only non-conflicting direction is $\mathbf{0}$. However, $\mathcal{A}_{\text{PCGrad}}(J)$ is uniform over the set $\left\{ \begin{bmatrix} 0.4 \\ 0.2 \end{bmatrix}, \begin{bmatrix} 0.8 \\ 0.2 \end{bmatrix}, \begin{bmatrix} 0.4 \\ -0.2 \end{bmatrix}, \begin{bmatrix} 0.8 \\ -0.2 \end{bmatrix} \right\}$, i.e. $\mathcal{A}_{\text{PCGrad}}(J)$ is in the dual cone of the rows of $J$ with probability $0$. $\mathcal{A}_{\text{PCGrad}}$ is thus not non-conflicting. Here, $\mathbb{E}[\mathcal{A}_{\text{PCGrad}}(J)] = [0.6 \quad 0]^{\top}$, so $\mathcal{A}_{\text{PCGrad}}$ is neither non-conflicting in expectation.

✓ **Linear under scaling.** To show that $\mathcal{A}_{\text{PCGrad}}$ is linear under scaling, let $\mathbf{0} \prec \boldsymbol{c} \in \mathbb{R}^m$, $\boldsymbol{g}_i' = c_i \boldsymbol{g}_i$, $\boldsymbol{g}_{i,1}'^{\text{PC}} = \boldsymbol{g}_i'$ and $\boldsymbol{g}_{i,k+1}'^{\text{PC}} = \pi_{\boldsymbol{g}_{j_k}'^{\top}}(\boldsymbol{g}_{i,k}'^{\text{PC}})$. We show by induction that $\boldsymbol{g}_{i,k}'^{\text{PC}} = c_i \boldsymbol{g}_{i,k}^{\text{PC}}$.

The base case is given by $\boldsymbol{g}_{i,1}'^{\text{PC}} = \boldsymbol{g}_i' = c_i \boldsymbol{g}_i = c_i \boldsymbol{g}_{i,1}^{\text{PC}}$.

Then, assuming the induction hypothesis $\boldsymbol{g}_{i,k}'^{\text{PC}} = c_i \boldsymbol{g}_{i,k}^{\text{PC}}$, we show $\boldsymbol{g}_{i,k+1}'^{\text{PC}} = c_i \boldsymbol{g}_{i,k+1}^{\text{PC}}$:

$$\boldsymbol{g}_{i,k+1}'^{\text{PC}} = \pi_{c_{j_k} \boldsymbol{g}_{j_k}^{\top}}(c_i \boldsymbol{g}_{i,k}^{\text{PC}}) \qquad \left(\text{Induction hypothesis}\right)$$

$$\boldsymbol{g}_{i,k+1}'^{\text{PC}} = c_i \pi_{\boldsymbol{g}_{j_k}^{\top}}(\boldsymbol{g}_{i,k}^{\text{PC}}) \qquad \left(0 < c_i \text{ and } 0 < c_{j_k}\right)$$

$$\boldsymbol{g}_{i,k+1}'^{\text{PC}} = c_i \boldsymbol{g}_{i,k+1}^{\text{PC}} \qquad \left(\text{By (28)}\right)$$

Therefore $\mathcal{A}_{\text{PCGrad}}\big(\text{diag}(\boldsymbol{c}) \cdot J\big) = \sum_{i=1}^{m} c_i \boldsymbol{g}_{i,m}^{\text{PC}}$, so it can be written as $\mathcal{A}_{\text{PCGrad}}\big(\text{diag}(\boldsymbol{c}) \cdot J\big) = \mathfrak{J}^{\top} \cdot \boldsymbol{c}$ with $\mathfrak{J} = \begin{bmatrix} \boldsymbol{g}_{1,m}^{\text{PC}} & \cdots & \boldsymbol{g}_{m,m}^{\text{PC}} \end{bmatrix}^{\top}$. Therefore, $\mathcal{A}_{\text{PCGrad}}$ is linear under scaling.

✓ **Weighted.** For all $i$, $\boldsymbol{g}_{i,m}^{\text{PC}}$ is always a random linear combination of rows of $J$. $\mathcal{A}_{\text{PCGrad}}$ is thus weighted.

## B.5 GRADDROP

The aggregator used by the GradDrop layer, which we denote $\mathcal{A}_{\text{GradDrop}}$, is described in Chen et al. (2020). It is non-deterministic, so $\mathcal{A}_{\text{GradDrop}}(J)$ is a random vector. Given $J \in \mathbb{R}^{m \times n}$, let $|J| \in \mathbb{R}^{m \times n}$ be the element-wise absolute value of $J$. Let $P = \frac{1}{2}\left(\mathbf{1} + \frac{J^{\top} \cdot \mathbf{1}}{|J|^{\top} \cdot \mathbf{1}}\right) \in \mathbb{R}^n$, where the division is element-wise. Each coordinate $i \in [n]$ is independently assigned to the set $\mathcal{I}_+$ with probability $P_i$ and to the set $\mathcal{I}_-$ otherwise. The aggregation at coordinate $i \in \mathcal{I}_+$ is given by the sum of all positive $J_{ji}$, for $j \in [m]$. The aggregation at coordinate $i \in \mathcal{I}_-$ is given by the sum of all negative $J_{ji}$, for $j \in [m]$. Formally,

$$\mathcal{A}_{\text{GradDrop}}(J) = \left( \sum_{i \in \mathcal{I}_+} \boldsymbol{e}_i \sum_{\substack{j \in [m]: \\ J_{ji} > 0}} J_{ji} \right) + \left( \sum_{i \in \mathcal{I}_-} \boldsymbol{e}_i \sum_{\substack{j \in [m]: \\ J_{ji} < 0}} J_{ji} \right) \tag{30}$$

✗ **Non-conflicting.** If $J = \begin{bmatrix} -2 \\ 1 \end{bmatrix}$, then $P = [1/3]$. Therefore, $\mathbb{P}\left[\mathcal{A}_{\text{GradDrop}}(J) = [-2]\right] = 2/3$ and $\mathbb{P}\left[\mathcal{A}_{\text{GradDrop}}(J) = [1]\right] = 1/3$, i.e. $\mathcal{A}_{\text{GradDrop}}(J)$ is in the dual cone of the rows of $J$ with probability 0. Therefore, $\mathcal{A}_{\text{GradDrop}}$ is not non-conflicting. Here, $\mathbb{E}\left[\mathcal{A}_{\text{GradDrop}}(J)\right] = [-1]^\top$, so $\mathcal{A}_{\text{GradDrop}}$ is neither non-conflicting in expectation.

✗ **Linear under scaling.** If $J = \begin{bmatrix} 1 & -1 \\ -1 & 1 \end{bmatrix}$, then $P = \frac{1}{2} \cdot \mathbf{1}$ and the aggregation is one of the four vectors $[\pm 1 \quad \pm 1]^\top$ with equal probability. Scaling the first line of $J$ by 2 yields $J = \begin{bmatrix} 2 & -2 \\ -1 & 1 \end{bmatrix}$ and $P = [2/3 \quad 1/3]^\top$, which cannot lead to a uniform distribution over four elements. Therefore, $\mathcal{A}_{\text{GradDrop}}$ is not linear under scaling.

✗ **Weighted.** With $J = \begin{bmatrix} 1 & -1 \\ -1 & 1 \end{bmatrix}$, the span of $J$ does not include $[1 \quad 1]^\top$ nor $[-1 \quad -1]^\top$. Therefore, $\mathcal{A}_{\text{GradDrop}}$ is not weighted.

### B.6 IMTL-G

In [Liu et al. (2021b)](), the authors describe a method to impartially balance gradients by weighting them. Let $\boldsymbol{g}_i$ be the $i$'th row of $J$ and let $\boldsymbol{u}_i = \frac{\boldsymbol{g}_i}{\|\boldsymbol{g}_i\|}$. They want to find a combination $\boldsymbol{g} = \sum_{i=1}^m \alpha_i \boldsymbol{g}_i$ such that $\boldsymbol{g}^\top \boldsymbol{u}_i$ is equal for all $i$. Let $U = [\boldsymbol{u}_1 - \boldsymbol{u}_2 \quad \ldots \quad \boldsymbol{u}_1 - \boldsymbol{u}_m]^\top$, $D = [\boldsymbol{g}_1 - \boldsymbol{g}_2 \quad \ldots \quad \boldsymbol{g}_1 - \boldsymbol{g}_m]^\top$. If $\boldsymbol{\alpha}_{2:m} = [\alpha_2 \quad \ldots \quad \alpha_m]^\top$, then $\boldsymbol{\alpha}_{2:m} = \left(UD^\top\right)^{-1} U \cdot \boldsymbol{g}_1$ and $\alpha_1 = 1 - \sum_{i=2}^m \alpha_i$. Notice that this is defined only when the gradients are linearly independent. Thus, this is not strictly speaking an aggregator since it can only be computed on matrices of rank $m$. We thus propose a generalization defined for matrices of any rank that is equivalent when the matrix has rank $m$. In the original formulation, requiring $\boldsymbol{g}^\top \boldsymbol{u}_i$ to be equal to some $c \in \mathbb{R}$ for all $i$, is equivalent to requiring that for all $i$, $\boldsymbol{g}^\top \boldsymbol{g}_i$ is equal to $c \|\boldsymbol{g}_i\|$. Writing $\boldsymbol{g} = J^\top \boldsymbol{\alpha}$ and letting $\boldsymbol{d} \in \mathbb{R}^m$ be the vector of norms of the rows of $J$, the objective is thus to find $\boldsymbol{\alpha}$ satisfying $JJ^\top \boldsymbol{\alpha} \propto \boldsymbol{d}$. Besides, to match the original formulation, the elements of $\boldsymbol{\alpha}$ should sum to 1.

Letting $\left(JJ^\top\right)^\dagger$ be the Moore-Penrose pseudo inverse of $JJ^\top$, we define

$$\mathcal{A}_{\text{IMTL-G}}(J) = J^\top \cdot \boldsymbol{w} \tag{31}$$

$$\text{with} \quad \boldsymbol{w} = \begin{cases} \frac{\boldsymbol{v}}{\mathbf{1}^\top \boldsymbol{v}}, & \text{if } \mathbf{1}^\top \boldsymbol{v} \neq \mathbf{0} \\ \mathbf{0}, & \text{otherwise} \end{cases} \tag{32}$$

$$\text{and} \quad \boldsymbol{v} = \left(JJ^\top\right)^\dagger \cdot \boldsymbol{d}. \tag{33}$$

✗ **Non-conflicting.** If $J = [1 \quad -1 \quad -1]^\top$, then $\boldsymbol{d} = [1 \quad 1 \quad 1]^\top$, $JJ^\top = \begin{bmatrix} 1 & -1 & -1 \\ -1 & 1 & 1 \\ -1 & 1 & 1 \end{bmatrix}$, and thus $\boldsymbol{v} = \frac{1}{9} [-1 \quad 1 \quad 1]^\top$. Therefore, $\boldsymbol{w} = [-1 \quad 1 \quad 1]^\top$, $\mathcal{A}_{\text{IMTL-G}}(J) = [-3]^\top$ and $J \cdot \mathcal{A}_{\text{IMTL-G}}(J) = [-3 \quad 3 \quad 3]^\top$. $\mathcal{A}_{\text{IMTL-G}}$ is thus not non-conflicting.

It should be noted that when $J$ has rank $m$, $\mathcal{A}_{\text{IMTL-G}}$ seems to be non-conflicting. Thus, it would be possible to make a different non-conflicting generalization, for instance, by deciding $\mathbf{0}$ when $J$ is not full rank.

✗ **Linear under scaling.** With $J = \begin{bmatrix} a & 0 \\ 0 & 1 \end{bmatrix}$ and $a > 0$, we have $\boldsymbol{v} = \begin{bmatrix} 1/a^2 & 0 \\ 0 & 1 \end{bmatrix} \cdot \begin{bmatrix} a \\ 1 \end{bmatrix} = \begin{bmatrix} 1/a \\ 1 \end{bmatrix}$ and $\mathcal{A}_{\text{IMTL-G}}(J) = \begin{bmatrix} a & 0 \\ 0 & 1 \end{bmatrix} \cdot \begin{bmatrix} 1/a \\ 1 \end{bmatrix} \cdot \frac{1}{\frac{1}{a}+1} = \frac{1}{\frac{1}{a}+1} \cdot \begin{bmatrix} 1 \\ 1 \end{bmatrix}$. This is not affine in $a$, so $\mathcal{A}_{\text{IMTL-G}}$ is not linear under scaling.

✓ **Weighted.** By (31), $\mathcal{A}_{\text{IMTL-G}}$ is weighted.

### B.7 CAGRAD

$\mathcal{A}_{\text{CAGrad}}$ is described in Liu et al. (2021a). It is parameterized by $c \in [0, 1[$. If $c = 0$, this is equivalent to $\mathcal{A}_{\text{Mean}}$. Therefore, we restrict our analysis to the case $c > 0$. For any $J \in \mathbb{R}^{m \times n}$, let $\bar{g}$ be the average gradient $\frac{1}{m} J^\top \cdot \mathbf{1}$, and let $e_i^\top J$ denote the $i$'th row of $J$. The aggregation is then defined as

$$\mathcal{A}_{\text{CAGrad}}(J) \in \underset{\substack{d \in \mathbb{R}^n: \\ \|d - \bar{g}\| \le c\|\bar{g}\|}}{\arg\max} \ \min_{i \in [m]} e_i^\top J d \tag{34}$$

**✗ Non-conflicting.** Let $J = \begin{bmatrix} 2 & 0 \\ -2a - 2 & 2 \end{bmatrix}$, with $a$ satisfying $-a + c\sqrt{a^2 + 1} < 0$. We have $\bar{g} = \begin{bmatrix} -a & 1 \end{bmatrix}^\top$ and $\|\bar{g}\| = \sqrt{a^2 + 1}$. Observe that any $d \in \mathbb{R}^n$ satisfying the constraint $\|d - \bar{g}\| \le c\|\bar{g}\|$ has first coordinate at most $-a + c\sqrt{a^2 + 1}$. Because $-a + c\sqrt{a^2 + 1} < 0$, any feasible $d$ has a negative first coordinate, making $d$ conflict with the first row of $J$. For any $c \in [0, 1[$, $-a + c\sqrt{a^2 + 1} < 0$ is equivalent to $\sqrt{\frac{c^2}{1-c^2}} < a$. Thus, this provides a counter-example to the non-conflicting property for any $c \in [0, 1[$, i.e. $\mathcal{A}_{\text{CAGrad}}$ is not non-conflicting.

If we generalize to the case $c \ge 1$, as suggested in the original paper, then $d = \mathbf{0}$ becomes feasible, which yields $\min_{i \in [m]} e_i^\top J d = 0$. Therefore the optimal $d$ satisfies $0 \le \min_{i \in [m]} e_i^\top J d$, i.e. $\mathbf{0} \le J d$. With $c \ge 1$, $\mathcal{A}_{\text{CAGrad}}$ would thus be non-conflicting.

**✗ Linear under scaling (sketch of proof).** Let $J = \begin{bmatrix} 2 & 0 \\ 0 & 2a \end{bmatrix}$, then $\bar{g} = \begin{bmatrix} 1 & a \end{bmatrix}^\top$ and $\|\bar{g}\| = \sqrt{1 + a^2}$. One can show that the constraint $\|d - \bar{g}\| \le c\|\bar{g}\|$ needs to be satisfied with equality since, otherwise, we can scale $d$ to make the objective larger. Substituting $J$ in $\min_{i \in [m]} e_i^\top J d$ yields $2\min(d_1, a d_2)$. For any $a$ satisfying $c\sqrt{1 + a^2} + 1 < a^2$, it can be shown that the optimal $d$ satisfies $d_1 < a d_2$. In that case the inner minimum over $i$ is $2d_1$ and, to satisfy $\|d - \bar{g}\| = c\|\bar{g}\|$, the KKT conditions over the Lagrangian yield $d - \bar{g} \propto \nabla_d d_1 = \begin{bmatrix} 1 & 0 \end{bmatrix}^\top$. This yields $d = \begin{bmatrix} c \cdot \|\bar{g}\| + 1 \\ a \end{bmatrix} = \begin{bmatrix} c\sqrt{1 + a^2} + 1 \\ a \end{bmatrix}$. This is not affine in $a$; therefore, $\mathcal{A}_{\text{CAGrad}}$ is not linear under scaling.

**✓ Weighted.** In Liu et al. (2021a), $\mathcal{A}_{\text{CAGrad}}$ is formulated via its dual: $\mathcal{A}_{\text{CAGrad}}(J) = \frac{1}{m} J^\top \left( \mathbf{1} + \frac{c\|J^\top \mathbf{1}\|}{\|J^\top w\|} w \right)$, with $w \in \arg\min_{w \in \Delta(m)} \mathbf{1}^\top J J^\top w + c \cdot \|J^\top \mathbf{1}\| \cdot \|J^\top w\|$, where is $\Delta(m)$ the probability simplex of dimension $m$. Therefore, $\mathcal{A}_{\text{CAGrad}}$ is weighted.

### B.8 RGW

$\mathcal{A}_{\text{RGW}}$ is defined in Lin et al. (2021) as the weighted sum of the rows of the input matrix, with a random weighting. The weighting is obtained by sampling $m$ i.i.d. normally distributed random variables and applying a softmax. Formally,

$$\mathcal{A}_{\text{RGW}}(J) = J^\top \cdot \text{softmax}(\mathbf{w}) \tag{35}$$
$$\text{with} \quad \mathbf{w} \sim \mathcal{N}(\mathbf{0}, I) \tag{36}$$

**✗ Non-conflicting.** When $J = \begin{bmatrix} 1 \\ -2 \end{bmatrix}$, the only non-conflicting solution is $\mathbf{0}$. However, $\mathbb{P}[\mathcal{A}_{\text{RGW}}(J) = \mathbf{0}] = 0$, i.e. $\mathcal{A}_{\text{RGW}}(J)$ is in the dual cone of the rows of $J$ with probability 0. $\mathcal{A}_{\text{RGW}}$ is thus not non-conflicting. Here,

$$\mathbb{E}[\mathcal{A}_{\text{RGW}}(J)] = \mathbb{E}\left[ \frac{e^{\mathbf{w}_1} - 2e^{\mathbf{w}_2}}{e^{\mathbf{w}_1} + e^{\mathbf{w}_2}} \right] \qquad (\text{By } (35))$$

$$= -\mathbb{E}\left[ \frac{e^{\mathbf{w}_1}}{e^{\mathbf{w}_1} + e^{\mathbf{w}_2}} \right] \qquad (\mathbf{w} \sim \mathcal{N}(\mathbf{0}, I))$$

$$< 0 \qquad (0 < e^{\mathbf{w}})$$

so $\mathcal{A}_{\text{RGW}}$ is neither non-conflicting in expectation.

✓ **Linear under scaling.** $\mathcal{A}_{\text{RGW}}\big(\operatorname{diag}(\boldsymbol{c}) \cdot J\big) = \big(\operatorname{diag}(\boldsymbol{c}) \cdot J\big)^{\top} \cdot \operatorname{softmax}(\mathbf{w}) = J^{\top} \cdot \operatorname{diag}(\boldsymbol{c}) \cdot \operatorname{softmax}(\mathbf{w}) = J^{\top} \cdot \operatorname{diag}\big(\operatorname{softmax}(\mathbf{w})\big) \cdot \boldsymbol{c}$. We thus have $\mathcal{A}_{\text{RGW}}\big(\operatorname{diag}(\boldsymbol{c}) \cdot J\big) = \mathfrak{J}^{\top} \cdot \boldsymbol{c}$ with $\mathfrak{J}^{\top} = J^{\top} \cdot \operatorname{diag}\big(\operatorname{softmax}(\mathbf{w})\big)$. Therefore, $\mathcal{A}_{\text{RGW}}$ is linear under scaling.

✓ **Weighted.** By (35), $\mathcal{A}_{\text{RGW}}$ is weighted.

## B.9 NASH-MTL

Nash-MTL is described in Navon et al. (2022). Unfortunately, we were not able to verify the proof of Claim 3.1, and we believe that the official implementation of Nash-MTL may mismatch the desired objective by which it is defined. Therefore, we only analyze the initial objective even though our experiments for this aggregator are conducted with the official implementation.

Let $J \in \mathbb{R}^{m \times n}$ and $\varepsilon > 0$. Let also $B_{\varepsilon} = \big\{\boldsymbol{d} \in \mathbb{R}^{n} : \|\boldsymbol{d}\| \leq \varepsilon, \boldsymbol{0} \leq J\boldsymbol{d}\big\}$. With $\boldsymbol{e}_i^{\top} J$ denoting the $i$'th row of $J$, $\mathcal{A}_{\text{Nash-MTL}}$ is then defined as

$$\mathcal{A}_{\text{Nash-MTL}}(J) = \underset{\boldsymbol{d} \in B_{\varepsilon}}{\arg\max} \sum_{i \in [m]} \log\big(\boldsymbol{e}_i^{\top} J\boldsymbol{d}\big) \tag{37}$$

✓ **Non-conflicting.** By the constraint, $\mathcal{A}_{\text{Nash-MTL}}$ is non-conflicting.

✗ **Linear under scaling.** If an aggregator $\mathcal{A}$ is linear under scaling, it should be the case that $\mathcal{A}(aJ) = a\mathcal{A}(J)$ for any scalar $a > 0$ and any $J \in \mathbb{R}^{m \times n}$. However, $\log(a\boldsymbol{e}_i^{\top} J\boldsymbol{d}) = \log(\boldsymbol{e}_i^{\top} J\boldsymbol{d}) + \log(a)$. This means that scaling by a scalar does not impact aggregation. Since this is not the trivial $\boldsymbol{0}$ aggregator, $\mathcal{A}_{\text{Nash-MTL}}$ is not linear under scaling.

✓ **Weighted.** Suppose towards contradiction that $\boldsymbol{d}$ is both optimal for (37) and not in the span of $J^{\top}$. Let $\boldsymbol{d}'$ be the projection of $\boldsymbol{d}$ onto the span of $J^{\top}$. Since $\|\boldsymbol{d}'\| < \|\boldsymbol{d}\| < \varepsilon$ and $J\boldsymbol{d} = J\boldsymbol{d}'$, we have $J\boldsymbol{d} \prec J\left(\frac{\|\boldsymbol{d}\|}{\|\boldsymbol{d}'\|}\boldsymbol{d}'\right)$, contradicting the optimality of $\boldsymbol{d}$. Therefore, $\mathcal{A}_{\text{Nash-MTL}}$ is weighted.

## B.10 ALIGNED-MTL

The Aligned-MTL method for balancing the Jacobian is described in Senushkin et al. (2023). For simplicity, we fix the vector of preferences to $\frac{1}{m}\mathbf{1}$, but the proofs can be adapted for any non-trivial vector. Given $J \in \mathbb{R}^{m \times n}$, let $V\Sigma^2 V^{\top}$ be the eigen-decomposition of $JJ^{\top}$, let $\Sigma^{\dagger}$ be the diagonal matrix whose non-zero elements are the inverse of corresponding non-zero diagonal elements of $\Sigma$ and let $\sigma_{\min} = \min_{i \in [m], \Sigma_{ii} \neq 0} \Sigma_{ii}$. The aggregation is then defined as

$$\mathcal{A}_{\text{Aligned-MTL}}(J) = \frac{1}{m} J^{\top} \cdot \boldsymbol{w} \tag{38}$$

$$\text{with} \qquad \boldsymbol{w} = \sigma_{\min} \cdot V\Sigma^{\dagger} V^{\top} \cdot \mathbf{1} \tag{39}$$

✗ **Non-conflicting.** If the SVD of $J$ is $V\Sigma U^{\top}$, then $J^{\top}\boldsymbol{w} = \sigma_{\min} U P V^{\top}\mathbf{1}$ with $P = \Sigma^{\dagger}\Sigma$ a diagonal projection matrix with 1s corresponding to non zero elements of $\Sigma$ and 0s everywhere else. Further, $J \cdot \mathcal{A}_{\text{Aligned-MTL}}(J) = \frac{\sigma_{\min}}{m} \cdot V\Sigma V^{\top}\mathbf{1}$. If $V = \frac{1}{2}\begin{bmatrix} \sqrt{3} & 1 \\ -1 & \sqrt{3} \end{bmatrix}$ and $\Sigma = \begin{bmatrix} 1 & 0 \\ 0 & 0 \end{bmatrix}$, we have $J \cdot \mathcal{A}_{\text{Aligned-MTL}}(J) = \frac{1}{2} \cdot V\Sigma V^{\top}\mathbf{1} = \frac{1}{8}\begin{bmatrix} 3 - \sqrt{3} & 1 - \sqrt{3} \end{bmatrix}^{\top}$ which is not non-negative. $\mathcal{A}_{\text{Aligned-MTL}}$ is thus not non-conflicting.

✗ **Linear under scaling.** If $J = \begin{bmatrix} 1 & 0 \\ 0 & a \end{bmatrix}$, then $U = V = I$, $\Sigma = J$. For $0 < a \leq 1$, $\sigma_{\min} = a$, therefore $\mathcal{A}_{\text{Aligned-MTL}}(J) = \frac{a}{2} \cdot \mathbf{1}$. For $1 < a$, $\sigma_{\min} = 1$ and therefore $\mathcal{A}_{\text{Aligned-MTL}}(J) = \frac{1}{2} \cdot \mathbf{1}$, which makes $\mathcal{A}_{\text{Aligned-MTL}}$ not linear under scaling.

✓ **Weighted.** By (38), $\mathcal{A}_{\text{Aligned-MTL}}$ is weighted.

## C  Experimental settings

For all of our experiments, we used `PyTorch` (Paszke et al., 2019). We have developed an open-source library[4] on top of it to enable Jacobian descent easily. This library is designed to be reusable for many other use cases than the experiments presented in our work. To separate them from the library, the experiments have been conducted with a different code repository[5] mainly using `PyTorch` and our library.

### C.1  Learning rate selection

The learning rate has a very important impact on the speed of optimization. To make the comparisons as fair as possible, we always show the results corresponding to the best learning rate. We have selected the area under the loss curve as the criterion to compare the learning rates. This choice is arbitrary but seems to work well in practice: a lower area under the loss curve means the optimization is fast (quick loss decrease) and stable (few bumps in the loss curve). Concretely, for each random rerun and each aggregator, we first try 22 learning rates from $10^{-5}$ to $10^2$, increasing by a factor $10^{\frac{1}{3}}$ every time. The two best learning rates from this range then define a refined range of plausible good learning rates, going from the smallest of those two multiplied by $10^{-\frac{1}{3}}$ to the largest of those two multiplied by $10^{\frac{1}{3}}$. This margin makes it unlikely for the best learning rate to lie out of the refined range. After this, 50 learning rates from the refined range are tried. These learning rates are evenly spaced in the exponent domain. The one with the best area under the loss curve is then selected and presented in the plots. For simplicity, we have always used a constant learning rate, i.e. no learning rate scheduler was used.

This approach has the advantage of being simple and precise, thus giving trustworthy results. However, it requires 72 trainings for each aggregator, random rerun, and dataset, i.e. a total of 43776 trainings for all of our experiments. For this reason, we have opted to work on small subsets of the original datasets.

### C.2  Random reruns and standard error of the mean

To get an idea of confidence in our results, every experiment is performed 8 times on a different seed and a different subset, of size 1024, of the training dataset. The seed used for run $i \in [8]$ is always simply set to $i$. Because each random rerun includes the full learning rate selection method described in Appendix C.1, it is sensible to consider the 8 sets of results as i.i.d. For each point of both the loss curves and the cosine similarity curves, we thus compute the estimated standard error of the mean with the usual formula $\frac{1}{\sqrt{8}} \sqrt{\frac{\sum_{i \in [8]} (v_i - \bar{v})^2}{8-1}}$, where $v_i$ is the value of a point of the curve for random rerun $i$, and $\bar{v}$ is the average value of this point over the 8 runs.

### C.3  Model architectures

In all experiments, the models are simple convolutional neural networks. All convolutions always have a stride of $1{\times}1$, a kernel size of $3{\times}3$, a learnable bias, and no padding. All linear layers always have a learnable bias. The activation function is the exponential linear unit (Clevert et al., 2015). The full architectures are given in Tables 2, 3, 4 and 5. Note that these architectures have been fixed arbitrarily, i.e. they were not optimized through some hyper-parameter selection. The weights of the model have been initialized with the default initialization scheme of `PyTorch`.

### C.4  Optimizer

For all experiments except those described in Appendix D.3, we always use the basic `SGD` optimizer of `PyTorch`, without any regularization or momentum. Here, `SGD` refers to the `PyTorch` optimizer that updates the parameters of the model in the opposite direction of the gradient, which, in our case, is replaced by the aggregation of the Jacobian matrix. In the rest of this paper, SGD refers to the

---

[4]Available at https://github.com/***/***
[5]Available at https://github.com/***/***

Table 2: Architecture used for SVHN

| |
|---|
| `Conv2d` (3 input channels, 16 output channels, 1 group), `ELU` |
| `Conv2d` (16 input channels, 32 output channels, 16 groups) |
| `MaxPool2d` (stride of 2×2, kernel size of 2×2), `ELU` |
| `Conv2d` (32 input channels, 32 output channels, 32 groups) |
| `MaxPool2d` (stride of 3×3, kernel size of 3×3), `ELU`, `Flatten` |
| `Linear` (512 input features, 64 output features), `ELU` |
| `Linear` (64 input features, 10 outputs) |

Table 3: Architecture used for CIFAR-10

| |
|---|
| `Conv2d` (3 input channels, 32 output channels, 1 group), `ELU` |
| `Conv2d` (32 input channels, 64 output channels, 32 groups) |
| `MaxPool2d` (stride of 2×2, kernel size of 2× 2), `ELU` |
| `Conv2d` (64 input channels, 64 output channels, 64 groups) |
| `MaxPool2d` (stride of 3×3, kernel size of 3×3), `ELU`, `Flatten` |
| `Linear` (1024 input features, 128 output features), `ELU` |
| `Linear` (128 input features, 10 outputs) |

Table 4: Architecture used for EuroSAT

| |
|---|
| `Conv2d` (3 input channels, 32 output channels, 1 group) |
| `MaxPool2d` (stride of 2×2, kernel size of 2×2), `ELU` |
| `Conv2d` (32 input channels, 64 output channels, 32 groups) |
| `MaxPool2d` (stride of 2×2, kernel size of 2×2), `ELU` |
| `Conv2d` (64 input channels, 64 output channels, 64 groups) |
| `MaxPool2d` (stride of 3×3, kernel size of 3×3), `ELU`, `Flatten` |
| `Linear` (1024 input features, 128 output features), `ELU` |
| `Linear` (128 input features, 10 outputs) |

Table 5: Architecture used for MNIST, Fashion-MNIST and Kuzushiji-MNIST

| |
|---|
| `Conv2d` (1 input channel, 32 output channels, 1 group), `ELU` |
| `Conv2d` (32 input channels, 64 output channels, 1 group) |
| `MaxPool2d` (stride of 2×2, kernel size of 2×2), `ELU` |
| `Conv2d` (64 input channels, 64 output channels, 1 group) |
| `MaxPool2d` (stride of 3×3, kernel size of 3×3), `ELU`, `Flatten` |
| `Linear` (576 input features, 128 output features), `ELU` |
| `Linear` (128 input features, 10 outputs) |

whole stochastic gradient descent algorithm. In the experiments of Appendix D.3, we instead use `Adam` to study its interactions with JD.

### C.5 LOSS FUNCTION

The loss function is always the usual cross-entropy, with the default parameters of `PyTorch`.

### C.6 PREPROCESSING

The inputs are always normalized per channel based on the mean and standard deviation computed on the entire training split of the dataset.

### C.7 ITERATIONS AND COMPUTATIONAL BUDGET

The numbers of epochs and the corresponding numbers of iterations for all datasets are provided in Table 6, along with the required number of NVIDIA L4 GPU-hours, to run all 72 learning rates for the 11 aggregators on a single seed. The total computational budget to run the main experiments on 8 seeds was thus around 760 GPU-hours. Additionally, we used a total of about 100 GPU hours for the experiments varying the batch size and using Adam, and about 200 more GPU hours were used for early investigations.

Table 6: Numbers of epochs, iterations, and GPU-hours for each dataset

| Dataset | Epochs | Iterations | GPU-Hours |
|---|---|---|---|
| SVHN | 25 | 800 | 17 |
| CIFAR-10 | 20 | 640 | 15 |
| EuroSAT | 30 | 960 | 32 |
| MNIST | 8 | 256 | 6 |
| Fashion-MNIST | 25 | 800 | 17 |
| Kuzushiji-MNIST | 10 | 320 | 8 |

# D  ADDITIONAL EXPERIMENTAL RESULTS

In this appendix, we provide additional experimental results about IWRM.

## D.1  ALL DATASETS AND ALL AGGREGATORS

Figures 3, 4, 5, 6, 7 and 8 show the full results of the experiments described in Section 5 on SVHN, CIFAR-10, EuroSAT, MNIST, Fashion-MNIST and Kuzushiji-MNIST, respectively. For readability, the results are displayed on three different plots for each dataset. We always show $\mathcal{A}_{\text{UPGrad}}$ and $\mathcal{A}_{\text{Mean}}$ for reference. The exact experimental settings are described in Appendix C.

It should be noted that some of these aggregators were not developed as general-purpose aggregators, but mainly for the use case of multi-task learning, with one gradient per task. Our experiments present a more challenging setting than multi-task learning optimization because conflict between rows of the Jacobian is typically higher. Besides, for some aggregators, e.g. $\mathcal{A}_{\text{GradDrop}}$ and $\mathcal{A}_{\text{IMTL-G}}$, it was advised to make the aggregation of gradients w.r.t. an internal activation (such as the last shared representation), rather than w.r.t. the parameters of the model (Chen et al., 2020; Liu et al., 2021b). To enable comparison, we instead always aggregated the Jacobian w.r.t. all parameters.

We can see that $\mathcal{A}_{\text{UPGrad}}$ provides a significant improvement over $\mathcal{A}_{\text{Mean}}$ on all datasets. Moreover, the performance gaps seem to be linked to the difficulty of the dataset, which suggests that experimenting with harder tasks is a promising future direction. The intrinsic randomness of $\mathcal{A}_{\text{RGW}}$ and $\mathcal{A}_{\text{GradDrop}}$ reduces the train set performance, but it could positively impact the generalization, which we do not study here. We suspect the disappointing results of $\mathcal{A}_{\text{Nash-MTL}}$ to be caused by issues in the official implementation that we used, leading to instability.

## D.2  VARYING THE BATCH SIZE

Figure 9 shows the results on CIFAR-10 with $\mathcal{A}_{\text{UPGrad}}$ when varying the batch size from 4 to 64. Concretely, because we are using SSJD, this makes the number of rows of the sub-Jacobian aggregated at each step vary from 4 to 64. Recall that IWRM with SSJD and $\mathcal{A}_{\text{Mean}}$ is equivalent to ERM with SGD. We observe that with a small batch size, $\mathcal{A}_{\text{UPGrad}}$ becomes very similar to $\mathcal{A}_{\text{Mean}}$. This is not surprising since both would be equivalent with a batch size of 1. Conversely, a larger batch size increases the gap between $\mathcal{A}_{\text{UPGrad}}$ and $\mathcal{A}_{\text{Mean}}$. Since the projections of $\mathcal{A}_{\text{UPGrad}}$ are onto the dual cone of more rows, each step becomes non-conflicting with respect to more of the original 1024 objectives, pushing even further the benefits of the non-conflicting property. In other words, increasing the batch size refines the dual cone, thereby improving the quality of the projections. It would be interesting to theoretically analyze the impact of the batch size in this setting.

## D.3  COMPATIBILITY WITH ADAM

Figure 10 gives the results on CIFAR-10 and SVHN when using `Adam` rather than the `SGD` optimizer. Concretely, this corresponds to the Adam algorithm in which the gradient is replaced by the aggregation of the Jacobian. The learning rate is still tuned as described in Appendix C.1, but the other hyperparameters of Adam are fixed to the default values of `PyTorch`, i.e. $\beta_1 = 0.9$, $\beta_2 = 0.999$ and $\epsilon = 10^{-8}$. Because optimization with Adam is faster, the number of epochs for SVHN and CIFAR-10 is reduced to 20 and 15, respectively. While the performance gap is smaller with this optimizer, it is still significant and suggests that our methods are beneficial with other optimizers than the simple `SGD`. Note that this analysis is fairly superficial. The thorough investigation of the interplay between aggregators and momentum-based optimizers is a compelling future research direction.

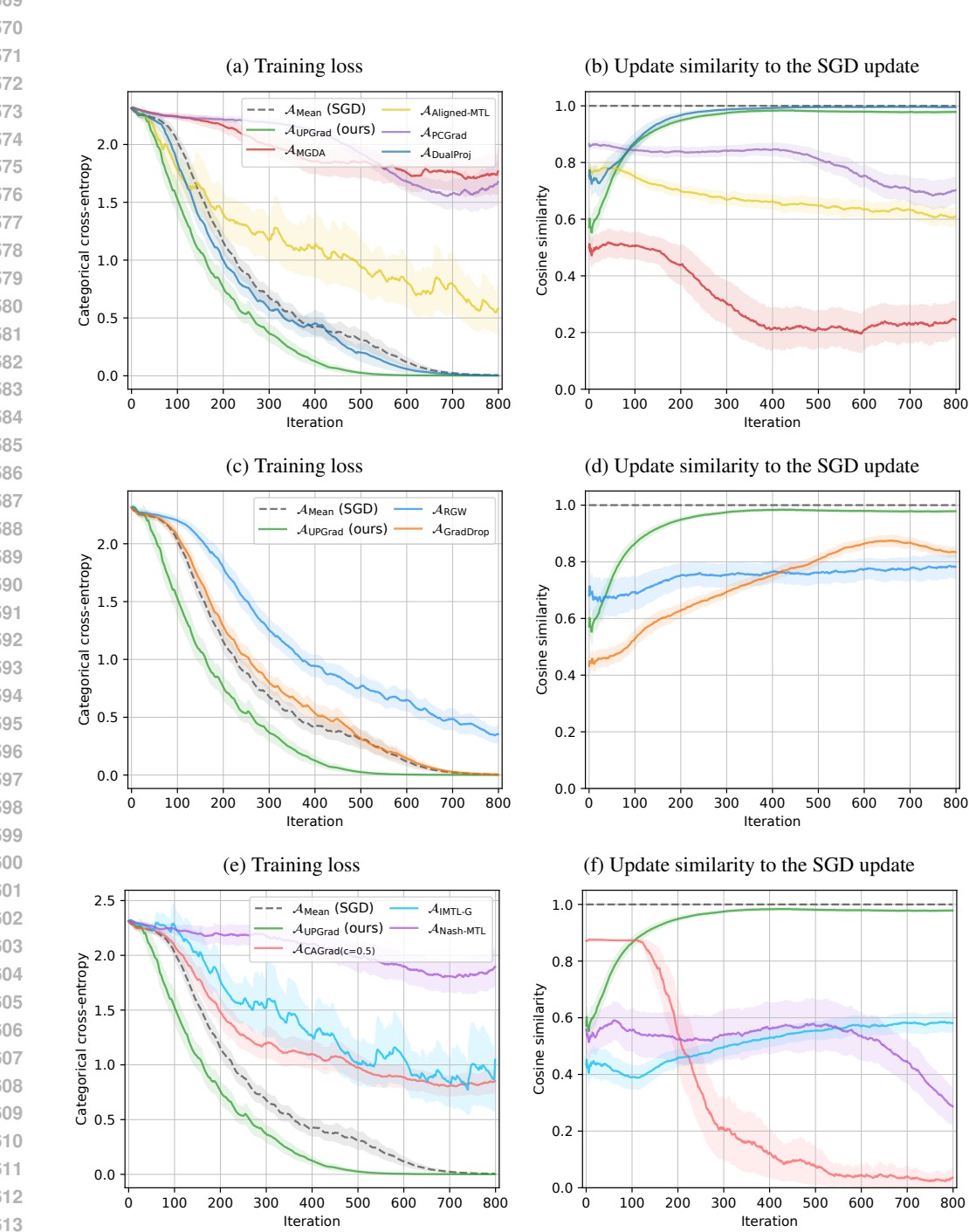

Figure 3: SVHN results.

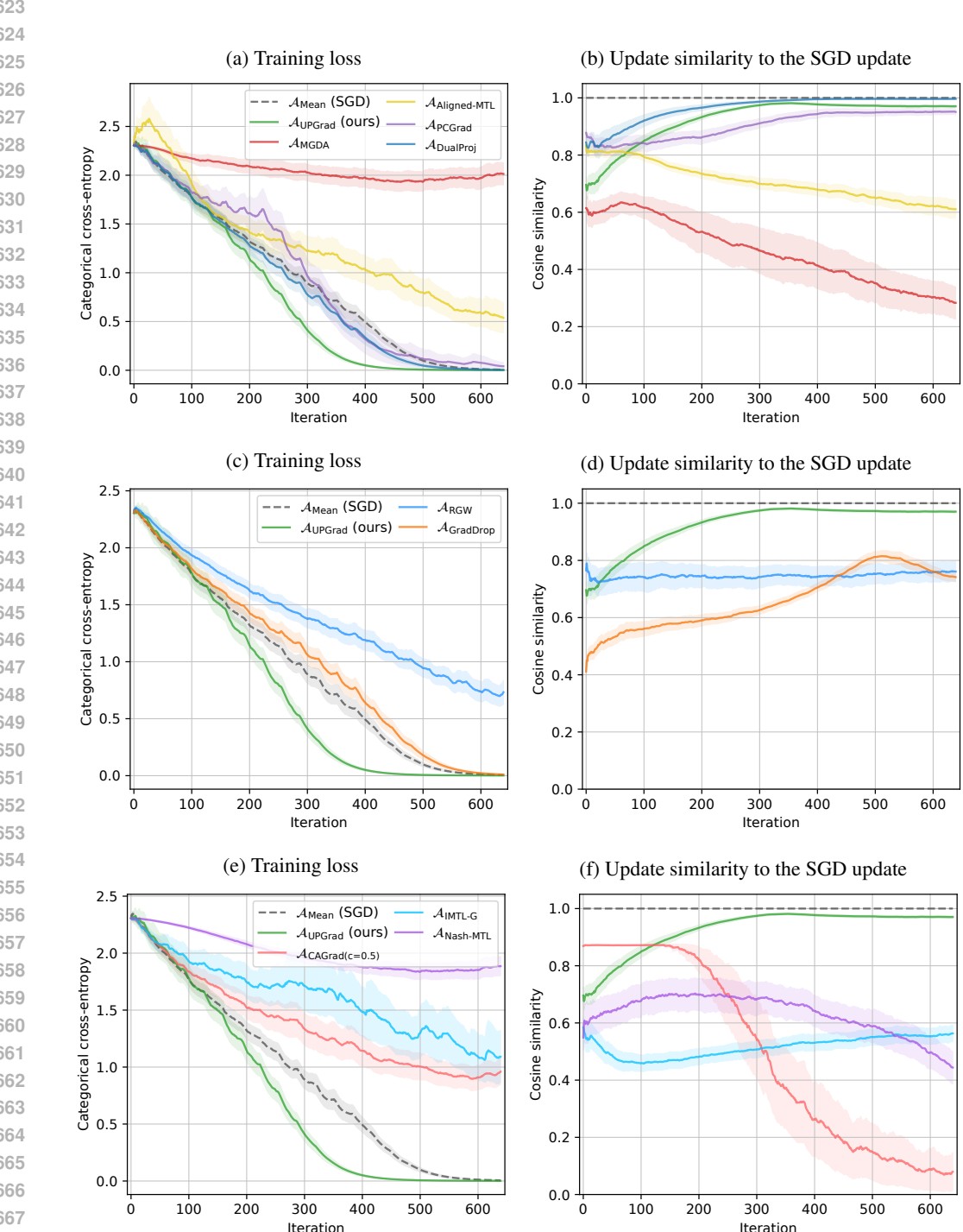

Figure 4: CIFAR-10 results.

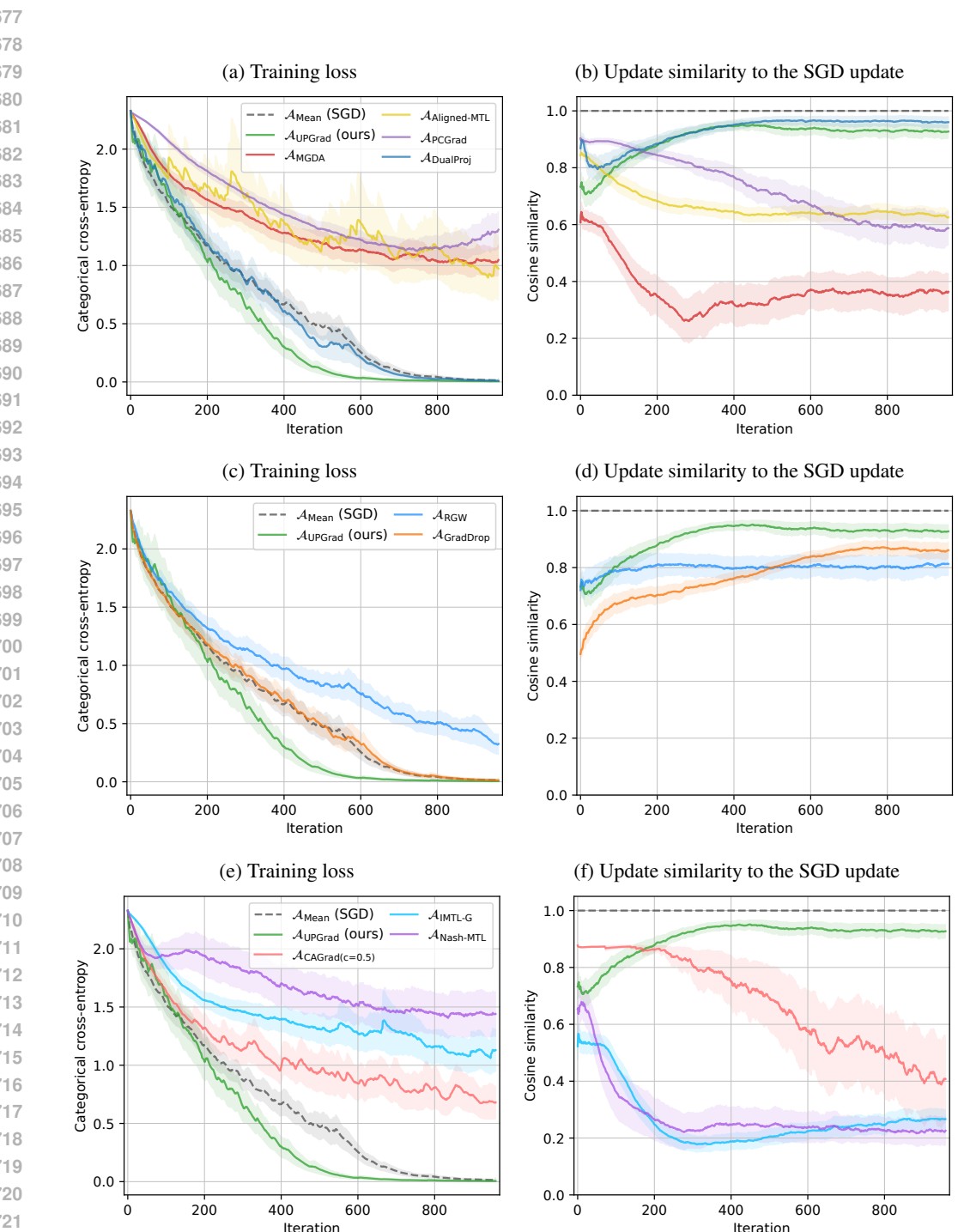

Figure 5: EuroSAT results.

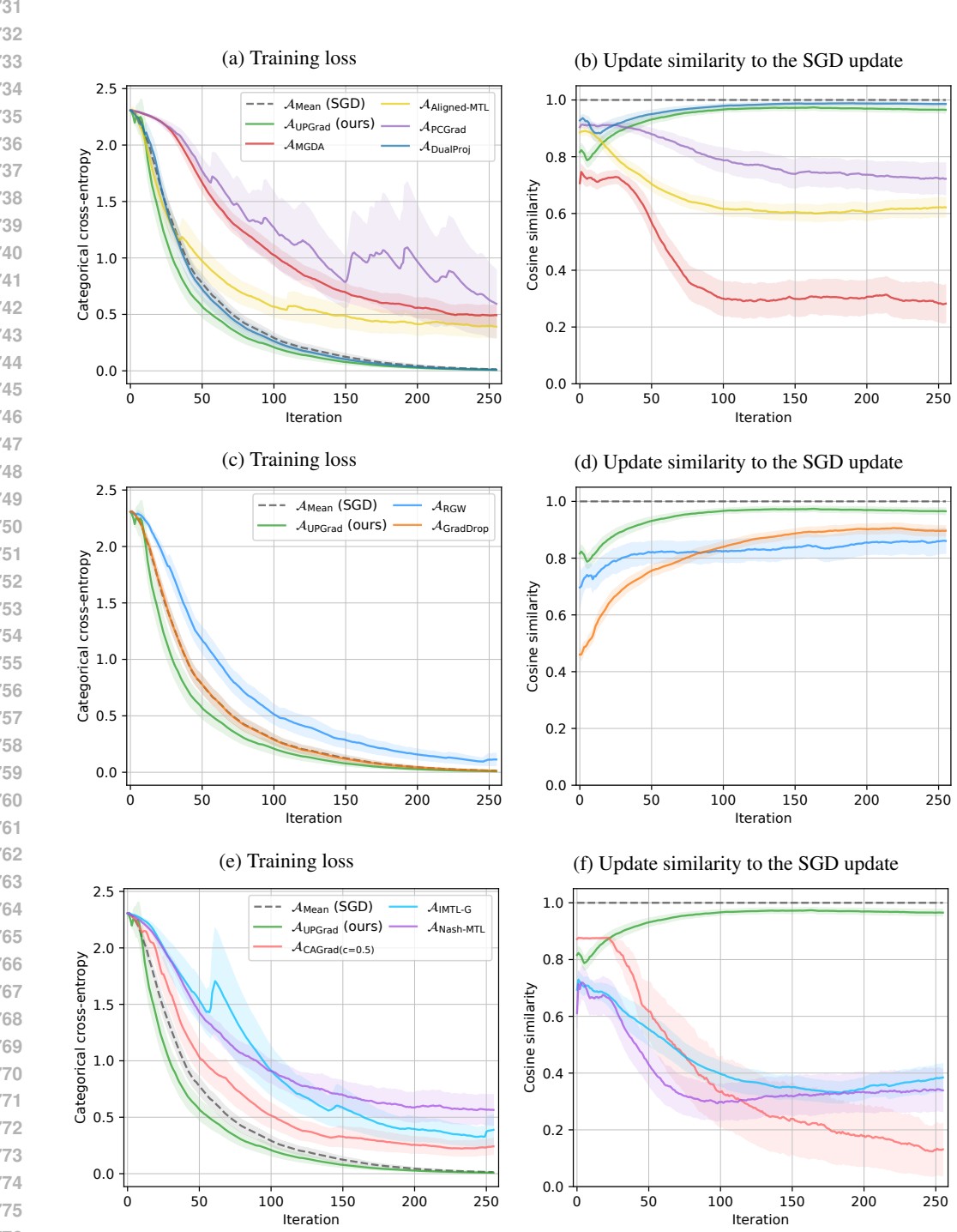

Figure 6: MNIST results.

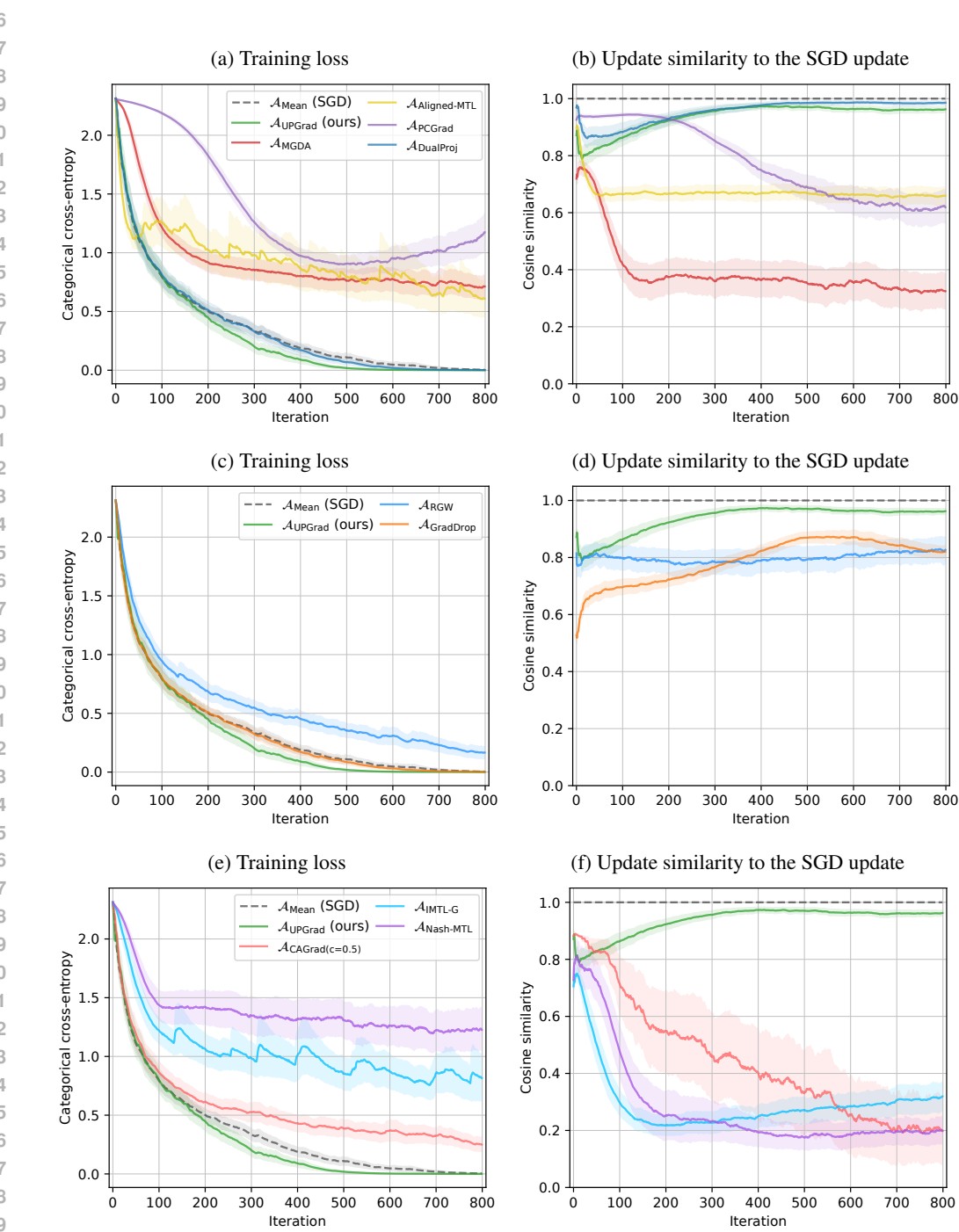

Figure 7: Fashion-MNIST results.

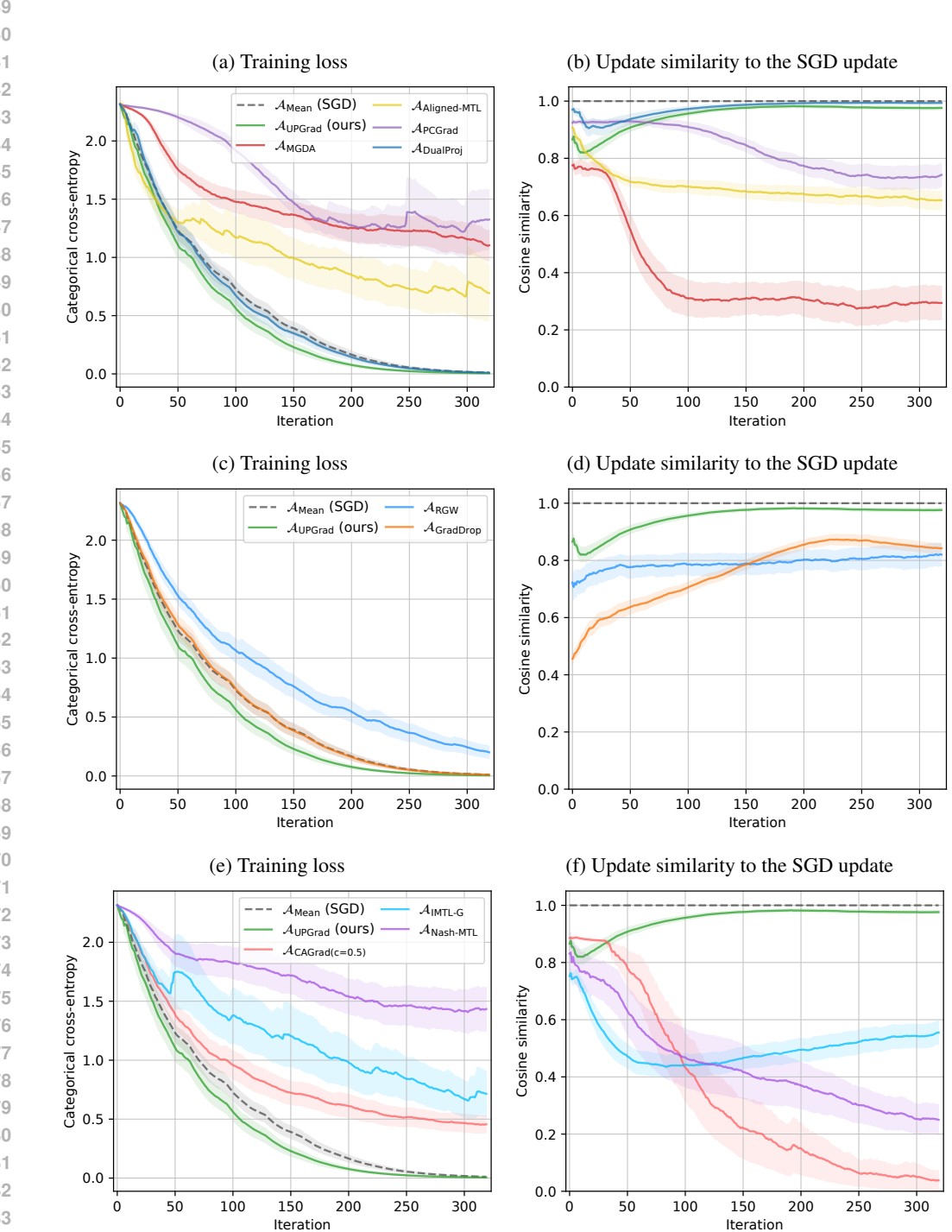

Figure 8: Kuzushiji-MNIST results.

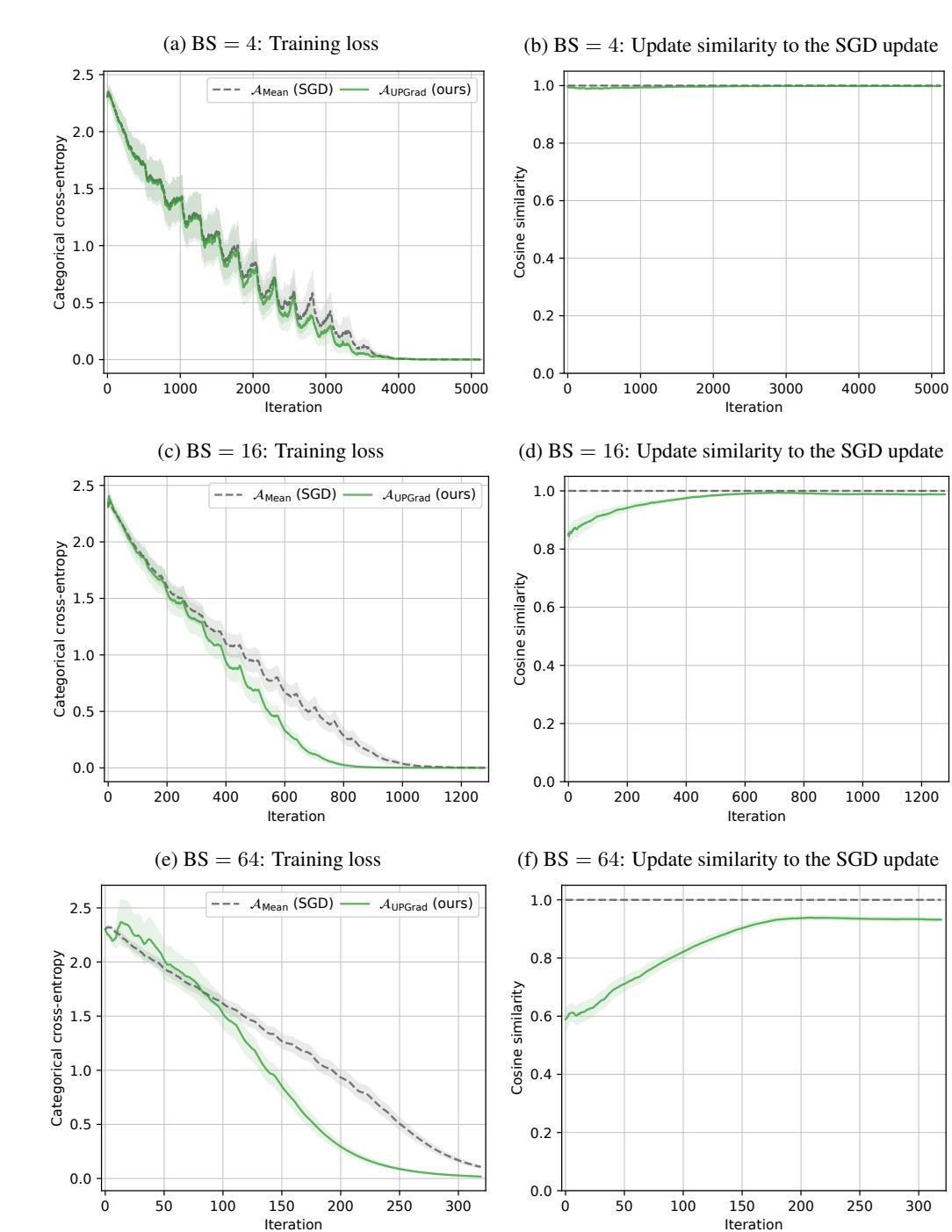

Figure 9: CIFAR-10 results with different batch sizes (BS). The number of epochs is always 20, so the number of iterations varies.

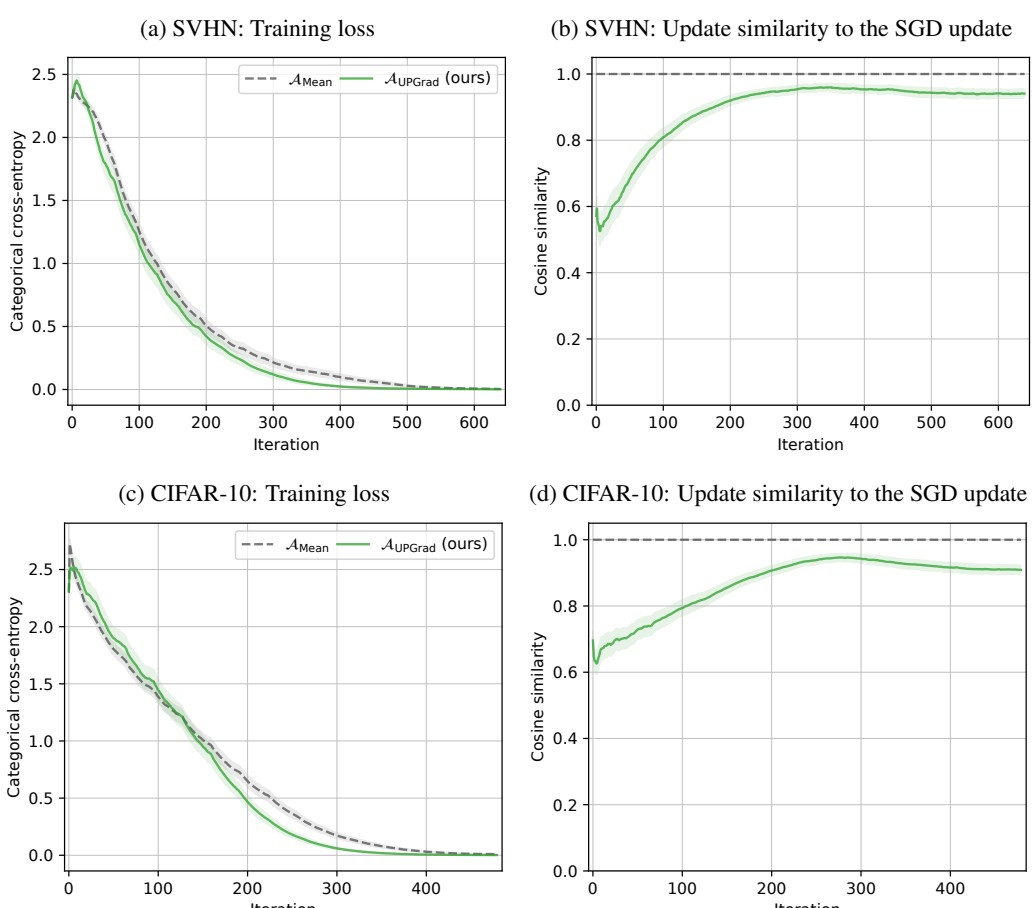

Figure 10: Results with the `Adam` optimizer.

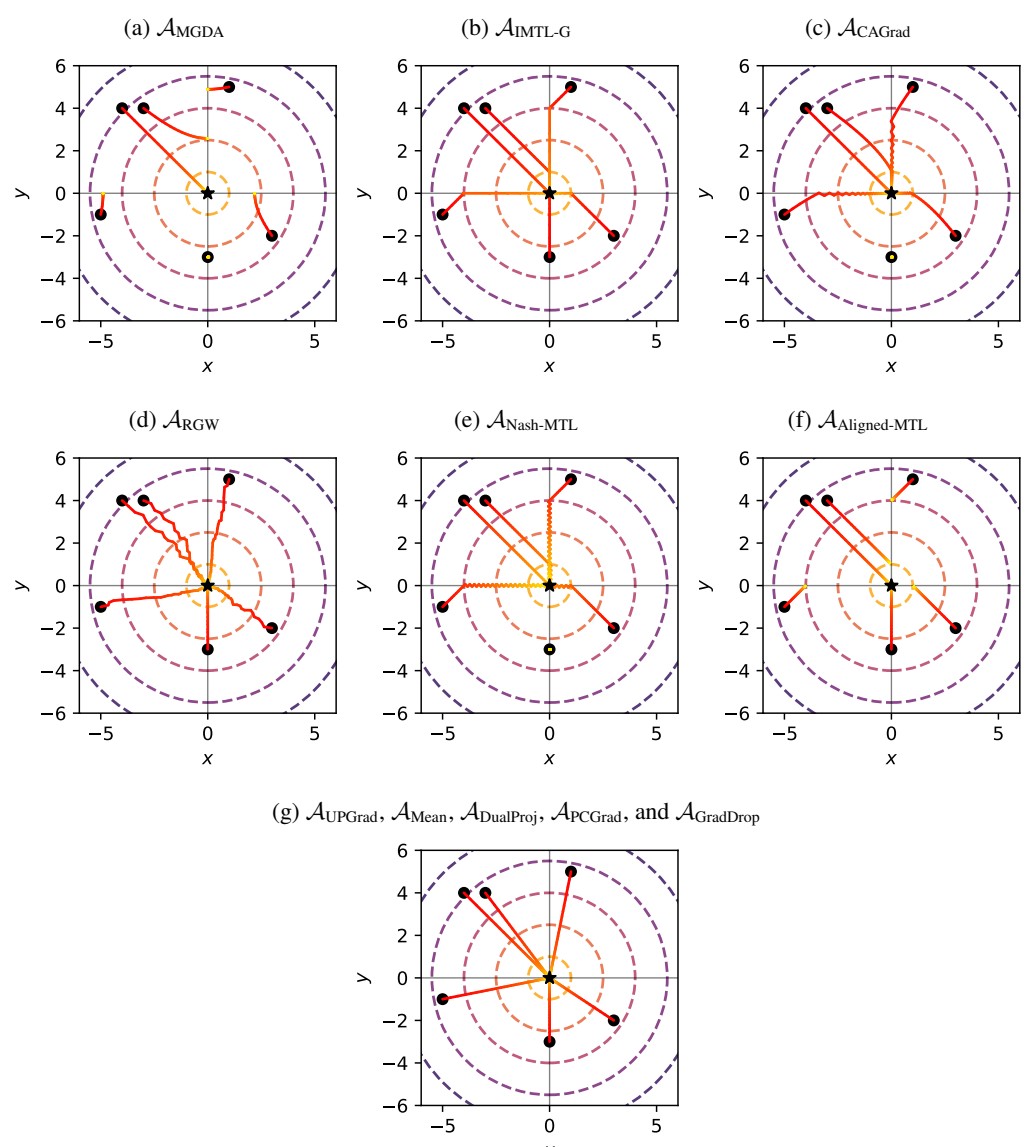

Figure 11: Optimization trajectories in the parameter space for various aggregators when optimizing $\boldsymbol{f} : \begin{bmatrix} x & y \end{bmatrix}^\top \mapsto \begin{bmatrix} x^2 & y^2 \end{bmatrix}^\top$ with JD. Black dots represent initial parameter values, the black star is the Pareto set, the trajectories start in red and evolve towards yellow. The dashed lines are contour lines of the mean objective.

## D.4 OPTIMIZATION TRAJECTORIES

Figure 11 illustrates the optimization trajectories of various aggregators from Table 1 for the function $\boldsymbol{f} : \begin{bmatrix} x & y \end{bmatrix}^\top \mapsto \begin{bmatrix} x^2 & y^2 \end{bmatrix}^\top$, with several initializations. Notably, the Pareto set only contains the origin, while the set of Pareto stationary points is the union of the two axes. The Jacobian of $\boldsymbol{f}$ at $\begin{bmatrix} x & y \end{bmatrix}^\top$ is $\begin{bmatrix} 2x & 0 \\ 0 & 2y \end{bmatrix}$, indicating that the rows do not conflict, which makes this function relatively simple to optimize. Nevertheless, several aggregators, including $\mathcal{A}_{\text{MGDA}}$, $\mathcal{A}_{\text{CAGrad}}$, $\mathcal{A}_{\text{Nash-MTL}}$ and $\mathcal{A}_{\text{Aligned-MTL}}$, fail to converge to the Pareto front for some initializations.

# E  Computation time and memory usage

## E.1  Memory considerations of SSJD for IWRM

The main overhead of SSJD on the IWRM objective comes from having to store the full Jacobian in memory. Remarkably, when we use SGD with ERM, every activation is a tensor whose first dimension is the batch size. Automatic differentiation engines thus have to compute the Jacobian anyway. Since the gradients can be averaged at each layer as soon as they are obtained, the full Jacobian does not have to be stored. In the naive implementation of SSJD, however, storing the full Jacobian costs memory, which given the high parallelization ability of GPUs, increases the computational time. With the Gramian-based method proposed in Section 6, only the Gramian, which is typically small, has to be stored: the memory requirement would then be similar to that of SGD.

## E.2  Time complexity of the unconflicting projection of gradients

Let $J \in \mathbb{R}^{m \times n}$ be the matrix to aggregate. Apart from computing the Gramian $JJ^\top$, applying $\mathcal{A}_{\text{UPGrad}}$ to $J$ requires solving $m$ instances of the quadratic program (5) of Proposition 1. Solvers for such problems of dimension $m$ typically have a computational complexity upper bounded by $\mathcal{O}(m^4)$ or less in recent implementations (e.g. $\mathcal{O}(m^{3.67} \log m)$). This induces a $\mathcal{O}(m^5)$ time complexity for extracting the weights of $\mathcal{A}_{\text{UPGrad}}$. Note that solving these $m$ problems in parallel would reduce this complexity to $\mathcal{O}(m^4)$.

## E.3  Empirical computational times

In Table 7, we compare the computation time of SGD with that of SSJD for all the aggregators that we experimented with. Since we used the same architecture for MNIST, Fashion-MNIST and Kuzushiji-MNIST, we only report the results for one of them. Several factors affect this computation time. First, the batch size affects the number of rows in the Jacobian to aggregate. Increasing the batch size thus requires more GPU memory and the aggregation of a taller matrix. Then, some aggregators, e.g. $\mathcal{A}_{\text{Nash-MTL}}$ and $\mathcal{A}_{\text{MGDA}}$, seem to greatly increase the run time. When the aggregation is the bottleneck, a faster implementation will be necessary to make them usable in practice. Lastly, the current implementation of JD in our library is still fairly inefficient in terms of memory management, which in turn limits how well the GPU can parallelize. Also, our implementation of $\mathcal{A}_{\text{UPGrad}}$ does not solve the $m$ quadratic programs in parallel. Therefore, these results just give a rough indication of the current computation times.

Table 7: Time required in seconds for one epoch of training on the ERM objective with SGD and on the IWRM objective with SSJD and different aggregators, on an NVIDIA L4 GPU. The batch size is always 32.

| Objective | Method | SVHN | CIFAR-10 | EuroSAT | MNIST |
|---|---|---|---|---|---|
| ERM | SGD | 0.79 | 0.50 | 0.81 | 0.47 |
| IWRM | SSJD–$\mathcal{A}_{\text{Mean}}$ | 1.41 | 1.76 | 2.93 | 1.64 |
| IWRM | SSJD–$\mathcal{A}_{\text{MGDA}}$ | 5.50 | 5.22 | 6.91 | 5.22 |
| IWRM | SSJD–$\mathcal{A}_{\text{DualProj}}$ | 1.51 | 1.88 | 3.02 | 1.76 |
| IWRM | SSJD–$\mathcal{A}_{\text{PCGrad}}$ | 2.78 | 3.13 | 4.18 | 3.01 |
| IWRM | SSJD–$\mathcal{A}_{\text{GradDrop}}$ | 1.57 | 1.90 | 3.06 | 1.78 |
| IWRM | SSJD–$\mathcal{A}_{\text{IMTL-G}}$ | 1.48 | 1.79 | 2.94 | 1.69 |
| IWRM | SSJD–$\mathcal{A}_{\text{CAGrad}}$ | 1.93 | 2.26 | 3.42 | 2.17 |
| IWRM | SSJD–$\mathcal{A}_{\text{RGW}}$ | 1.42 | 1.76 | 2.89 | 1.73 |
| IWRM | SSJD–$\mathcal{A}_{\text{Nash-MTL}}$ | 7.88 | 8.12 | 9.33 | 7.91 |
| IWRM | SSJD–$\mathcal{A}_{\text{Aligned-MTL}}$ | 1.53 | 1.98 | 2.97 | 1.71 |
| IWRM | SSJD–$\mathcal{A}_{\text{UPGrad}}$ | 1.80 | 2.01 | 3.21 | 1.90 |

