# OpenReview forum: "Jacobian Descent for Multi-Objective Optimization"
_ICLR.cc/2025/Conference — ICLR 2025 Conference Withdrawn Submission_

### Official Review · Reviewer_G6p6 · 2024-10-21

**Soundness:** 2
**Presentation:** 1
**Contribution:** 1
**Rating:** 5
**Confidence:** 3

**Summary:**

This paper first proposes using the Jacobian matrix to analyze the multi-objective algorithms and provides convergence guarantees. A new method called AUPGrad was proposed, which can project and resolve conflicts while preserving the relative influence of each target gradient (In fact, I am very skeptical about this). It also proposed reducing the time and memory complexity of the algorithm by using the Gramian of the Jacobian matrix.

**Strengths:**

A new algorithm has been proposed to reduce complexity using the Gramian of the Jacobian matrix

**Weaknesses:**

1. First of all, the writing of this paper is disastrous due to the lack of necessary introduction to multi-objective algorithms. For those unfamiliar with multi-objective algorithms, especially Pareto optimality, it is really difficult to understand this paper. The Pareto frontier, as an important concept in this paper, is not introduced at all. Additionally, there is a significant lack of explanation for various details. For example, in Figure 1, we are dealing with a multi-objective optimization problem, but why does Figure 1 only consider two objectives? (There are many such issues!)

2. Regarding the claim in the abstract about "completely resolving conflicts," I have serious doubts. As far as I know, when an algorithm reaches the Pareto frontier, no direction can make the objective function decrease simultaneously. If this algorithm resolves conflicts, how does it converge to the Pareto frontier?

3. In Figure 1, we observe that MGDA (Multi-Gradient Descent Algorithm) is also in the dual cone. I find it hard to understand what distinct advantages this paper has over MGDA. (It’s also hard to see some advantage of this paper compared to the SMG[1] algorithm.)

4. In fact, projection-based multi-objective algorithms have been researched in recent years (e.g., MoCo[2]), but this paper does not show any advantage over these works, nor does it compare with them.

Given these issues, I highly doubt the contribution of this paper. Studying algorithms from the perspective of the Jacobian matrix's properties seems like an interesting direction, but the writing problems and lack of novelty in this paper make it difficult for me to remain positive.

[1]Liu, Suyun, and Luis Nunes Vicente. "The stochastic multi-gradient algorithm for multi-objective optimization and its application to supervised machine learning." Annals of Operations Research 339.3 (2024): 1119-1148.
[2]Fernando, Heshan Devaka, et al. "Mitigating Gradient Bias in Multi-objective Learning: A Provably Convergent Approach." The Eleventh International Conference on Learning Representations.

**Questions:**

The presentation of this paper needs significant revisions. Additionally, the article lacks innovation and requires a stronger comparison with similar algorithms.

---

> ### Author Response · Authors · 2024-11-16
>
> Thank you for your review. We are sorry to see that our work was not to your liking. In the following, we hope to address most of your concerns:
>
> 1. As convergence to the Pareto front is our most remarkable theoretical contribution, we have indeed introduced it in Section 2.4. To better reflect its content, we have renamed this section to "Convergence to the Pareto front".
>
>     Regarding Figure 1, we have limited our representation to the simple case of a problem with two objectives to provide intuition while preserving clarity.
>
>     If there is any issue on the presentation of our paper that you would like to discuss, please let us know.
>
> 2. To clarify, our use of the term "resolving conflict" does not imply that our method aligns future gradients. Rather, it follows from our Definition 1 (Non-conflicting property): our method ensures that updates have a non-positive inner product with all considered gradients.
>
> 3. In Section 5, we discuss specifically about MGDA's issues compared to our method, both theoretically and empirically.
>
>     Regarding comparison with the SMG algorithm, it's important to notice that in the absence of constraint, SMG combines noisy gradients by solving the minimization problem defined in their equation 10, which is equivalent to the first maximization problem of Section 7 of [3]. We show in Appendix B.2. that this is equivalent to the MGDA aggregation rule. Our theoretical and empirical analyses of MGDA thus also apply to SMG.
>
>     As mentioned in part 3 of our global comment, we have added a discussion about trajectories of various methods from the literature on the function $[x, y]^\top \mapsto [x^2, y^2]^\top$.
>     This highlights the significant gap between Pareto optimality and stationarity, and demonstrates the limitations of MGDA and methods that are based on it.
>
>     [3] Jörg Fliege and Benar Fux Svaiter. Steepest descent methods for multicriteria optimization. Mathematical Methods of Operations Research, 2000.
>
> 4. As stated in Section 4, we deliberately do not compare our method with MoCo, because this algorithm is stateful: the direction of the update depends not only on the current Jacobian matrix but also on past information. We think that this is a very interesting direction of research, but it goes beyond our formulation of Jacobian descent that we wanted to analyse rigorously and in isolation from additional complexity.

---

> ### Comment · Reviewer_G6p6 · 2024-11-19
>
> Thank you for your response. Unfortunately, it did not address my concerns. I would like to explain in detail why I must maintain my current score.
>
> 1.**Regarding Section 2.4**: The authors emphasize that their main contribution lies in converging to the Pareto front. However, the discussion of existing literature remains insufficient. For instance, under convex conditions, the Pareto stationary point to which MGDA converges is equivalent to weak Pareto optimality, whereas under strong convex conditions, it corresponds to Pareto optimality. This article only considers convex conditions yet continues to use Pareto stationarity for comparison instead of weak Pareto optimality. This choice makes it challenging for readers to fully appreciate the contribution of the work. As a case in point, the function $[x, y]^{\top} \mapsto\left[x^2, y^2\right]^{\top}$ provided in the authors’ global rebuttal exemplifies MGDA converging to a weak Pareto optimal point.[1,Lemma2.2]
>
> 2.**On "fully resolve conflict"**: I understand and agree with the definition of "no conflict". However, updating along the 'no conflict' direction does not mean resolving conflicts. I would recommend adopting phrasing such as "completely avoiding conflicts" instead of "completely resolving conflicts," as it would better align with the content and findings of this article.
>
> 3.**On comparisons with other algorithms**:
> As mentioned in my first point, this paper improves MGDA by enhancing its convergence from a Pareto suboptimal point to a Pareto optimal point. But, when compared to CAGrad, the primary difference lies in switching the order of operations—changing from "summation first, then project" to "project first, then summation." While this improvement is valid, it seems relatively straightforward and expected, as UPGrad inherently performs projection for each objective, naturally avoiding suboptimality. Thus, the novelty in this aspect appears limited. Furthermore, the lack of investigation into non-convex and stochastic cases raises concerns about the comprehensiveness of the study. The title of the paper implies a broader exploration of Pareto optimization, but these critical scenarios remain unexplored, which diminishes the scope and impact of the work.
>
> To conclude, I am interesting in the research on the convergence domain of Pareto optimization, and this article does present new ideas for this problem. I believe the proposed approach can achieve promising results, particularly in scenarios with relatively few objectives, and it holds potential for broad applicability.  **However, this work focuses solely on convex conditions, improving the convergence point from Pareto suboptimality to Pareto optimality. It does not extend the exploration to the more general and challenging non-convex settings, where investigating whether Pareto stationarity can be elevated to Pareto optimality would significantly enhance the impact and novelty of the study.** Given these limitations, I must maintain my current score regarding the novelty and presentation of the work. That said, if the authors can extend their research to non-convex conditions, I would be pleased to provide a more positive evaluation.\
> [1] Hiroki Tanabe, Ellen H. Fukuda, and Nobuo Yamashita. Proximal gradient methods for multi-objective optimization and their applications. Computational Optimization and Applications, 72(2):339–361, 2019.

---

> > ### Author Response · Authors · 2024-11-19
> >
> > We appreciate the time you've taken to review our paper, and we're happy to address the points you've raised. We'll respond to each of your concerns.
> >
> > 1. We acknowledge that in the convex case, Pareto stationarity and weak Pareto optimality are indeed equivalent. Therefore, on our example function $[x^2, y^2]^\top$, you are correct that MGDA does converge to a weak Pareto optimal point.
> >
> >     The argument remains: the set of weak Pareto optimal points is the union of the two axes (i.e. $x = 0$ **or** $y = 0$). Clearly, those points can be arbitrarily far from being Pareto optimal. Convergence to a Pareto optimal point (i.e. $x=0$ **and** $y=0$) is thus evidently a much stronger result.
> >
> >     We believe that this example alone is sufficient to challenge methods based on MGDA and Pareto stationarity (or weak Pareto optimality), and demonstrates the significance of our contribution.
> >
> > 2. We appreciate your suggestion and have taken it into consideration. As a result, we have revised the abstract to replace the word 'resolve' with 'avoid'.
> >
> > 3. **Novelty**. We would like to clarify that our work does not aim to improve the MGDA method, but rather introduces a novel aggregator that differs fundamentally from existing approaches. Additionally, we note that your comment appears to refer to the DualProj method, rather than CAGrad (as highlighted in Table 1, CAGrad does not even satisfy the non-conflicting property). Please note that, as explained in Section 4, we are the first to introduce DualProj, which is an adaptation of the work from [1], to the multi-objective optimization setting.
> > Still, as you correctly noted, our aggregator UPGrad is typically preferable over DualProj.
> >
> >     **Non-convex setting.**
> >     Note that in the non-convex setting, even in single-objective optimization with GD and under smoothness conditions, one can only prove the convergence of $\frac{1}{T}\sum_{t=1}^T \big\| \nabla f(x_t)\big\|^2$ to $0$, which does not guarantee the convergence of $x_t$ itself to a local minimizer of $f$. A classical example of this behavior is $f(x) = \frac{1}{1 + x^2}$. Because of this, in the multi-objective setting, we cannot hope to achieve convergence to an (even weak) Pareto optimal point.
> >
> >     That being said, a common approach in the MOO literature is to establish a bound of the type:
> >     $$
> >     \frac{1}{T}\sum_{t=1}^T \big\| J_t^\top w_t \big\|^2 \leq \frac{C}{T}
> >     $$
> >     where $C$ is some positive constant, $J_t$ is the Jacobian of $f$ at $x_t$ and $w_t$ is some non-negative weights vector. In our paper, we derive a similar result, as stated in Equation 17 (which does not rely on convexity):
> >     $$
> >     \frac{1}{T}\sum_{t=1}^T \big\| J_t^\top w_t \big\|^2 \leq \frac{2\beta\sqrt{m}}{T}\mathbf 1^\top \big( f(x_1)-f(x^\ast) \big)
> >     $$
> >
> >     Based on this equation, we have added a discussion of convergence in the non-convex case in Appendix A, after the proof of Theorem 1. We thank you for this suggestion, and we hope that in light of this improvement, you could reassess your evaluation.
> >
> >     [1] Gradient episodic memory for continual learning. In Advances in Neural Information Processing Systems, 2017.

---

> > > ### Author Response · Authors · 2024-11-24
> > >
> > > In light of our detailed responses, we believe we have provided sufficient clarification and addressed all of the issues you raised. We respectfully request that you reconsider your score in accordance with the updated information. If you have any major concern, we would appreciate the opportunity to discuss it further and provide additional clarification.

---

> > > ### Comment · Reviewer_G6p6 · 2024-11-25
> > >
> > > Thank you for your response. While your explanation is reasonable(This is an interesting theoretical point, but as far as I know, it rarely arises in practical applications and is not central to my question), I believe my question remains unresolved. \
> > > **My core question is whether the UPGrad algorithm, in the non-convex setting, can achieve convergence to a set that is tighter than Pareto stationarity.** For example, can it converge to a set that contains the Pareto front but is itself contained within the set of Pareto stationary points? （I am not sure about the answer to this question, but I suggest the authors include a similar discussion in the paper.）Demonstrating this would effectively showcase the potential of UPGard in more machine-learning problems.\
> > > ﻿
> > > Conversely, if your results only provide a convergence bound for $ \frac {1}{T}\sum_{t=1}^T \big| J_t^\top w_t \big|^2$ in the non-convex case, it would suggest that the algorithm primarily improves weak Pareto optimality to Pareto optimality without further enhancing Pareto stationarity, which seems to suggest that such an improvement appears difficult to generalize to broader scenarios. It significantly limits the contribution of this work.

---

> > > > ### Author Response · Authors · 2024-11-25
> > > >
> > > > Please keep in mind that the theoretical guarantees of $\mathcal A_{UPGrad}$ are only a small part of our contributions to the field.
> > > >
> > > > Regarding your question, one typically considers that convergence of $J_t^\top w_t$ to $0$ suggests convergence to a point where there exists $0\preceq w$ with $0\neq w$ such that $J^\top w=0$, i.e., a Pareto stationary point.
> > > > In our case, as you correctly noted, we have convergence of $J_t^\top w_t$ to $0$, with $\frac{1}{n}\mathbf 1\preceq w_t$. In the same spirit, this suggests convergence to a point such that there exists $w$ with $0\prec w$ such that $J^\top w=0$, sometimes called a **strong Pareto stationary point**.
> > > > This is indeed a much stronger result in the non-convex setting. We thank you for your constructive suggestion.
> > > >
> > > > We hope this is enough evidence for you to reconsider your rejection.

---

### Official Review · Reviewer_MDCi · 2024-10-27

**Soundness:** 2
**Presentation:** 2
**Contribution:** 2
**Rating:** 5
**Confidence:** 3

**Summary:**

The paper introduces the Jacobian Descent (JD) algorithm, an extension of gradient descent designed for multi-objective optimization. Unlike traditional scalarized multi-objective methods, JD optimizes vector-valued objective functions by updating model parameters with the Jacobian matrix.  Additionally, the paper introduces Instance-wise Risk Minimization (IWRM), a new optimization paradigm that considers each training sample as an independent objective. This allows JD to optimize at an instance level, yielding promising results in empirical evaluations on image classification tasks.

**Strengths:**

Innovative Contribution: The JD method presents a clear and novel perspective to gradient descent for multi-objective optimization, an area of significant relevance to fields like multi-task learning and adversarial training.

Applicability to Emerging Areas: By addressing applications such as federated learning and distributed optimization, the paper suggests impactful future directions for JD and AUPGrad beyond standard optimization problems.

**Weaknesses:**

1. Not Very Well-Structured: The organization of the content in this article is  inconsiderate. For example, in the beginning of Section 2, the definition of the relationship between vectors u and v is actually related to the Pareto optimal condition. It would have been more appropriate to clearly introduce the definitions related to Pareto before presenting the definitions used in this paper. However, the paper  provides the Pareto optimal condition for the first time in Section 2.4. If the reader is not familiar with the multi-objective problem, they may struggle to understand the author's intent while reading the paper.


2.Limited Comparison to State-of-the-Art: Theoretically,  The paper didn't provide a specific convergence rate for JD  and avoided comparing it with the convergence rates of other gradient-based multi-objective optimization algorithms. This omission may limit the reader's ability to assess the efficiency and competitiveness of the proposed method against existing approaches in the field.


3.Scalability Concerns: While the proposed Gramian-based approach aims to improve efficiency, the JD method remains computationally expensive for large-scale problems, which might hinder practical applications, especially in high-dimensional or real-time scenarios.

**Questions:**

1) This paper seems to deliberately avoid comparison and discussion with multi-objective algorithms based on Pareto stationary. However, in reality, whether it is the definition of 'non-conflicting' or the design of the AUPGrad algorithm, the paper merely shifts the perspective for solving the problem to the Jacobian matrix. The core idea is not fundamentally different from previous algorithms. Could you provide a detailed discussion on the differences between this algorithm and MGDA, as well as its stochastic variants?

2)  Theoretically, since this algorithm does not provide specific convergence results, how can we evaluate the performance of this algorithm compared to other multi-objective optimization algorithms?

3) This paper lacks visualizations of the algorithm's convergence trajectories to the Pareto front. Visualizing the optimization paths of multiple initial points is also a crucial aspect of presenting the optimization effects of multi-objective algorithms from an experimental perspective.

---

> ### Author Response · Authors · 2024-11-16
>
> We appreciate the time and effort you dedicated to reviewing our work, and we are pleased that your feedback has led to significant improvements of our paper. We will now address your concerns.
>
> ---
>
> ### Weaknesses
> 1. Since the definition of the Pareto concepts (optimality, set, front) depends on the selection of the partial order, we still think that these concepts should be defined after defining the relations ($\prec$, $\preceq$, etc.) in the beginning of Section 2. Because we mainly use those concepts in Section 2.4, we prefer to keep the definitions there.
> Nevertheless, we thank you for raising this concern and we agree that unfamiliar readers would benefit from an earlier introduction to those concepts. We have revised our introduction to provide some intuition about the Pareto front as early as the first paragraph. The added sentences are:
> "In opposition, multi-objective optimization methods typically attempt to optimize all objectives simultaneously, without making arbitrary compromises: the goal is to find points for which no improvement can be made on some objectives without degrading others."
>
> 2. Please refer to part 1 of our global comment for the discussion about the convergence rate. Regarding comparison to the state of the art, we have provided a proof of convergence to the Pareto front, which is a stronger result than the convergence proofs of the other methods from Table 1. Note that we have added evidence that some of these methods do not even converge to the Pareto front when the function to optimize is $[x, y]^\top \mapsto [x^2, y^2]^\top$ (see part 3 of our global comment).  We have not analyzed nor experimented against some of the recent methods for stochastic MOO, such as CR-MOGM[1], MoCo[2] and MoDo[3], because these methods are stateful: the update direction also depends on the "accelerated" Jacobian for MoCo, and on the previous weights for CR-MOGM, MoCo and MoDo. Clearly, preserving some state over the iterations has the potential to improve the convergence guarantees in some settings (similarly as momentum-based SGD can improve the convergence rate compared to SGD).
> Generalizing Jacobian descent to support stateful aggregators is a promising future direction that would deserve further attention.
>
>     [1] Zhou et. al. On the convergence of stochastic multi-objective gradient manipulation and beyond.
>
>     [2] Fernando et. al. Mitigating gradient bias in multi-objective learning: A provably convergent approach.
>
>     [3] Chen et al. Three-way trade-off in multi-objective learning: Optimization, generalization and conflict-avoidance.
>
> 3. We agree with this weakness. That being said, the goal of our paper is mostly to provide a solid theoretical framework for descent methods in multi-objective optimization. Our methods could be fairly easily adapted to be much more practical for real-world applications. For instance, we could alternate between a step of Jacobian descent and several steps of gradient descent, on the scalarization of the objective using the weights that the Jacobian descent step has used to combine the gradients. When the number of objectives is large, we could also group them into a smaller number of objectives depending on their average gradient similarity. Finally, we could use JD for a subset of the parameters and GD for the rest (See e.g. MGDA-UB [4]). These ideas are not really new, so we did not mention them in our article, but they would easily enhance the practicality of all methods that are expressable as Jacobian descent.
>
>     [4] Sener et. al. Multi-task learning as multi-objective optimization.
>
> ---
>
> ### Questions
>
> 1. We do compare ourselves with 9 methods from the literature, both theoretically (Appendix B) and empirically (Section 5 and Appendix D). We also have a discussion about the drawbacks of MGDA in Section 5, and more specifically about the insufficiency of a proof of convergence to a Pareto stationary point in Section 2.4. Please refer to our answer to weakness 2 about why we do not compare against the stochastic variants of MGDA (CR-MOGM, MoCo and MoDo).
>
>     We do not claim that Jacobian descent is an entirely new idea, but we rather propose this framework to unify many existing methods. The novel contributions that our paper provides are the UPGrad aggregator, the IWRM objective, the idea of SSJD and Gramian reverse accumulation algorithm.
>
> 2. Theorem 1 and the following discussion provide important theoretical improvements compared to the existing literature. Please refer to part 1 of our global comment for the discussion about the convergence rate.
>
> 3. Thank you for this constructive comment. We have added a visualization of the optimization trajectory of methods from Table 1 when optimizing $[x, y]^\top \mapsto [x^2, y^2]^\top$. Since this has led to very interesting results, and to the discrimination of some methods, we have decided to add those plots in Appendix D4. Please refer to part 3 of our global comment for more details.

---

> > ### Author Response · Authors · 2024-11-24
> >
> > We appreciate the time you took to review our paper and provide feedback. We believe that we have addressed all of your concerns and significantly improved the paper based on your suggestions. As the discussion period is nearing its end, we would appreciate the opportunity to engage with you further and ensure that all issues have been fully addressed. If you do not have any outstanding concerns or weaknesses, we would be grateful if you could reconsider your score.

---

### Official Review · Reviewer_DmCe · 2024-10-29

**Soundness:** 2
**Presentation:** 3
**Contribution:** 2
**Rating:** 3
**Confidence:** 4

**Summary:**

The authors extend the concept of gradient descent to JD by using the Jacobian matrix of a vector-valued objective function, with each row representing the gradient of a different objective. A key contribution is the AUPGrad (Averaged Unconflicting Projection of Gradients) aggregator, which is proposed to resolve conflicts between gradients in a way that preserves their relative influence.

The paper provides theoretical convergence guarantees for JD with AUPGrad, especially in the context of smooth and convex functions, and demonstrates its effectiveness through experiments on image classification tasks. Additionally, the paper introduces Instance-wise Risk Minimization, a new learning paradigm that treats the loss of each training example as a distinct objective, and shows promising results compared to traditional loss minimization methods.

**Strengths:**

1.	Theory: The paper provides a solid theoretical foundation, including convergence proofs for JD with AUPGrad under certain conditions, which adds rigor to the proposed method.

2.	Reproduction: The source codes are provided, which is helpful to validate the realistic performance.

3.	Detailed introductions to baselines. Appendix B states the properties and computation steps of several compared algorithms, which helps to highlight and better understand the advantages of this work.

**Weaknesses:**

1.	Lack of mathematical backgrounds. It’s kindly suggested to add more detailed introductions to the investigated multi-objective learning and definitions on Pareto fronts.

2.	Theory. The theoretical guarantees provided for JD with AUPGrad rely on assumptions of smoothness and convexity, which may not hold in some practical scenarios, limiting the applicability of the results. More practical examples are required. Could you please further derive the convergence rate? Some computation complexity from the theoretical perspective is also crucial for new algorithms.

3.	Broader baselines and empirical settings.
The experiment setting is limited, while comprehensive benchmarks are required to validate across a variety of problems, especially those with inherently conflicting objectives.

Besides, more empirical scenarios of classical multi-objective learning tasks [1] are also helpful.

[1] Three-way trade-off in multi-objective learning: Optimization, generalization and conflict-avoidance

4.	Moreover, the authors are kindly suggested to adopt some accelerate techniques on approximating the Jacobian counterpart, instead of merely taking averages. As present in Table 7, the proposed A_{UPGrad} consumes more time for training.

**Questions:**

Please see the comments in Weakness.

I would be willing to raise my score if these questions concerned are well addressed.

---

> ### Author Response · Authors · 2024-11-16
>
> We appreciate the time and effort you took to review our work, and we're grateful for your constructive feedback. Your review led to several improvements of the paper. We now address your concerns.
>
> 1. Thank you for this suggestion. We have improved the paper with a brief explanation in the introduction to outline the goal of multi-objective optimization, specifically to decrease every coordinate at each step, and to reach a point where this is no longer possible, i.e., a Pareto optimal point. We hope that this addition will improve clarity, and we welcome any further feedback on the presentation.
>
> 2. We have added a new experiment on a toy example in Appendix D4. See part 3 of our global comment for more details. Regarding the convergence rate, please refer to part 1 of our global comment. We have also added the computational complexity of $\mathcal A_{\text{UPGrad}}$ in Appendix E2.
>
> 3. Thank you for raising this concern. As mentioned above, we have added a new experiment on a toy problem in Appendix D4. Besides, we agree that exploring settings with high conflict (e.g. adversarial training) is a very promising direction.
>
> 4. We are currently developing a framework to improve the running speed of Jacobian Descent by incorporating Gramian-based methods, as outlined in Section 6. This has the potential to significantly reduce memory usage and increase speed, particularly in the context of IWRM. Moreover, this approach could benefit not only $\mathcal A_{\text{UPGrad}}$, but also other aggregators such as $\mathcal A_{\text{MGDA}}$, $\mathcal A_{\text{CAGrad}}$, etc. The full development of this method would deserve a separate paper.
>
>     We would appreciate clarification on your suggestion to use "some accelerate techniques on approximating the Jacobian counterpart, instead of merely taking averages". Could you please provide more details on what you mean by this?

---

> > ### Comment · Reviewer_DmCe · 2024-11-21
> > **Feedback**
> >
> > Thanks for the response. Some of my concerns have been addressed. After reading the responses and the comments from other reviewers, I would rethink the given score, as some key issues still remain:
> >
> > **1) experiments reflecting the commonly discussed Multi-Objective scenarios with conflicting objectives.**
> >
> > **2) the convergence analysis for your new algorithm.** The new convergence guarantees are interesting. Thanks to the public comment, stating its similarity to some existing work [1]. But I have not carefully checked the detailed proof.
> >
> > I would be grateful if the authors could derive some theoretical conclusions which reflect their specific algorithmic settings.
> >
> > **3) the contributions w.r.t. the Pareto stationarity analysis under different convexity conditions.**  I'm also interested in this learning theory.
> >
> > Based on the latest response, the derived bound in this paper depends on the $\beta$, m and the initial point.
> > Could the authors add more discussions with experiments for validation to better enhance the readability?
> >
> > [1] Hwang, Youngsik, and Dong-Young Lim. "Dual Cone Gradient Descent for Training Physics-Informed Neural Networks." NeurIPS 2024.

---

> > > ### Author Response · Authors · 2024-11-22
> > >
> > > We appreciate your engagement in this review process. However, we are surprised to see that you have decreased your score from 6 to 3, based on other reviews. We believe that our responses, including our recent answer to Dongyoung Lim's comment, have satisfactorily addressed the issues raised.
> > >
> > > 1. The claim by Dongyoung Lim that our experimental setting does not have conflict is false (as justified in our answer to their comment). Also, as explained Section 3 of our paper, multitask learning settings often have limited conflict, typically less than our experimental setting.
> > >
> > > 2. Our theoretical convergence results are presented in Theorem 1, with a detailed proof provided in Appendix A2. If you have any concerns about our theorem or its proof, let us know.
> > >
> > >     Regarding the concurrent work of Dongyoung Lim, our work has some limited similarity in the case of two losses ($m=2$). In a more general setting, their approach does not generalize to $m>2$ and the comparison is therefore impossible. Please read our answer to Dongyoung Lim for more details.
> > >
> > > 3. We have previously addressed this question in the non-convex case in our response to G6p6. To summarize, existing methods in the literature cannot be proven to converge to a Pareto stationary point, even in the simpler case of single-objective optimization. Instead, the typical convergence proof only establishes that the gradient norms converge to $0$. We will not make the claim that our method converges to a Pareto stationary point in the non-convex case, because this would just reinforce the ongoing lack of mathematical rigor that exists in the field.
> > >
> > >     In the convex case, our algorithm converges to a Pareto optimal point, which is also a Pareto stationary point. See Theorem 1.
> > >
> > >     Since the strongly-convex case is a special case of the convex case, our algorithm also converges to a Pareto optimal point, which is, again, a Pareto stationary point.
> > >
> > >     In the setting of our Theorem, $\beta$ is the smoothness parameter, it is related to the step-size $\frac{1}{\beta\sqrt{m}}$. The parameter $m$ is the number of objectives. Note that the convergence rate we have is $\frac{\sqrt{m}}{T}$ scaled with a constant that depends only on the initial point. In our experiment we have carefully selected the learning rate (See details in Appendix C1), which is similar to finding the appropriate $\beta$ and we have also tested different batch sizes which correspond to $m$ (See results in Appendix D, Figure 9). Regarding the initial point, our models are initialized at random, following the default initialization of PyTorch (See details in Appendix C3).
> > >
> > > If you have any unaddressed concerns, please let us know.

---

> > > > ### Author Response · Authors · 2024-11-24
> > > >
> > > > We hope our previous comment has effectively clarified your concerns. Since, as you mentioned, you revised your score based on other reviews (including the public comment of Dongyoung Lim), we kindly encourage you to read our detailed responses to them as well.
> > > >
> > > > We are confident that we have thoroughly addressed all major concerns, and we would greatly appreciate it if you could reconsider your rejecting score or share a legitimate justification for it.

---

### Official Review · Reviewer_LBbZ · 2024-11-02

**Soundness:** 3
**Presentation:** 3
**Contribution:** 3
**Rating:** 6
**Confidence:** 3

**Summary:**

In this paper, the authors propose a general framework for multi-objective optimization (MOO), termed Jacobian Descent (JD), where the multi-objective optimization update direction is viewed as the output of an aggregator mapping. This mapping takes the Jacobian of the multi-objective vector and maps it to a vector in the dimension of the parameters to be updated. The authors introduce stochastic implementations of JD, termed Stochastic Jacobian Descent (SJD) and Stochastic Sub-Jacobian Descent (SSJD). The paper then proposes the Unconflicting Projection of Gradients aggregator, denoted as $A_{\text{UPGrad}}$, as a desirable aggregator for MOO updates. This aggregator has properties such as non-conflicting behavior (i.e., the update direction does not conflict with any individual gradient), linear scaling, and weighted combinations (the update direction is a weighted combination of the rows of the Jacobian). Theoretical results are provided for the convergence of MOO updates using $A_{\text{UPGrad}}$ to the Pareto front under certain assumptions. The authors further introduce Instance-wise Risk Minimization (IWRM) as an application of MOO for traditionally single-objective problems, where the loss at each data point is considered as an individual objective. They also discuss how some existing MOO aggregators fit within the JD framework. Experimental results are presented to demonstrate the applicability of $A_{\text{UPGrad}}$ in practical IWRM problems. Additionally, the paper outlines a more efficient implementation of JD, known as Gramian-based JD, to address the computational overhead associated with the proposed JD approach.

**Strengths:**

* The proposed JD framework generalizes many existing MOO approaches, providing insights into the similarities, differences, and behaviors of these approaches.
* The unconflicting projection of gradients aggregator, UPGrad, has provable convergence guarantees and demonstrates faster convergence emperically in practical applications compared to existing MOO methods.
* The paper is fairly easy to follow, the proposed methods are well motivated, and the build-up to the proposed methods are clearly explained.

**Weaknesses:**

* The theoretical convergence guarantees are only asymptotic, so it is unclear how JD compares with standard gradient descent in terms of theoretical convergence rate.
* As the authors acknowledge, the proposed method is computationally intensive. It is also unclear, even with the outlined method for increasing computational efficiency, whether the computational overhead can be reduced (compared to, for example, standard SGD on the ERM objective) due to the need to compute the Jacobian. For example, even with a smaller batch size than that used for the Mean objective, applying JD for IWRM involves separate per-datapoint gradient calculations. On the other hand, applying the Mean aggregator for the ERM problem with a larger batch size might result in faster convergence with lower computational overhead.
* While viewing ERM as an IWRM and adaptively optimizing the objective may result in faster convergence for the empirical loss, it is unclear how this kind of adaptive weight selection could affect test performance (which is the ultimate goal of most ERM applications), since the weights are assigned based on the training dataset.

**Questions:**

* What are the challenges that prevent deriving non-asymptotic convergence guarantees for UPGrad?
* Can the authors provide a runtime comparison for the standard SGD with ERM vs UPGrad applied for IWRM?
* Can the authors comment/provide evidence how reformulating ERM as IWRM affect the test performance of the final model?

---

> ### Author Response · Authors · 2024-11-16
>
> We would like to thank you for the time and effort you invested in reviewing our work. We are pleased to hear that you found our research valuable, and we appreciate your thoughtful feedback. We will now address your concerns.
>
> ---
>
> ### Weaknesses
> 1. Please refer to part 1 of our global comment for the discussion about the convergence rate.
>
> 2. Thank you for raising this point. The main overhead of SSJD on the IWRM objective comes from having to store the full Jacobian in memory. Remarkably, when we use SGD with ERM, every activation is a tensor whose first dimension is the batch size. Automatic differentiation engines thus also have to compute the Jacobian. Since the gradients can be averaged at each layer as soon as they are obtained, the full Jacobian does not have to be stored. In the naive implementation of SSJD, however, storing the full Jacobian costs memory, which, given the high parallelization ability of GPUs, increases the computational time. With the Gramian-based method proposed in Section 6, only the Gramian, which is typically small, has to be stored: the memory requirement would then be similar to that of SGD.
>
>     We have included these considerations in a new subsection of Appendix E, called "Memory considerations of SSJD for IWRM".
>
>     Note that since there is still some computational overhead from the aggregation, our method should be used in the presence of non-negligible conflict.
>
> 3. Please refer to our response to your third question.
>
> ---
>
> ### Questions
> 1. Please refer to part 1 of our global comment for the discussion about the convergence rate.
>
> 2. This empirical runtime comparison can be found in Appendix E, Table 7. To improve clarity, we have revised the table by adding and additional column with the objective (ERM or IWRM), and by improving its caption.
>
> 3. Since our goal was to analyze JD and aggregators in isolation, we have deliberately limited our scope to a pure optimization setting. This removed the complexity of having to select generalization-related hyper-parameters (e.g. dropout, regularization, etc.), which enabled a reproducible, precise and fair learning rate selection strategy, increasing the reliability of our experimental results.
>
>     Still, we can reuse the results of our experiments to easily obtain test accuracies. In the following table, we have reported the test accuracies obtained by each model and on each problem, with the learning rates obtained in Section 5, averaged over the 8 random reruns.
>
> 	| Aggregator  | SVHN     | CIFAR-10   | EuroSAT  | MNIST    | F-MNIST  | K-MNIST  | Average  |
> 	|-------------|----------|------------|----------|----------|----------|----------|----------|
> 	| Mean        | 69.6     | 47.7       | 68.5     | **96.3** | 80.7     | 76.9     | 73.3     |
> 	| UPGrad      | **69.7** | **48.1**   | **73.0** | 96.0     | 80.8     | **77.1** | **74.1** |
> 	| MGDA        | 41.7     | 32.9       | 62.5     | 85.2     | 73.4     | 53.7     | 58.2     |
> 	| Aligned-MTL | 61.3     | 47.3       | 65.3     | 88.3     | 78.5     | 67.9     | 68.1     |
> 	| PCGrad      | 49.1     | **48.1**   | 59.3     | 85.9     | 69.3     | 54.7     | 61.1     |
> 	| DualProj    | 69.3     | 47.2       | 72.7     | 96.1     | 80.8     | 76.4     | 73.7     |
> 	| RGW         | 66.2     | 39.5       | 67.9     | 94.0     | 77.1     | 73.6     | 69.7     |
> 	| GradDrop    | 68.9     | 46.8       | 71.9     | **96.3** | **80.9** | 76.9     | 73.6     |
> 	| CAGrad      | 60.7     | 42.4       | 68.1     | 90.3     | 78.1     | 65.3     | 67.5     |
> 	| IMTL-G      | 58.7     | 44.8       | 65.2     | 87.2     | 75.0     | 65.9     | 66.1     |
> 	| Nash-MTL    | 39.7     | 31.3       | 60.8     | 87.6     | 72.0     | 52.7     | 57.3     |
>
>     Note that SSJD with Mean aggregator on the IWRM objective is equivalent to the standard SGD on the ERM objective.
>
>     This gives a rough evidence that the improved optimization of our method does indeed translate into a better test performance.
>
>     Having established the efficacy of our approach in a controlled setting, our next step will be to evaluate its performance in practice, assessing whether removing conflict in optimization can lead to improved outcomes in real-world scenarios.

---

> > ### Author Response · Authors · 2024-11-24
> >
> > We appreciate your thoughtful review and the recommendations you provided. We hope that our clarifications and revisions have addressed your concerns, and we would appreciate any final feedback you may have before the discussion period closes.
> >
> > If you are satisfied with our updates, we would be grateful for your continued support. We believe that our work has the potential to make a positive impact on the community, and your endorsement would be a valuable contribution to its success.

---

### Author Response · Authors · 2024-11-16

### 1. Convergence rate
We appreciate the reviewers' comments regarding the lack of analysis of the rate of convergence. In Theorem 1, we establish convergence to the Pareto front, rather than to the set of Pareto stationary points. The gap between these two concepts is discussed in part 3 below. In particular, MGDA fails to converge to the Pareto front in general. Therefore, comparing the convergence rate of our method with that of MGDA is not meaningful.

Nevertheless, we acknowledge the importance of considering convergence rates, and to address this, we have revised Section 2.4 and Appendix A.3 to include the following bound (which follows directly from our proof of Theorem 1):
$$\frac{1}{T}\sum_{t\in[T]} w_t^\top\big(f(x_t) - f(x^\ast)\big) \leq \frac{\sqrt{m}}{T} \left(\big\| f(x_1)-f(x^\ast) \big\| + \frac{\beta}{2}\| x_1-x^\ast \|^2\right)$$

As every coordinate of $w_t$ is lower-bounded by $1$, this suggests a convergence rate for $f(x_t)$ to $f(x^\ast)$ of order $O(1/T)$, aligning with the typical rate of gradient descent without acceleration.

---

### 2. Computational efficiency
Several reviewers have raised concerns about computation time. We have thus added considerations about memory usage and computation time as new subsections of Appendix E.

---

### 3. Trajectories for aggregators
In response to MDCi's suggestion, we have added optimization trajectories for various methods from the literature to further illustrate the performance of different aggregators.

We consider the function $[x, y]^\top \mapsto [x^2, y^2]^\top$, which has independent gradients and is therefore relatively easy to optimize. However, despite this simplicity, several aggregators, including MGDA, Nash-MTL, CAGrad, and Aligned-MTL, fail to converge to the Pareto front $\{ (0,0) \}$ for some initializations. These trajectories converge to the set of Pareto stationary points (the union of the two axes), which is coherent with the analysis provided by these works.

This highlights the importance of our Theorem 1, which establishes convergence to the Pareto front rather than to a Pareto stationary point. To the best of our knowledge, $\mathcal A_{\text{UPGrad}}$ is the only non-conflicting aggregator that guarantees this property.

---

> ### Author Response · Authors · 2024-11-30
>
> ### 4. Non-convex analysis
> Some reviewers raised concerns about the non-convex analysis. We thank them as it led to an improvement of the paper.
>
> Specifically, in the smooth (and possibly non-convex) case, one typically considers that convergence of $J_t^\top w_t$ to $0$ suggests convergence to a point where there exists $0\preceq w$ with $0\neq w$ such that $J^\top w=0$, i.e., a Pareto stationary point.
> We have updated the paper with the following inequality, which can be directly derived from the proof of Theorem 1:
> $$
> \frac{1}{T}\sum_{t=1}^T \big\| J_t^\top w_t \big\|^2 \leq \frac{2\beta\sqrt{m}}{T}\mathbf 1^\top \big( f(x_1)-f(x^\ast) \big)
> $$
> We therefore have convergence of $J_t^\top w_t$ to $0$.
> Remarkably, as $\frac{1}{n}\mathbf 1\preceq w_t$, this suggests convergence to a point such that there exists $w$ with $0\prec w$ such that $J^\top w=0$. Such a point is sometimes refered to as a **strong Pareto stationary point**.
>
> The gap between those two types of convergence can again be seen on our example function $[x,y]^\top\mapsto[x^2,y^2]^\top$, as only the origin is a strong Pareto stationary point, but the set of Pareto stationary points is the union of the two axes. Of course this gap remains in non-convex settings.
> We thus believe that algorithms expressable as Jacobian descent should aim for this stronger notion of convergence in the non-convex settings.

---

### Public Comment · ~Dongyoung_Lim1 · 2024-11-21
**Public comment**

Thank you very much for your interesting work! I found that this work has a significant theoretical resemblance to the paper [1], which was published at NeurIPS 2024. The concept of resolving conflicts in multi-objective optimization by leveraging properties of dual cones seems to overlap considerably between the two approaches.

It would help readers to clearly understand how the authors' work extends or differs from the approach of [1]. In addition, I would like to leave a few comments:

1. **Convergence Results**

In the global rebuttal, the authors state that "In Theorem 1, we establish convergence to the Pareto front, rather than to the set of Pareto stationary points." However, this answer remains somewhat ambiguous. Theorem 1 is limited to the convex setting, which, while valuable, does not reflect the more practical non-convex problems in real-world applications. In such cases, convergence to Pareto stationary points is often more meaningful. More importantly, the result of convergence to the Pareto frontier in a convex setting cannot be claimed to be superior to the convergence to Pareto stationary points in a non-convex setting.

2. **Experimental Design**

I believe the current experiments do not effectively highlight the advantages of multi-objective optimization. It is difficult to justify that classification losses on different datasets represent conflicting objectives. Furthermore, the current experimental results, summarized in Figure 2, do not provide sufficient insight into the superiority of different algorithms. It would be more appropriate to utilize proper evaluation metrics specific to multi-objective optimization to conduct a more rigorous and meaningful comparison.

Lastly, I understand that [1] and the current paper are independently developed concurrent works. Therefore, I believe the existence of [1], which is already published, should not negatively impact the evaluation of the authors' paper.

[1] Hwang, Youngsik, and Dong-Young Lim. "Dual Cone Gradient Descent for Training Physics-Informed Neural Networks." NeurIPS 2024.

---

> ### Author Response · Authors · 2024-11-22
>
> Thank you for your interest in our research and for sharing your concerns. We appreciate the opportunity to engage in a discussion about our work.
>
> We have read your paper and acknowledge that your method shares some similarities with ours, particularly in the consideration of the dual cone concept from convex analysis. We agree that this notion holds great importance for multi-objective optimization (MOO).
> However, we believe that your method may face challenges when generalizing to higher dimensions (it is limited to 2 objectives). Specifically, the projection of the average onto the orthogonal hyper space of a gradient may not lie within the dual cone in general. This could limit the applicability of your approach and in particular, as we consider many simulatenous objectives, we cannot use your method on our problems.
>
> **Non-convex setting**
>
> Regarding convergence in the non-convex case, note that if $\frac{1}{T} \sum_{t=1}^T\| \nabla\mathcal L(x_t)\|^2$ converges to $0$, this does not imply convergence to a Pareto stationary point or even convergence at all. For example, consider the function $\frac{1}{1+x^2}$, on which convergence of the gradient to $0$ implies divergence of $x_t$ (this example can easily be extended to any number of objectives).
> Nevertheless, we have also added a similar result to our paper, as it was a popular request from our reviewers. See our answer to reviewer G6p6.
>
> We also agree that convexity can make the problem easier, but we argue that failing to converge to a Pareto optimal point in the convex case is a significant limitation for a given MOO method. Our method, on the other hand, is designed to address this challenge.
>
> **Experimental setting**
>
> As indicated in our conclusion, working with datasets with very conflicting objectives is a very promising future direction.
> However, your claim that the objectives that we consider are not conflicting is false. This can be seen by the plots showing the cosine similarity between the update performed by UPGrad and the update performed by SGD. If the individual gradients were not conflicting, they would all be in the dual cone even without projection, and we would always observe a similarity of 1 between UPGrad and SGD. This is clearly not the case here.
>
> The results from Figure 2 show a clear advantage of our method over all over methods, that is easily reproducible accross all datasets that we have experimented with. In opposition, many multi-objective optimization methods from the literature fail drastically on this simple setting. This is a very interesting result, once again adding empirical evidence to our theoretical claims.

---

> ### Public Comment · ~Dongyoung_Lim1 · 2024-11-23
> **Public comment**
>
> Thank you for your response. I would like to make a few additional comments regarding your response.
>
> **1. Dual Cone and DCGD**
>
> In [1], DCGD focuses on the problem of PINNs where two loss functions are considered. However, the **DCGD framework** is applicable to high-dimensional settings. It is true that **specific** DCGD algorithms may not guarantee that the updated gradient lies within the dual cone in more than two losses, but this is a concern related to a specific DCGD algorithm, not the general DCGD framework itself. As we are reviewing your paper, I will not provide further detailed explanations about [1] here.
>
> **2. Non-Convex Setting and Misleading Example**
>
> I am afraid that your response regarding the standard argument in the nonconvex literature about $1/T\sum_{t=1}^T |\nabla \mathcal L(x_t)|^2 \rightarrow 0$, using $\frac{1}{1+x^2}$ as an example, could be highly misleading and deserves clarification. Your example of $\frac{1}{1+x^2}$ represents a pathological case that can be easily excluded by incorporatindg conditions such as coercvity or dissipativie conditions [2,3], which ensure that staionary points lie within the interior orf the domain. Therefore, $\frac{1}{1+x^2}$ does not invalidate the standard aurgument in non-convex literature, and it is not an appropriate example to challenge the convergence metric.
>
> **3. Insufficient literature review**
>
> If the focus of your work is on convergence in convex settings, then the comparison with multi-task learning (MTL) optimization methods may not be entirely appropriate. A thorough literature review of other references from optimization fields that consider similar settings is necessary. For example, see [4].
>
> **4. Experiment setup**
>
> Your explanation regarding the experimental setting remains insufficient. Demonstrating the superiority of your algorithm over others solely based on the training loss curve is overly optimistic and lacks rigor. To compare with multi-task learning algorithms, extensive experiments on standard multi-task learning datasets are necessary in terms of proper evaluation metrics. Alternatively, experiments on convex objectives that align with your theoretical setting and comparisons with benchmark algorithms in that setting would be more appropriate.
>
> Best of luck with your review process, and I hope your interesting paper achieves positive outcomes.
>
>
> [1] Hwang, Youngsik, and Dong-Young Lim. "Dual Cone Gradient Descent for Training Physics-Informed Neural Networks." NeurIPS 2024.
>
> [2] Hazan, Elad. "Lecture notes: Optimization for machine learning." arXiv preprint arXiv:1909.03550 (2019).
>
> [3] Xu, Pan, et al. "Global convergence of Langevin dynamics based algorithms for nonconvex optimization." Advances in Neural Information Processing Systems 31 (2018).
>
> [4] Nie, Jiawang, and Zi Yang. "The multi-objective polynomial optimization." Mathematics of Operations Research (2024).

---

> > ### Author Response · Authors · 2024-11-23
> >
> > Thank you for your response. We will address your additional comments.
> >
> > 1. Thank you for this clarification. The core idea behind what you call DCGD is that the negative of the update vector lies within the Dual Cone (hence the name). Since your method does not guarantee this property — which we term "Non-conflicting" in our paper — in higher dimensions, we think that its name is quite misleading. We appreciate this discussion and, since it is unrelated to our paper, we would be happy to continue it somewhere else.
> >
> > 2. Among the two papers that you just provided as reference for coercitive and dissipative conditions, [2] does not even contain any of those terms in its entire text and [3] only talks about the dissipative condition (Assumption 3.2). We kindly ask you to include at least one reference about the coercitive conditions if you wish to continue discussing on this topic.
> >     Those conditions are not usual assumptions in our domain. In fact, none of the papers that we cite in our work assumes any of those two conditions. Your paper [1] does not seem to assume any of them neither.
> >     All we are saying is that convergence of the gradients to zero does not imply convergence of the iterates ($x_t$) even under standard assumptions. This fact implies that in the non-convex setting and under standard assumptions, algorithms are not proved to make the iterates converge to a Pareto stationary point (or to a stationary point in the case of single-objective optimization). Only their gradients may converge to zero.
> >     We therefore have a disagreement on the way to present our results. You seem to prefer an overly optimistic formulation that is not mathematically rigorous, while we are firmly opposed to that. We therefore claim in Equation 20 (Appendix A3) a result of convergence of our gradients in the non-convex setting rather than a result of convergence for our iterates. We will not change this unless we are proven to be wrong.
> >
> > 3. The focus of our work is very far from being solely on convergence guarantees in the convex setting. In fact, this is only the focus of Section 2.4. [4] is an interesting reference but it focuses on scalarizations (quite arbitrarily turning the multi-objective problems into single-objective) and on the optimization of polynomial functions. This is clearly not applicable when optimizing the parameters of neural networks, as in our main experiments.
> > If you have other references that we may have missed, please share them with us.
> >
> >
> > 4. Since we're purely on an optimization problem, the training loss evolution is the most appropriate metric. As reviewer LBbZ had a similar concern to yours regarding generalization, we have answered to him with a table showing the performance on the test set obtained with the various methods. It shows $\mathcal A_{\text{UPGrad}}$ ahead of all considered alternatives.
> >
> >     When it comes to optimizing neural network parameters, most of the existing methods come from the multitask learning literature. It doesn't mean that multitask learning is the most appropriate use-case for multi-objective optimization. As stated in section 3, MTL standard benchmarks often do not contain enough conflict to justify the usage of methods based on the Jacobian. We compare against these algorithms because they are special cases of Jacobian descent and because they are obviously not limited to multitask learning.
> >
> >     We now do have an experiment in the convex setting showing the optimization trajectories of various methods, in Appendix D4. Nevertheless, we are developping a general algorithm for the optimization of differentiable vector-valued functions. It turns out that our algorithm has better convergence guarantees in the convex case than most existing methods. This should not be a reason to limit ourselves to the convex setting in our experimentation. In fact, more experimentation in the convex setting would mostly be redundant with our proof of Theorem 1.
> >
> > If you have any remaining concerns, we are open to discussion.
> >
> > [1] Hwang, Youngsik, and Dong-Young Lim. "Dual Cone Gradient Descent for Training Physics-Informed Neural Networks." NeurIPS 2024.
> >
> > [2] Hazan, Elad. "Lecture notes: Optimization for machine learning." arXiv preprint arXiv:1909.03550 (2019).
> >
> > [3] Xu, Pan, et al. "Global convergence of Langevin dynamics based algorithms for nonconvex optimization." Advances in Neural Information Processing Systems 31 (2018).
> >
> > [4] Nie, Jiawang, and Zi Yang. "The multi-objective polynomial optimization." Mathematics of Operations Research (2024).

---

> ### Public Comment · ~Dongyoung_Lim1 · 2024-11-24
> **Public comment**
>
> Coercivity can be expressed in various forms. For instance, strong convexity satisfies coercivity. The useful property of coercivity is that it ensures the existence of an optimal solution while excluding cases where solutions are asymptotically attained at infinity.
>
> I believe that multi-objective learning is a highly significant field for the future, and I am personally very interested in it. I also think your research has the potential to make a impactful and significant contribution in this direction. Since you’ve submitted a paper to an AI conference, I raised questions from the perspective of the community’s readers who might inquire about similar aspects. I hope you did not take this personally or find it offensive.  I will no longer engage further in this discussion.
>
> I sincerely find your research interesting and believe it offers meaningful contributions. I wish you the best of luck with a successful review process.

---

### Note · Authors · 2025-01-22

I have read and agree with the venue's withdrawal policy on behalf of myself and my co-authors.